

# On the descent of the Alpine south foehn

Lukas Jansing[1], Lukas Papritz[1], and Michael Sprenger[1]

[1]Institute for Atmospheric and Climate Science, ETH Zürich, Zurich, Switzerland

**Correspondence:** Lukas Jansing (lukas.jansing@env.ethz.ch)

**Abstract.** When foehn winds surmount the Alps from the south, they often abruptly and vigorously descend into the leeside valleys on the Alpine north side. Scientists have long been intrigued by the underlying cause of this pronounced descent. While mountain gravity waves provide a modern theoretical foundation to explain the phenomenon, the descent of the Alpine south foehn has, so far, not been studied quantitatively and for a series of real-case events. To fill this research gap, the present
study employs kilometer-scale numerical simulations, combined with online trajectories calculated during model integration, to investigate the descent process with unprecedented detail. Adopting the Lagrangian perspective, the locations of descent are explicitly identified and the key characteristics are determined, thereby encompassing foehn regions spanning from the Western to the Eastern Alps.

In the first part of the study, we find the descent of foehn air parcels to be primarily confined to distinct hotspots in the
immediate lee of local mountain peaks and chains, underlining the fundamental role of local topography in providing a natural anchor for the descent during south foehn. Consequently, the small-scale elevation differences of the underlying terrain largely determine the magnitude of the descent. Combined with the fact that the descent is mostly dry-adiabatic, these results suggest that the descending motion occurs along downward-sloping isentropes generated by gravity waves.

The second part of the study aims to elucidate the different factors affecting the descent on a local scale. To this end, a partic-
ularly prominent hotspot situated along the Rätikon, a regional mountain range adjacent to the Rhine Valley, is examined in two detailed case studies. During periods characterized by intensified descent, local peaks along the Rätikon excite gravity waves, consequently leading to the descent of air parcels into the northern tributaries of the Rätikon and into the Rhine Valley. The two case studies reveal that different wave regimes, including vertically propagating waves, breaking waves, and horizontally propagating lee waves, are associated with the descent, indicating the lack of a preferential wave regime that is most conducive
for descent along the Rätikon. In addition to gravity waves, other effects likewise influence the descent activity. First of all, a topographic concavity deflects the near-surface flow and thus promotes strong descent of air parcels towards the floor of the Rhine Valley. Secondly, nocturnal cooling can inhibit the formation of pronounced gravity waves and thus impede the descent of foehn air parcels into the valley atmosphere.

In summary, this study approaches a long-standing topic in foehn research from a new angle. Using online trajectories,
the descent of foehn is explicitly identified and quantified, encompassing multiple case studies and a wide range of different foehn regions. The findings highlight the benefits offered by the Lagrangian perspective, which not only complements but also substantially extends the previously predominant Eulerian perspective on the descent of foehn.



# 1   Introduction

A major fraction of Earth's surface is characterized by complex terrain (Rotach et al., 2014). Downslope winds, forming in
the lee of orographic obstacles, therefore constitute an ubiquitous phenomenon of mountain meteorology (e.g., Smith, 1979).
While these winds are referred to as *foehn* in the Alps, they are given different local names in numerous regions worldwide and
have already been extensively documented (e.g., Raphael, 2003; Elvidge et al., 2014; Muñoz et al., 2020; Kusaka et al., 2021).
Foehn winds are well-known for their typical characteristics: Their onset in a valley is usually marked by an abrupt increase in
temperature and a decrease in relative humidity, while the wind and gust speeds pick up markedly (e.g., Richner and Hächler,
2013; Sprenger et al., 2016).

In the Alpine region, foehn winds are infamous for their beneficial, yet even more for their adverse impacts. Folkloristic
narratives blame the foehn for a range of negative effects on human health, such as insomnia, migraine, and a general discomfort
for parts of the population (e.g., Strauss, 2007). Besides, the gale-force winds can damage buildings and forests (e.g., Stucki
et al., 2015). They also pose a danger to aviation and cable car operators (Richner and Hächler, 2013) and notoriously accelerate
snowmelt over mountainous regions (e.g., Streiff-Becker, 1930). Furthermore, the occurrence of south foehn has been linked
to peaks in the ozone concentration in northern foehn valleys (Baumann et al., 2001; Seibert et al., 2000). In the Alps, the
arguably most hazardous impact of foehn concerns its potential to ignite and spread forest fires (Zumbrunnen et al., 2009;
Wastl et al., 2013; Pezzatti et al., 2016; Mony, 2020). In the polar regions, foehn flows have been found to enhance surface
melt and, for example, increase melt rates over Antarctic ice shelves (e.g., Elvidge et al., 2020; Zou et al., 2021). Similarly,
foehn effects are attributed a role in surface melt over the Greenland Ice Sheet (Mattingly et al., 2020, 2023).

Owing to their fierce nature and the associated impacts, foehn winds have attracted the interest of scientists going back
to the 19th century (e.g., Hann, 1866). Two key questions were posed early on (e.g., Steinacker, 2006) and heavily debated
throughout: Why is the foehn so extraordinarily warm when arriving in the valleys? And why does the potentially warmer
foehn air descend from aloft to replace potentially colder air in the valleys? The first of these questions (i.e., the warming)
lately received a lot of attention by scientific research (e.g., Elvidge and Renfrew, 2016; Miltenberger et al., 2016; Kusaka et al.,
2021; Jansing and Sprenger, 2022). Interestingly, the second question (i.e., the descent), while likewise being an archetypal
feature of foehn flows, has not received the same consideration in recent work. Accordingly, this paper shifts the focus towards
the descent of the foehn.

As mentioned above, the physical cause for the descent was in the epicenter of the early scientific debates on foehn. Over
the course of the 19th and the 20th century, a whole range of "foehn theories" was proposed. Some researchers attributed an
active role to the foehn flow (e.g., Wild, 1868; Streiff-Becker, 1930), while others interpreted the foehn descent as a passive
replacement flow when air is aspirated out of the valleys upon the approach of a synoptic low-pressure system (e.g., Billwiller,
1878; Ficker, 1931). With the *solenoid theory*, Frey (1945) proposed the first fluid-dynamical explanation of the foehn descent.
However, according to Richner and Hächler (2013), it remains unclear whether the solenoid field is, in fact, an effect of the
foehn rather than its cause. Rossmann (1950) and Schüepp (1952), in turn, assumed that the foehn wall might inherit a key
role for the downward acceleration: As air starts to descend downstream of the crest, evaporative cooling induces a negative



buoyancy compared to the environment, which would result in descending motion (the *waterfall theory*). Yet, since not all foehn events are accompanied by clouds on the Alpine south side, this hypothesis might at most explain a downward acceleration of the flow for some cases. For further details on the different foehn theories, it is referred to the existing literature (e.g., Lehmann, 65   1937; Gubser, 2006; Steinacker, 2006; Richner and Hächler, 2013; Sprenger et al., 2016).

An alternative concept to explain the dynamics of foehn was introduced by Schweitzer (1952). He used the analogy to a shallow layer of water flowing in a canal to explain foehn as a flow transitioning from subcritical to supercritical state. During the Mesoscale Alpine Programme (MAP; Bougeault et al., 2001), the applicability of the hydraulic analogue to gap flows in the Wipp Valley was extensively investigated. Indeed, several features reminiscent of hydraulic flow, like a transition of the 70   flow regime and hydraulic jumps, were observed and modelled during the campaign (Flamant et al., 2002; Gohm and Mayr, 2004; Armi and Mayr, 2007). With regard to the descent of the gap flow, Mayr et al. (2007) stated that *"Buoyancy forces as used in hydraulics are the key mechanism behind the descent."* In other words, potentially colder air upstream of a gap will flow downwards until reaching the level of neutral buoyancy. Mayr and Armi (2008) even argue that potential temperature differences between upstream and downstream air masses are the prerequisite for a cross-barrier flow to descend and form a 75   foehn. The conceptual model of hydraulics is well-suited to explain the dynamics of *shallow foehn*, where the southerly flow is restricted to levels below the crest. If, however, a deep layer of cross-Alpine flow is present (*deep foehn*), mountain gravity waves can substantially modify the foehn flow (Drobinski et al., 2007; Mayr et al., 2007). In fact, orographic gravity wave theory provides an alternative explanation for the descending motion of air and is therefore introduced next.

Largely separate from the above-mentioned foehn theories, mountain gravity waves were discovered and studied using 80   theory, observations, and later on also numerical models (e.g., Blumen, 1990). Queney (1948) was among the first to apply gravity wave theory to explain the strong acceleration of downslope winds. Essentially, the downslope motion in the lee of a mountain can be interpreted as an adiabatic descent along deflected isentropes. This deflection is caused by the orographic drag exerted on the atmospheric flow. Later on, researchers found two theories how gravity waves potentially amplify downslope flows. On the one hand, Klemp and Lilly (1975) suggested an amplification process based on the partial internal reflection and 85   constructive superposition of propagating gravity waves, inducing strong surface winds downstream of the obstacle. On the other hand, it was found that, under certain conditions, gravity waves can become convectively unstable, which is commonly referred to as wave breaking (e.g., Durran, 1990). The breaking region then acts as an internal reflector (i.e., a critical level) on upward-propagating gravity waves (Clark and Peltier, 1977). Consequently, a strong surface response in the form of a downslope windstorm emerges within the resonant cavity. It can thus be concluded that foehn *"... is an intrinsic feature of* 90   *mountain gravity waves"* (Elvidge et al., 2020).

During MAP, the role of gravity waves for the foehn flow in the Rhine Valley and the Wipp Valley was also investigated. In the Rhine Valley, strong downward motion and a modulation of the near-surface flow by the gravity waves aloft were diagnosed during Intensive Observation Period (IOP) 12 (Drobinski et al., 2003). During IOP 10, the gravity waves excited by the surrounding topography propagated into the region of the valley axis and their amplification concurred with striking 95   maxima in the low-level wind field near Vaduz (Zängl et al., 2004a). An analogous acceleration of low-level winds due to large-amplitude gravity waves has been reported for the Wipp Valley using idealized (Zängl, 2003) and real-case simulations



(Gohm et al., 2004; Zängl et al., 2004b). Interestingly, a further case study of a south-foehn event indicated that also trapped lee waves might be related to the occurrence of severe near-surface winds (Zängl and Hornsteiner, 2007). In this case, the descending motion would be caused by downward sloping isentropes on the upstream side of the wave troughs. In essence, the MAP findings highlighted the vital importance of mountain waves for the meso- and small-scale characteristics and the evolution of the foehn flow (Drobinski et al., 2007).

However, in order for the foehn flow to actually penetrate down to the surface of the valleys, gravity wave amplification is not the only decisive factor. In addition, the frequently present cold-air pools within Alpine valleys need to be eroded in order for the foehn air to reach the surface. The literature presents three mechanisms that potentially support the erosion of preceding cold-air pools, namely bottom-up erosion by solar radiation and associated surface sensible heat fluxes, top-down erosion by shear-induced turbulence, or displacement of the cold-air pool, for example by gravity waves (e.g., Flamant et al., 2006; Haid et al., 2020). In the recent field experiment named *Penetration and Interruption of Alpine Foehn* (PIANO), the oftentimes transient nature of foehn breakthrough in the region of Innsbruck was studied comprehensively. During IOP 2, a first and brief penetration of foehn air to the surface in the afternoon hours was attributed to the diurnal heating and resulting destabilization of the cold-air pool from the bottom (Haid et al., 2020). This mechanism is also made responsible for the breakthrough of a foehn event in the Sierra Nevada (Mayr and Armi, 2010). Many Alpine foehn stations exhibit a strong daily cycle in the climatological foehn frequency with a peak during the midday and afternoon hours (Mayr et al., 2007; Gutermann et al., 2012), providing further indication that this process is of key importance. The combined evidence from observations and a large-eddy simulation for the second stage of PIANO IOP 2, in turn, demonstrate the relevant contribution by shear-induced turbulence in weakening the nighttime cold-air pool east of Innsbruck (Umek et al., 2021), while the final breakthrough was related to cold-air pool displacement by the foehn flow (Haid et al., 2020). The multi-case analysis conducted by Haid et al. (2022) additionally emphasized the role of shear-induced instabilities in generating cold-air pool heterogeneity and turbulent warming. It is to note that the resolution of the mesoscale simulations used in the present study (cf. Sect. 2.1) do not suffice to explicitly study foehn-cold-air pool interactions. Instead, the descending motion from crest levels into the valley atmosphere will be studied.

Considering the body of literature on the Alpine foehn, several aspects related to the descent of foehn air remain open. For example, Zängl et al. (2004a) found a local maximum in the surface potential temperature field north of Vaduz and attributed it to the local sourcing of air from higher levels, descending along the northern slope of the adjacent mountain chain (the Rätikon). They hypothesized that upstream flow splitting at the Seez Valley junction promotes the descent of air from aloft by reasons of continuity. In general, the MAP results revealed an often substantial downvalley increase in potential temperature along the Rhine Valley during foehn, which cannot merely be explained by vertical turbulent mixing; instead it points towards the direct descent of higher-level, potentially warmer air (Drobinski et al., 2007). It is yet to be answered whether the descending motion always occurs in the same region of the Rhine Valley during south foehn. Analogously to Zängl et al. (2004a), two additional studies focusing on the same event qualitatively identified several regions of descent along the Wipp Valley (Gohm et al., 2004; Zängl et al., 2004b). Beyond these two valleys, it is presently unknown where locations of preferential descent are located in other foehn regions north of the Alps. In a recent study, Saigger and Gohm (2022) showcased that a combination of the Eulerian





and Lagrangian perspective can be advantageous to investigate the descent of foehn air. Using trajectories during a northwest-foehn event, they explicitly illustrated how air parcels descend into the Inn Valley due to a gravity wave. However, as of yet, descending air parcels have not been systematically linked to mountain gravity waves during south-foehn events. It is also unclear if the descending motion more often occurs within horizontally propagating lee waves, vertically propagating waves, or whether breaking gravity waves play a substantial role. Another interesting feature emphasized by Saigger and Gohm (2022) concerns the presence of a rotor circulation below trapped lee waves in the Inn Valley. The rotor enables air parcels to penetrate further downward into the valley atmosphere. It remains open whether such a flow pattern is peculiar to northwest foehn in the Inn Valley, or whether it can also be observed in other foehn regions. Finally, it is largely unknown how descending air parcels can be characterized in terms of fundamental properties such as the typical magnitude of the foehn descent, or the potential influence of diabatic processes. While Saigger and Gohm (2022) reasoned that the penetration of the foehn flow in their event was not significantly influenced by leeside diabatic processes, Zängl (2006) argued that evaporative cooling suppresses the formation of large-amplitude gravity waves and thus also the wave-induced subsidence.

In summary, a qualitative linkage of descending motion to gravity waves and low-level wind maxima in the Rhine Valley and the Wipp Valley has been established during MAP IOP 10 (Gohm et al., 2004; Zängl et al., 2004a, b). Using the Lagrangian perspective, Saigger and Gohm (2022) explicitly illustrated that descending foehn air parcels are tied to gravity waves. However, there exists no extensive investigation of the foehn air descent for other foehn regions besides the above-mentioned valleys. Previous research thus motivates us to formulate the following questions:

1. Where do air parcels descend during south foehn?

2. How can the descent be characterized?

3. What governs the descent on a local scale?

To address these open questions, we employ a series of mesoscale hindcast simulations for 15 Alpine south-foehn events. Within the last years, the continuous technical progress allowed the grid resolution of mesoscale simulations to be refined to the kilometer-scale. Though not fully, such resolutions at least partly resolve individual foehn valleys (e.g., Jansing and Sprenger, 2022) and therefore allow air parcel trajectories to be computed over complex terrain. Accordingly, an increasing number of recent studies invoked a Lagrangian view to study foehn flows in different regions of the world (e.g., Elvidge and Renfrew, 2016; Miltenberger et al., 2016; Kusaka et al., 2021; Jansing and Sprenger, 2022; Saigger and Gohm, 2022). In alignment, we likewise adopt the Lagrangian perspective. While most of the recent studies put their key emphasis on the warming mechanisms, we will focus on the descent of foehn air, which is particularly feasible since we calculated online trajectories (see Sect. 2.2). These trajectories explicitly resolve the descending motion of air parcels in the lee of major mountain barriers.

In the following, the used datasets (model hindcasts, online trajectories) and the Lagrangian method to identify descending motion along the pathways of air parcels are presented (Sect. 2). Subsequently, a spatial analysis of the descent is conducted in Sect. 3, including a localization of the descent and a quantification of kinematic and thermodynamic properties of the descending air parcels using the extensive set of foehn trajectories covering regions from the Western Alps to the Eastern Alps.




Thereafter, a particular region of strong descent (the aforementioned Rätikon), referred to as a hotspot, is scrutinized in Sect. 4. Two case studies serve to describe the ambient conditions leading to a temporally varying descent of foehn air parcels. Finally, Sect. 5 discusses the main findings of the study with respect to previous literature and Sect. 6 summarizes the key results.

## 2 Data and methods

### 2.1 COSMO simulations

To investigate the descent of the Alpine south foehn, we performed numerical simulations using the COnsortium for Small-Scale MOdelling (COSMO) model. COSMO is a non-hydrostatic, mesoscale model that solves a fully compressible formulation of the governing thermo-hydrodynamical equations (Steppeler et al., 2003; Baldauf et al., 2011; Schättler et al., 2021). The model is discretized on a structured grid with terrain-following vertical coordinates (Schär et al., 2002; Leuenberger et al., 2010) and is integrated using a split-explicit third-order Runge-Kutta scheme (Wicker and Skamarock, 2002). Besides the dynamical core, several parameterizations for subgrid-scale physical processes are used in the setup for this study. While radiation is described using a $\delta$-two-stream scheme (Ritter and Geleyn, 1992), a single-moment bulk microphysics scheme with five prognostic species is employed (Reinhardt and Seifert, 2006). Vertical turbulent mixing and surface transfer are parameterized with a prognostic turbulent kinetic energy (TKE) equation for the turbulence closure (Mellor and Yamada, 1982; Raschendorfer, 2001). To dampen small-scale noise and ensure numerical stability, horizontal mixing is performed using 4th order horizontal diffusion (Xue, 2000; Doms and Baldauf, 2021) and horizontal nonlinear Smagorinsky diffusion (Baldauf and Zängl, 2012). Soil processes are parameterized with a multilayer soil model (Heise et al., 2006).

The present study performed hindcast simulations using a model setup that closely follows the operational COSMO-1 setup used by the Swiss national weather service (MeteoSwiss). The computational domain of COSMO-1 encompasses the full Alpine range (1158 × 774 grid points; see domain in Fig. 1a). The simulations were conducted with a horizontal grid spacing of 1.1 km and 80 vertical levels, with the lowest half-level located 10 m above ground level (AGL), and were integrated using a time step of 10 s. Subgrid-scale processes were parameterized except for subgrid-scale orographic drag and convection, which were assumed to be sufficiently resolved. Running the model without any convection parameterization is fortified by a sensitivity study of Vergara-Temprado et al. (2020), where they identified a similar performance of COSMO at 2.2 km horizontal grid spacing for both explicit convection and parameterized shallow convection. Therefore, and in alignment with the operational model setup of MeteoSwiss, all convection parameterizations were switched off for the simulations performed. The necessary initial and boundary conditions to drive the model were derived from operational COSMO-1 analyses provided by MeteoSwiss. As the analyses and the hindcasts are of the same resolution, this setup prevents the necessity of additional spin-up time at the beginning of the model integration.

An overview over the case studies (simulated period, observed foehn period, output frequency of 3D fields, model version employed, predominant foehn type) is provided in Table A1. The events have been selected to represent the entire spectrum of different foehn types (Jansing et al., 2022). For most events, the model is initialized 6 h prior to foehn onset at Altdorf (a well-known Swiss foehn location; e.g., Richner et al., 2014), which is diagnosed using the enhanced version of the Dürr



**Figure 1.** (a) Model domain of the COSMO-1 hindcasts (red frame) and model topography (colormap). (b) Trajectory starting points for all simulations except for the gegenstrom-foehn events. The green dots indicate the horizontal starting positions used for all of these events, while the blue dots denote additional starting positions not used for the Nov 2016 and Feb 2017 events (see text for further explanation). Additional geographic features comprise the 50 km distance polygon (violet contour), the innermost closed 1500 m contour of the Alps (blue), additional 1500 m contours (orange), the Alpine crest line (green) and the cross section used for the trajectory selection (the "foehn cross section"; pink line with black crosses). (c) same as (b) but for the two gegenstrom-foehn events (Nov 2019 (1) and Feb 2020).

index (Dürr, 2008) aggregated to hourly resolution (Jansing et al., 2022). Accordingly, the simulations run for 6 h after foehn cessation at Altdorf. For more details on the case study selection and the exact model setups, the reader is referred to Appendix A1.

As one of the first and only atmospheric models, COSMO has been ported to graphical processing units (GPUs) to leverage the performance advantages from modern hybrid computing architectures (Fuhrer et al., 2014; Leutwyler et al., 2016). Ac-




cordingly, most of the hindcasts have been conducted using the enhanced, GPU-capable version of COSMO (see Appendix A1).

## 2.2 Online trajectories

The COSMO model offers the option to calculate Lagrangian air parcel trajectories along with model integration (Miltenberger et al., 2013). These online trajectories are compiled within the framework of the NWP simulation, thereby making use of the prognostic 3D wind field at every native model time step (10 s). Both truncation and interpolation errors, two of the main error sources associated with the computation of trajectories, can be reduced when increasing the spatial and temporal resolution of
the driving wind field. Online trajectories are thus of superior accuracy compared to the more traditional offline trajectories, which is particularly valuable when investigating non-stationary flows over complex terrain (Miltenberger et al., 2013). Since the numerical model integrates the governing equations forward in time, the online trajectory module is likewise limited to the calculation of forward trajectories. Therefore, it is imperative to release air parcels within all upwind regions that potentially contribute to the foehn flow within northern Alpine valleys. Consequently, trajectories have been started in extensive 3D
latitude-longitude boxes on the Alpine south side. The horizontal extent of these boxes varies between the different simulations (Figs. 1b,c and Table A2). Owing to the performance benefit of the GPU-enabled online trajectory module, which has been used for all simulations except Nov 2016 and Feb 2017, the number of trajectories per starting time has been increased for these simulations (compare green to blue dots in Fig. 1b). The different starting boxes for Nov 2019 (1) and Feb 2020 (Fig. 1c) compared to the other events (Fig. 1b) are motivated by the fact that these two events belong to the so-called gegenstrom-foehn
type (see Table A1). During such events, a strong zonal large-scale flow causes a more westerly origin of foehn air parcels compared to other foehn types (Jansing et al., 2022). For more details on the exact setup of the online trajectory module, the reader is referred to Appendix A2.

Trajectories are selected if they, at first, intersect with the Alpine crest line (see green line in Figs. 1b and c) and, afterwards, intersect with a cross section in the lee of the Alps (the pink line in Figs. 1b and c; hereafter referred to as the "foehn cross
section") into northerly direction and below 2500 m above mean sea level (AMSL). This selection procedure ensures that only trajectories traversing the Alpine ridge and, subsequently, diving into northerly foehn regions, are classified as foehn trajectories.

### 2.3 Descent identification

In order to identify locations of strong descent, an algorithm is applied to each selected foehn trajectory (illustrated in Fig. 2):

1. Local maxima and local minima in trajectory altitude are identified by comparing each trajectory altitude to its neighboring values, whereby the prominence needs to exceed 30 m. The prominence of a local maximum is defined as the height difference with respect to the lowermost closed contour line encompassing the respective maximum. The prominence of a local minimum, in turn, is defined as the height difference between the minimum and the lower of the two surrounding





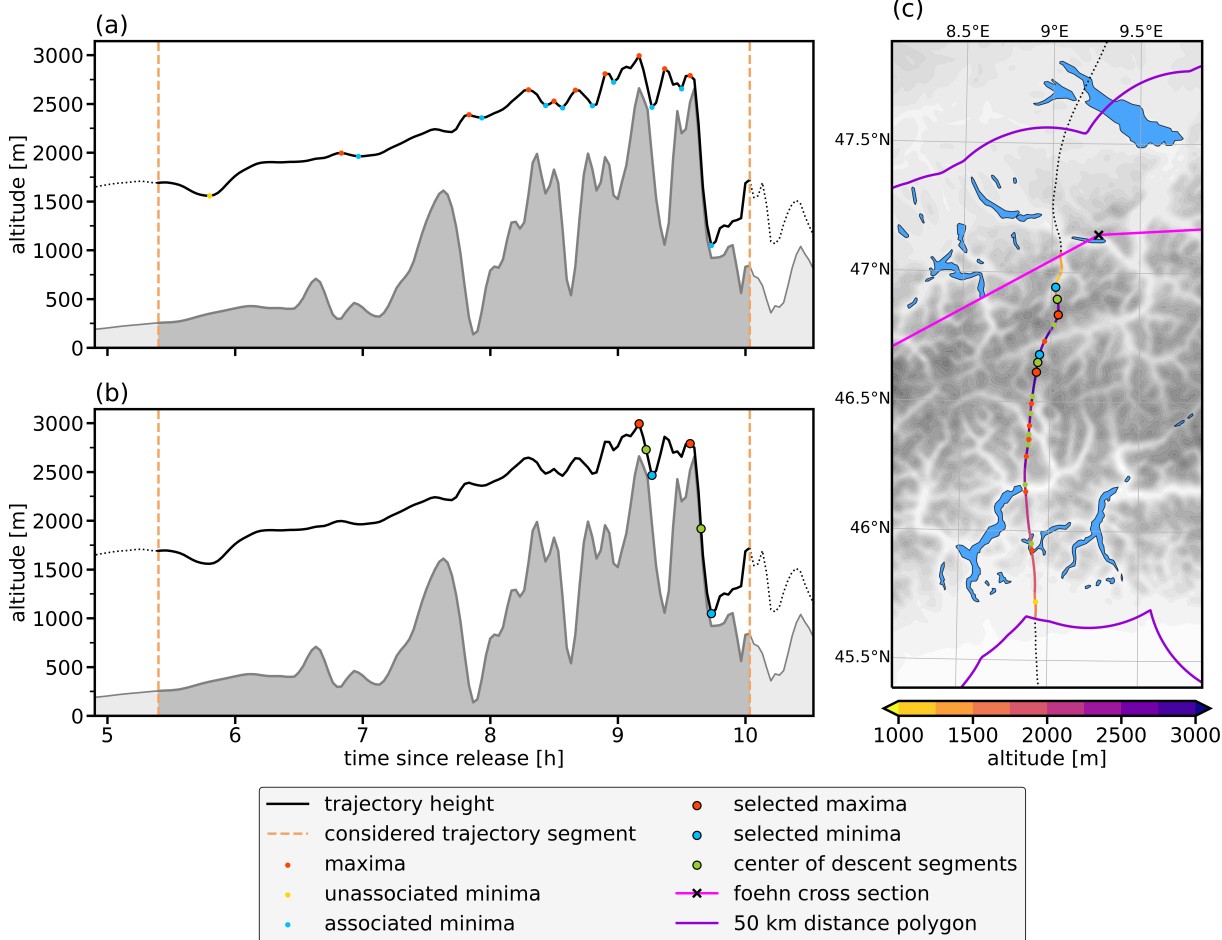

**Figure 2.** Illustration of the algorithm to identify segments of strong descent by means of an example trajectory from the Mar 2016 event. (a) depicts the altitude AMSL of the trajectory. The considered trajectory segment is highlighted by the dashed orange lines. Red dots mark identified local maxima, green dots the associated local minima, yellow dots the unassociated minima. (b) depicts the altitude AMSL of the trajectory and the selected descent segments fulfilling the magnitudinal and temporal criteria. Additionally, the center of each descent segment is indicated by a blue dot. (c) illustrates the horizontal pathway of the same trajectory, likewise indicating the local maxima, minima, as well as the selected descent segments.

maxima that encircle the minimum. Applying a threshold in prominence ascertains that altitudinal fluctuations of very

small amplitude are excluded in the further processing.

2. Each minimum is assigned to a preceding maximum to obtain pairs of maxima and minima, corresponding to *descent segments* (red and blue points in Figs. 2a and c). If a trajectory first reaches a local minimum, this minimum is unassociated with any maximum and not considered for further analysis (yellow point in Figs. 2a and c).





3. Only segments with a descent of at least 500 m within a maximum time span of 30 min are kept. This filtering ensures that only rapidly occurring descent segments of substantial vertical magnitude are selected.[1]. Enlarged red and blue points in Figs. 2b and c highlight the two selected descent segments that fulfill these criteria in the present example.

4. The center point of each descent segment is identified by linear interpolation between the adjacent maximum and minimum (green points in Figs. 2b and c).

This procedure is applied to each trajectory starting from the last trajectory time step prior to intersection with the Alpine polygon (violet contour in Fig. 2c) until the first time step after having intersected with the foehn cross section (pink line in Fig. 2c). Altogether, a total of 912 425 descent segments are identified in the 15 simulated cases. Note that multiple descent segments can be identified per trajectory.

## 3 Characteristics of foehn descent on the Alpine scale

### 3.1 Spatial extent of descent regions

Having identified strong descent along the air parcels (see Sect. 2.3), we first assess where along the Alpine arc the air parcels actually descend during south foehn. To this end, the center points of all descent segments (green dots in Figs. 2b,c) from the 15 foehn events are displayed in a two-dimensional, binned histogram (Fig. 3). In addition, one-dimensional histograms along the upper and right edges illustrate the variability in zonal and meridional direction, respectively. In all foehn regions, extending from the westernmost Haute-Savoie in France to the easternmost Ziller Valley in Austria, strongly descending air parcels are discernible. The number of descent segments however varies in between the different regions. It peaks in an area covering the central Swiss Alps and the Rhine Valley, while fewer air parcels descend in the Western and the Eastern Alps (see also histogram in upper part of Fig. 3). Note that the area spanning from the Central Alps to the Rhine Valley also corresponds to the region where the overall highest number of foehn trajectories is selected. On a smaller scale, peaks in descent activity emerge in proximity to the incisions of major foehn valleys, such as for example the Lower Valais, the Reuss Valley Valley, the Rhine Valley and the Wipp Valley (see locations of valleys in Jansing and Sprenger, 2022). This clearly highlights the relevance of major foehn valleys as preferential regions for strong descent of air parcels.

Although the identification method does not explicitly limit strong descent to regions north of the Alpine crest, descent indeed predominantly occurs within these areas (Fig. 3). This north-south gradient in descent activity is not surprising, as strong wave-induced descent is expected to occur, if at all, downstream of an orographic obstacle (e.g., Durran, 1990). Instances of strong descent south of the Alpine crest are primarily confined to local mountain chains and valleys that exhibit an east-west orientation (e.g., the Valtellina in Italy). These mountain chains act as a local barrier to the southerly flow, allowing for leeside descent despite being situated on the southern side of the Alpine crest.

At the regional to local scale, a striking feature concerns the uneven spatial distribution of descent segments (Fig. 3). Instead, pronounced maxima in descent activity emerge, which we refer to as *hotspots*. These maxima are strongly confined in space

---

[1]The sensitivity with respect to the thresholds in altitude and time has been tested (Fig. S4 in Supplement). See Sect. 3.1 for further details.





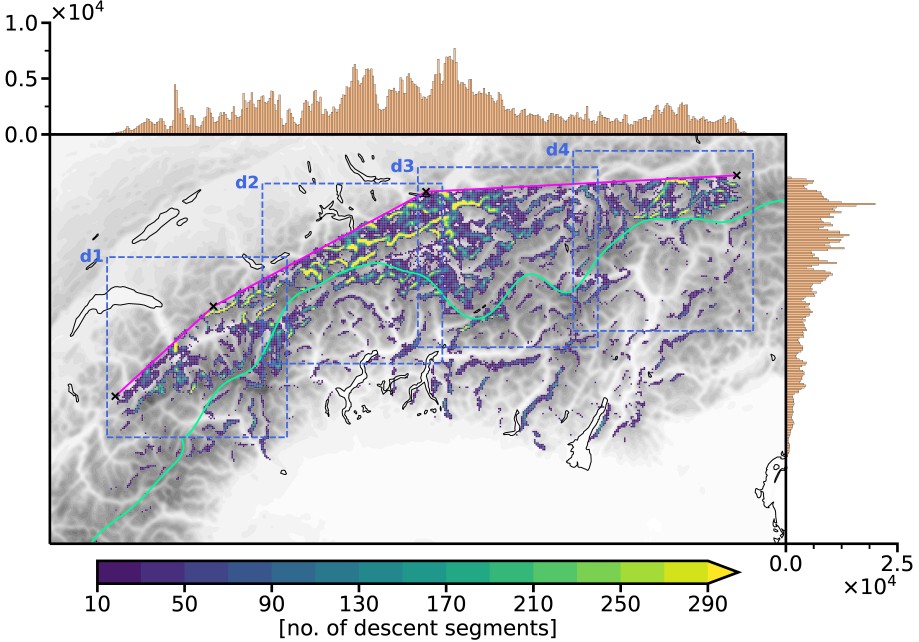

**Figure 3.** Number of descent segments from all foehn events within equally spaced 0.01° x 0.01° bins on the rotated latitude-longitude grid (colormap). Note that values below 10 are masked to enhance visibility. The pink line with black crosses indicates the foehn cross section, while the green line corresponds to the crest line (used for orientation). Additionally, the number of trajectories within each rotated latitude-longitude segment is shown by the means of two histograms along the upper and the right edges of the map, respectively. The dashed blue boxes (labelled d1 to d4) denote four subdomains that provide a zoomed view on the descent regions in Fig. S2 in the Supplement.

and frequently situated in the immediate lee of elongated orographic obstacles, resulting in a correspondingly elongated shape of many hotspots. This observation highlights the significant influence of local topography, which serves as a natural anchor for the downslope flow during foehn events. A minority of the hotspots, however, also extend across the valley floor of foehn valleys (e.g., Lower Valais, Reuss Valley, Rhine Valley). It can be speculated that gravity waves triggered by the surrounding mountain peaks propagate to the valley axes in these regions and force descending motion of air parcels towards the valley floor.

Such a mechanism has previously been attributed a role in enhancing low-level winds during a foehn event in the Rhine Valley (Zängl et al., 2004a). In this regard, it is particularly highlighted that the overall strongest hotspot (see peaks in histograms along the edges of Fig. 3) emerges along the mountain chain of the Rätikon, whose slopes face towards the Rhine Valley in the west and towards the Walgau in the north (see Fig. 7 for orientation). Zängl et al. (2004a) also described the northern slopes of the Rätikon as a preferential region for descending motion, a finding clearly corroborated by our Lagrangian analysis of

descent locations. Accordingly, this hotspot will be subject to more detailed investigations in Sect. 4.

Moreover, most identified descent regions are not located in the immediate vicinity of the Alpine crest, but rather in close proximity to the arrival locations of the respective air parcels (pink line in Fig. 3). However, there exist some noteworthy exceptions to this pattern, particularly in the upper Valais region. There, a considerable amount of air parcels descends into the





Valais and is presumably channelled downvalley. Such pathways of air parcels have also been observed during the Nov 2016

event (see in Jansing and Sprenger, 2022). Additionally, several hotspots emerge in the tributaries of the upper Rhine Valley. It

is worth noting that in this particular region, the main Alpine crest is situated at a greater distance from the arrival locations.

In the comprehensive analysis of descent locations, trajectories from all events were collectively considered to examine the

spatial variability in descent activity. An extended investigation of the case-to-case variability, along with a sensitivity analysis

involving different thresholds in altitude ($\Delta z$) and time ($\Delta t$), can be found in the Supplement (Sect. 1 and Figs. S3 and S4).

**3.2   Kinematic and thermodynamic characteristics of descent regions**

The preceding section elucidated the distinct spatial variability of the descending motion related to foehn, establishing preferen-

tial regions of descent. But do these regions diverge in terms of the characteristics (e.g., magnitude, thermodynamic evolution)

of the air parcels' descent, or do the air parcels descend uniformly across all foehn regions north of the Alps? To address this

question, the subsequent section explores the potential variability in the descent characteristics. In this regard, two different

types of characteristics are of particular interest. The *kinematic* characteristics encompass the vertical magnitude of the descent

segments ($\Delta z$) and their time span ($\Delta t$). The *thermodynamic* characteristics include the potential temperature difference ($\Delta \theta$)

and the specific humidity difference ($\Delta q_v$) of the end points to the start points of each descent segment. To investigate all of

these characteristics, one-dimensional histograms are used to illustrate their overall distribution and the range of typical values

(Figs. 4a,c and 5a,c) and two-dimensional binned histograms to highlight the spatial variability (Figs. 4b,d and 5b,d).

Focusing on the kinematic characteristics, it becomes evident that the descent of air parcels exhibits substantial variations,

both in terms of magnitude and time span. The frequency of descent segments decreases exponentially with increasing descent

magnitude (Fig. 4a). Air parcels thus rarely descend more than 1500 m within one single descent segment. The time span to

cover the descent ranges from 2 to 30 min, however most of the air parcels need about 4–10 min to descend (Fig. 4c).

The descent magnitude exhibits particularly strong spatial variability (Fig. 4b). Pronounced maxima emerge downwind of

some of the highest peaks of the Western Alps, such as the Mont Blanc massif or the Bernese Alps, where mean values even

exceed 1500 m. In stark contrast, the descent magnitude is substantially smaller in regions close to the Alpine crest (see green

line in Fig. 4b). These spatial differences result from variations in the local terrain characteristics that appear to be strongly

related to the descent: Since valley incisions tend to be less deep in regions closer to the crest, the elevation differences

between the local valleys and the surrounding mountain peaks are smaller, thus inhibiting descent of greater magnitude. In

most of the localized hotspot regions, the mean descent magnitude is between 750 and 1000 m. Therefore, the terrain does not

only determine the preferred locations for descent, but also to a large extent dictates its magnitude. This exact relation is further

illustrated in Fig. 5. The descent magnitude ($\Delta z$) is closely correlated to the change in the underlying topography ($\Delta topo$).

Descent of very large magnitude thus almost exclusively occurs along steeply sloping flanks on the lee side of the highest

Alpine peaks.

The time span needed to cover the descent segments likewise varies regionally (Fig. 4c). However, no clear linkage to the

local terrain characteristics emerges. While most regions exhibit a mean time span of approximately 7–11 min, some localized

regions, especially in the Central Alps, are characterized by shorter descent time spans. The magnitude and the time span are







**Figure 4.** Kinematic characteristics of foehn descent segments. (a) and (c) display histograms of the kinematic characteristics ($\Delta z$ and $\Delta t$). (b) and (d) show the same kinematic characteristics, but in two-dimensional binned histograms (as in Fig. 3). The bins are colored according to the mean characteristics within each bin. The dashed blue boxes (labelled d1 to d4) in panel (b) denote four subdomains that provide a zoomed view on the descent characteristics in Figs. S5 to S8 in the Supplement.

not clearly anticorrelated, meaning that rapid descent (small $\Delta t$) does not need to be co-located with descent of large magnitude (large $\Delta z$) and vice versa.

The clear correlation between the descent magnitude and the changes in the underlying topography strongly indicates that the downslope flow is often essentially terrain-following. It is likely that mountain gravity waves, anchored to local peaks, induce the descent of foehn air parcels. It can thus be assumed that variations in the descent magnitude across different regions are





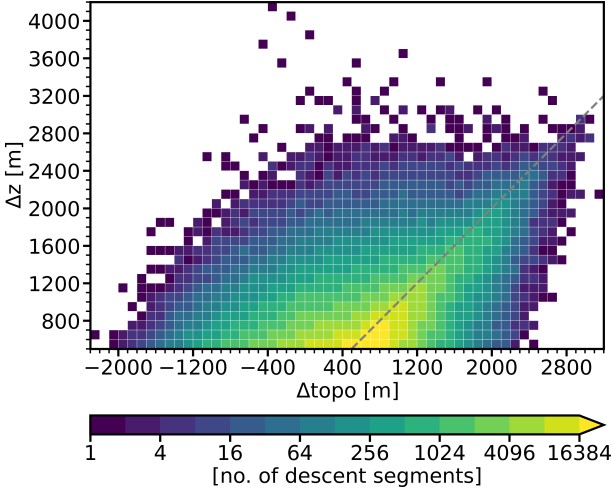

**Figure 5.** Relation of the descent magnitude ($\Delta z$) to the change in the underlying terrain ($\Delta topo$) in a two-dimensional histogram colored according to the number of descent segments within each bin. The 1:1 line is included as a gray dashed line. Note the exponential color scale.

caused by spatially varying wave amplitudes, which are, in turn, influenced by the local mountain height. However, other factors (e.g., impinging wind speed) certainly contribute to the variability in the wave amplitudes, which explains the evident spread

in Fig. 5. Furthermore, other local factors potentially affect the descent characteristics, and these will be further discussed in Sect. 4, thereby focusing on the hotspot along the Rätikon.

Subsequent to the kinematic, two thermodynamic characteristics are analyzed ($\Delta\theta$, $\Delta q_v$). The majority of descending air parcels experiences no noteworthy change in potential temperature (Fig. 6a). This finding implies that the descent happens approximately adiabatically. Consequently, the air parcels follow steeply downward-sloping isentropes in the lee of the moun-

tain peaks, which further substantiates the hypothesis of gravity-wave-induced descent. Nevertheless, the distribution of $\Delta\theta$ is slightly skewed towards negative values, revealing that a minor share of the air parcels experiences diabatic cooling. Similarly, most air parcels are not subject to specific humidity changes, yet a slight humidity uptake is registered for some of the descent segments (Fig. 6c). The minority of descent segments associated with diabatic cooling and a specific humidity increase are clearly co-located (Figs. 6b,d). They predominantly occur either south, or in small distance to the Alpine crest. On the southern

side of the Alps, the impinging air parcels during foehn oftentimes form clouds and precipitation. When these air parcels locally descend and therefore start to warm, the cloud and rain water at least partially evaporates, resulting in diabatic cooling and a specific humidity gain. This peculiarity in the thermodynamic characteristics of descending air parcels south of the Alpine crest might also explain the relatively low number of descent segments in these regions (Fig. 3) and their small magnitude (Fig. 4b): The evaporative cooling during the leeside descent potentially reduces the amplitude of the local gravity waves and thus

impedes stronger descent, an effect previously described by Zängl (2006). In contrast to the pattern south of the Alpine crest, a few regions north of the crest feature a minor increase in potential temperature. This diabatic heating is most probably caused by turbulent mixing within the stably stratified flow, as condensational heating cannot occur along descending air parcels and





**Figure 6.** Same as Fig. 4 but for the thermodynamic characteristics (potential temperature difference $\Delta\theta$ and specific humidity difference $\Delta q_v$).

radiative heating likely plays a minor role considering the short time spans of individual descent segments. Analogously, a local drying could be related to turbulent mixing of descending air parcels with drier air from higher levels.

## 4 The descent of foehn air parcels into the Rhine Valley

So far, the spatial variability in the descending motion of foehn air parcels and the associated characteristics were investigated on the Alpine to the regional scales. Strong descent turned out to be spatially confined to distinct hotspots in the lee of local



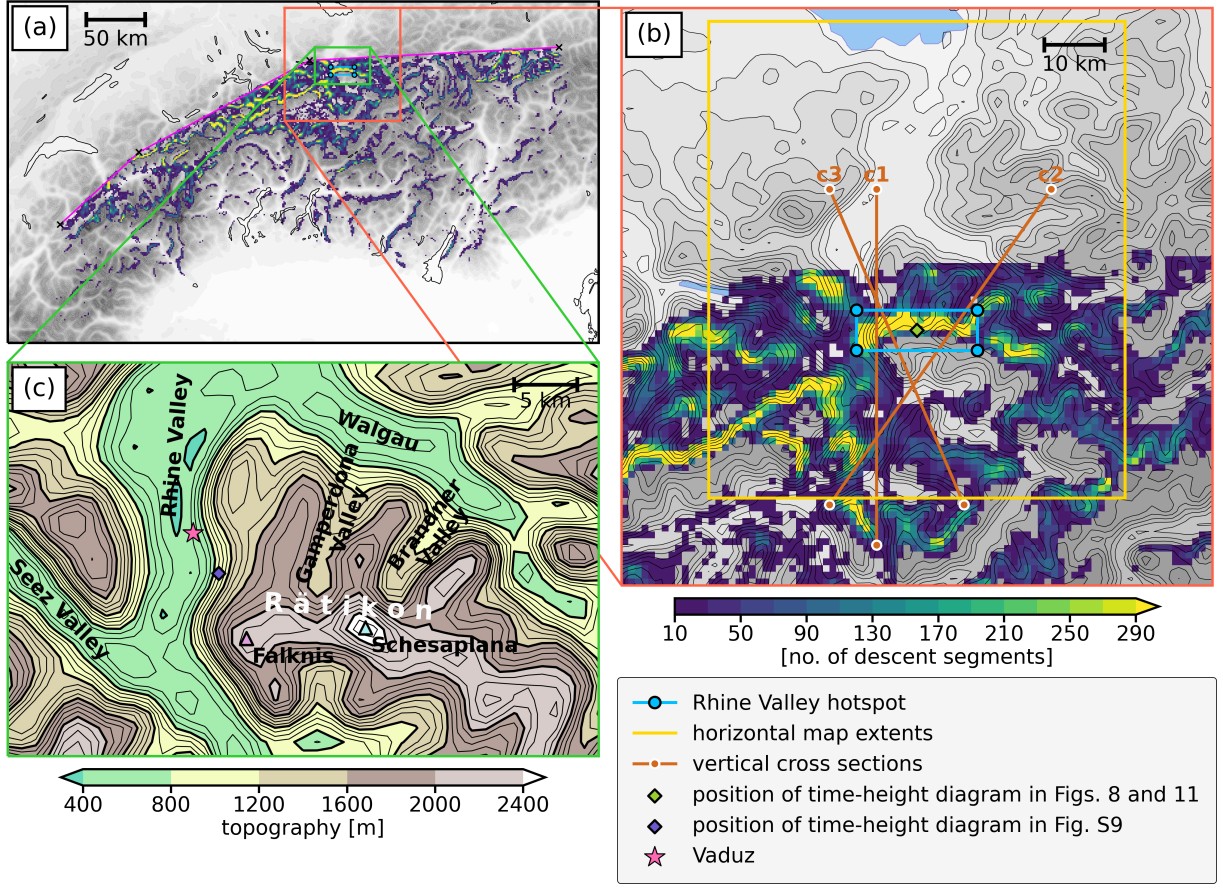

**Figure 7.** Geographic overview of the region in proximity to the Rhine Valley hotspot. (a) depicts the extended Alpine region that has been in focus in Sect. 3. (b) shows an enlarged map of the central and lower Rhine Valley with the blue box indicating the investigated hotspot. The yellow subregion marks the extent of the maps depicted in Figs. 9 and 12. The brown lines show the locations of the vertical cross sections (labelled c1 to c3), which are displayed in the two case studies (Figs. 9 and 12). The green marker depicts the location of the time-height diagrams in Figs. 8 and 11. (a) and (b) are both colored according to the number of descent segments within equally spaced bins (same as in Fig. 3), and include the topography of COSMO-1 in gray shading (and contours with 200 m spacing in (b)). (c) shows an enlarged contour map of the Rätikon mountain chain using the model topography (colormap and contours with 100 m spacing). Important valleys, mountain peaks and locations are labelled. The dark-blue marker shows the location of the time-height diagram in Fig. S9.

mountain chains and peaks. On this basis, the next goal of this paper is therefore to examine one of these hotspots, and especially the cause for its formation. This question is tackled by focusing on the hotspot along the Rätikon (hereafter denoted 350 as "Rhine Valley hotspot"). The selection of this hotspot is motivated by two reasons: First of all, it coincides with the location where the overall number of descent segments reaches its maximum (Fig. 3). Secondly, the northern slopes of the Rätikon close to Vaduz (pink star in Fig. 7c) have previously been identified as a preferential region for descending motion during MAP (Zängl et al., 2004a), which allows us to discuss our results in relation to existing literature. The following in-depth analysis of





the Rhine Valley hotspot will unravel the ambient atmospheric conditions related to strong descent and reveal the drivers of a
temporally varying descent activity by the means of two case studies (Sect. 4.1 and 4.2). Thereby, the Rhine Valley hotspot is
defined as a rectangular box (blue box in Fig. 7b) and all descent segments whose center points lie within this box are selected
for the analysis. For the two case studies, horizontal maps (yellow frame in Fig. 7b) and vertical cross sections (brown lines in
Fig. 7b) are additionally utilized to investigate the meteorological conditions in the immediate vicinity of the hotspot.

## 4.1 Feb 2017 case study

In the early afternoon hours of 27 February 2017, foehn broke through in Vaduz (see Jansing, 2023), which corresponds to
25 h since the event start.[2] In accordance with the observed onset of foehn at the surface, the first descending air parcels are
detected a few hours prior to foehn onset at Vaduz (Fig. 8b). Thereafter, the number of descent segments increases over time
and peaks after 37 h, followed by a sharp decrease and a transient break in descent activity, before a second period with a
lower number of descent segments is detected between 46 and 52 h since event start. The goal of the following section is to
explain this very distinct temporal evolution of the descent activity by linking it to the local atmospheric conditions during the
course of the event. To this end, four interesting time instants (see olive arrows in Fig. 8b) are selected and further investigated
using horizontal maps and vertical cross sections (25 h: onset of descent, but still weak descent activity; 37 h: peak descent
activity; 45 h: a temporary break; 49 h: secondary peak before final cessation). It is highlighted where and under which ambient
conditions air parcels descend in the hotspot region.

The Feb 2017 event is categorized as a deep-foehn event and occurred downstream of a broad upper-level trough and a
cold front that approached the Alpine region from the northwest (not shown). At first, weak to moderate west-southwesterlies
prevail at all levels in the region of the hotspot (Fig. 8a). After 20 h, the winds below 3 km start to blow from sector south.
Hence, the horizontal winds turn clockwise with height and a pronounced warming in the mid-troposphere sets in, presumably
due to warm-air advection and wave-induced subsidence. Accordingly at lower altitudes, gravity-wave activity might lead to
the strong downward motion (blue contours in Fig. 8a). Concurrently, the west-southwesterlies at mid- and upper-tropospheric
levels continuously intensify until 35 h since event start. In the lower troposphere, between 2–3 km, a layer of high stability
forms during this time period. In the time window of 30–37 h, a low-level wind maximum occurs within the region of the
stable layer, which aligns well with the peak period of descent activity (Fig. 8b). As the low-level winds reach their highest
intensity, the wave signal (downward motion) extends throughout the troposphere, indicating vertically propagating gravity
waves. Subsequently (at 40 h), the wind speeds below 3 km temporarily decrease, before a second maximum is detected at
50 h. This temporal evolution likewise aligns with the transient break and secondary peak in descent activity. Finally, the winds
below 4 km turn to northwest and isentropes rise to higher altitudes, indicating the arrival of the cold front. Overall, a very clear
correspondence of the large-scale winds above crest levels and the descent activity is identified (cf. Figs. 8a and 8b).

---

[2]The Feb 2017 event was simulated over an extended time period (see Appendix A1 and Table A1). Therefore, a longer period prior to and after the foehn
episode was captured by the simulation compared to other events. In the context of the case studies, the different time instants are given in "hours since event
start", whereas the event start and end are defined by the simulated period.



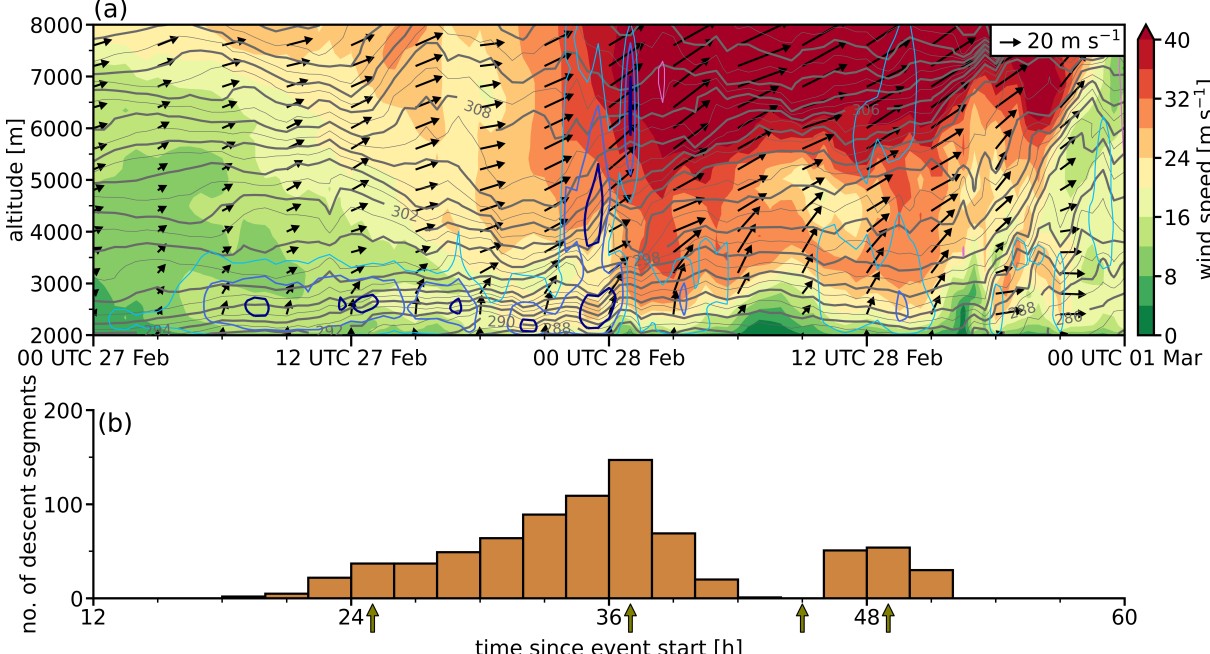

**Figure 8.** (a) COSMO-based time-height diagram of horizontal wind speed (colormap and arrows) in the center of the hotspot region for the Feb 2017 event (location of time-height diagram is indicated by green marker in Fig. 7b). Arrows pointing to the right correspond to eastward winds and arrows pointing upward to northward winds. Isentropes are indicated as gray contours with a spacing of 1 K. Additionally, vertical winds are displayed using blue and violet contours for negative and positive values, respectively (spacing of $1\,\mathrm{m\,s^{-1}}$; starting from $\pm 1.5\,\mathrm{m\,s^{-1}}$). (b) Temporal evolution of the number of descent segments in the Rhine Valley hotspot within two-hourly windows. The selected time instants for the horizontal and vertical cross sections shown in Figs. 9 and 10 are highlighted by olive arrows along the x-axis.

To further examine how the local conditions are related to the temporal evolution of the descent activity during the Feb 2017
event, horizontal maps and vertical cross sections during four time instants with a distinctively different number of descent segments (see olive arrows in Fig. 8b) are utilized, each of them indicating horizontal winds (arrows), vertical wind (colormap) and the positions of descent segments.

The first time instant, 25 h after event start, corresponds to the time the foehn reached Vaduz. While weak southwesterly winds prevail above crest height (Fig. 8a), elevated wind speeds at 2000 m AMSL (Fig. 9a) are restricted to the northern slopes
of the Rätikon and the upper Rhine Valley at this time. Two distinct groups of descending air parcels in the west and east of the hotspot area are discernible (Fig. 9a). The western air parcels start to descend along the northwestern slope of the Falknis peak (pink triangle in Fig. 9a). They reach levels of 1 km AMSL (not shown) when arriving over the Rhine Valley floor close to Vaduz. The eastern air parcels originate to the southwest of the Schesaplana and subside into the Brandner Valley (see Fig. 7c). Both groups of air parcels descend within regions of moderate downward motion ($2\text{–}4\,\mathrm{m\,s^{-1}}$). The two vertical cross
sections during the same time instant (Figs. 10a, b; see lines c1 and c2 in Fig. 7b), which traverse both descent regions in the lee of the Falknis and Schesaplana, unveil the presence of two gravity waves at low levels. These gravity waves emanate



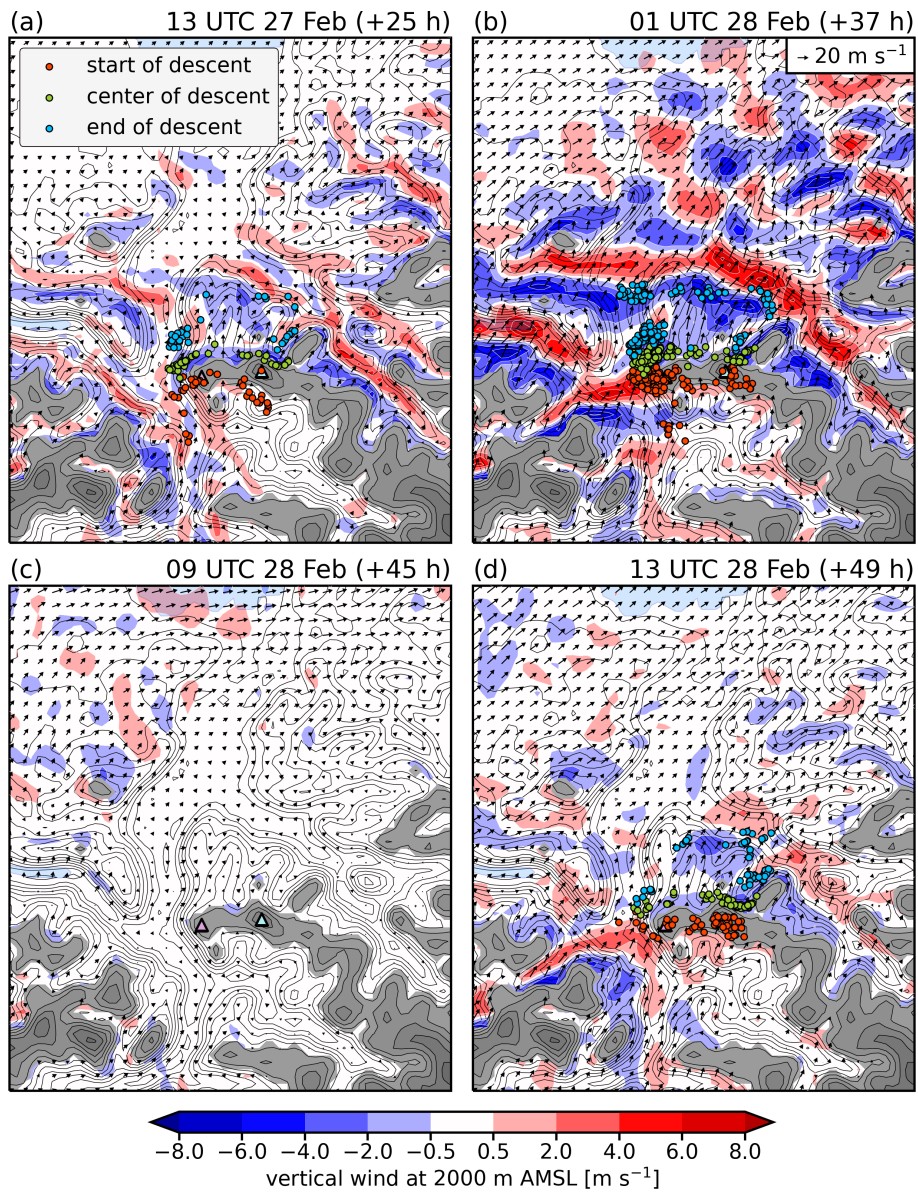

**Figure 9.** Vertical wind (colormap) and horizontal wind (arrows) at 2000 m AMSL for: (a) 13 UTC 27 February 2017; (b) 01 UTC 28 February 2017; (c) 09 UTC 28 February 2017; (d) 13 UTC 28 February 2017. Additionally, the figure shows start (red), center (green) and end positions (blue) of the trajectory descent segments that occurred within a two-hourly window centered around the displayed dates. The COSMO topography is included in the background for orientation (gray shading starting at 2000 m AMSL and contours with 200 m spacing). The peaks of the Falknis and the Schesaplana (see also Fig. 7c) are indicated as pink and light-blue triangles, respectively.

downwind of the two local peaks. They do not propagate vertically, instead both feature a convectively unstable region where the isentropes attain vertical orientation. Due to relatively weak impinging flow, the flow upstream of both peaks is, to a large





**Figure 10.** Vertical cross sections of vertical wind (colormap), isentropes (gray contours) and wind along cross sections (vectors). The topography is indicated by gray shading. The left column shows the cross section c1 (a, c, e, g), while the right column shows the cross section c2 (b, d, f, h). The four rows correspond to the four selected time instants (see also titles of each panel). Descent segments, which are located closer than 2 km to the cross section, are indicated as green dots.

extent, blocked and a nonlinear wave regime establishes. Such a regime is typically reminiscent of nonlinear phenomena, such
as wave breaking or hydraulic jumps (e.g., Durran, 1990).





Twelve hours later (37 h since event start), the peak in the descent activity has been reached (Fig. 8b). During this time instant, substantially stronger horizontal winds are observed at 2000 m AMSL (Fig. 9b). Accordingly, intense wave activity of larger amplitude is registered in the region of the Rhine Valley, as can be seen by alternating regions of strong upward and downward motion. The mountain waves emanate from local mountain peaks and chains, but are able to propagate horizontally away from
their source, as for example over the lower Rhine Valley. The majority of trajectories descends along the northwestern slope of the Falknis (Fig. 9b) within the gravity-wave induced downward motion in the immediate lee. Considering the two vertical cross sections at the same point in time (Figs. 10c, d), a striking feature constitutes the above-mentioned stable layer between 2.5 and 4 km AMSL. Pronounced vertical variations in static stability are known to inhibit vertical wave propagation (e.g., Jackson et al., 2013; Durran, 2015). Indeed, the gravity wave downwind of the Falknis seems to be partially trapped within the
stable layer below 4 km. Downstream of the main peak, further mountain waves are discernible. However, it remains unclear whether this wave activity actually originates from the Falknis or is primarily caused by secondary peaks to the north, which potentially excite additional gravity waves. Since the mountain wave in the lee of the Schesaplana is able to propagate vertically (Fig. 10d), the ambient conditions do not seem to clearly favor the formation of horizontally propagating lee waves.

In the region to the northwest of the Falknis, which is associated with the strongest descent activity at this time instant (37 h
since event start), the orography features a local concavity. The concave shape of the terrain redirects the low-level flow and, consequently, southeasterlies prevail close to the surface despite the southwesterlies at higher levels (see time-height diagram in Fig. S9a in Supplement). This peculiarity of the local terrain presumably deflects the descending air parcels along the northwestern slopes of the Falknis and and therefore promotes the descent into the valley atmosphere of the Rhine Valley.

The next highlighted time instant (45 h) corresponds to the time when a temporary break in descent activity is registered.
While the southwesterlies continue to blow in the middle and upper troposphere, the wind speeds dramatically decrease below 3 km (Fig. 8a) and the stratification increases (Figs. 10e,f). Below 2 km, the conditions within the Rhine Valley and along the slopes of the Rätikon are essentially calm. As a consequence of the weak winds at crest level, no notable wave activity and vertical motion is present (Figs. 9c and 10e,f), which explains the temporary break in descent activity. Based on horizontal maps of the low-level wind field of the region, a transient interruption of the foehn occurred (not shown).

In the early afternoon of 28 February (49 h since the start of the event), a second, weaker peak in descent activity was detected (Fig. 8b). Compared to four hours earlier, the southwesterly winds at 2 km AMSL slightly reintensified (Fig. 9d). As a result, a vertically propagating gravity wave forms downwind of the Schesaplana (Fig. 10h), transporting air parcels into the Brandner Valley. Interestingly, the descent activity is now primarily concentrated in this region. In the lee of Falknis, only a weak downslope flow is discernible (Fig. 10g). Minor changes in the local ambient conditions, such as local wind speed and
direction, or the absence of a stable layer, appear to inhibit the formation of a mountain wave or hydraulic jump structure as observed at earlier times (Figs. 10a, c). However, further scrutiny would be required to elucidate the conditions provoking the formation of gravity waves in the lee of the Falknis.

The investigation of these four time instants shows a clear correlation between the number of diagnosed descent segments and the presence and intensity of mountain gravity waves in this region. It is explicitly shown that foehn air parcels descend
within gravity-wave induced downward motion, confirming their key role in the descent of foehn air parcels. The observed wave



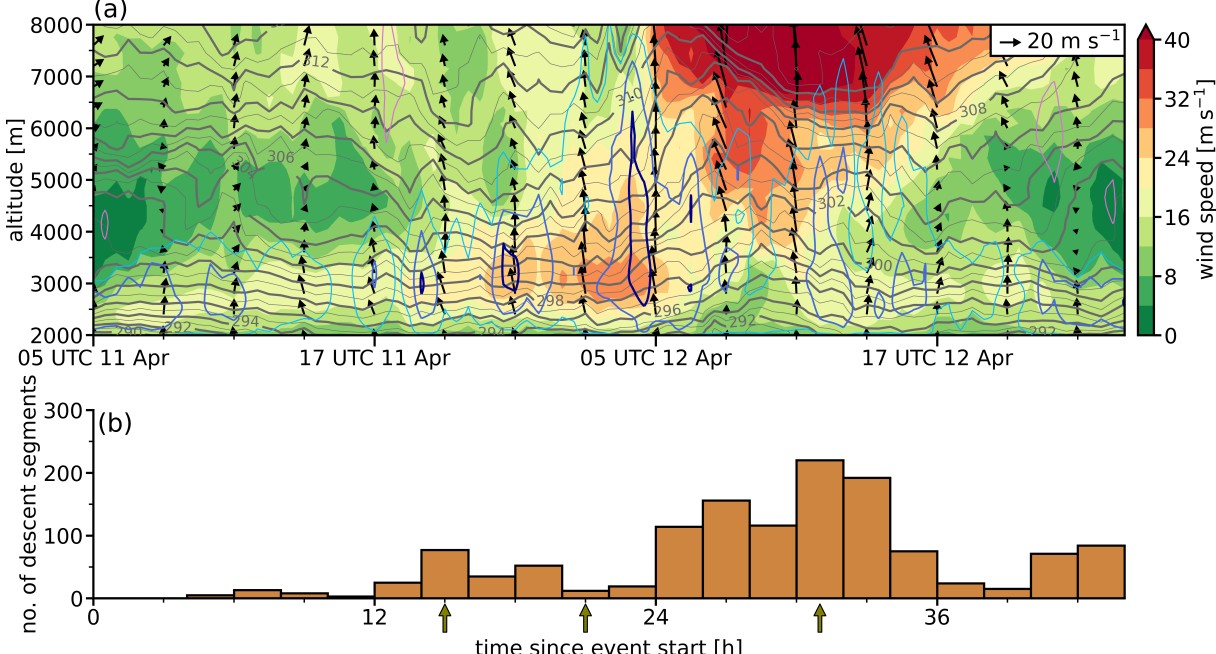

**Figure 11.** Same as Fig. 8 but for the Apr 2018 event.

amplitude is influenced by the wind speed and direction of the impinging flow. Furthermore, the local topographic concavity to the northwest of the Falknis seems to favorably steer descending air parcels downward into the boundary layer of the Rhine Valley.

## 4.2 Apr 2018 case study

In the following, a second case study of the Apr 2018 event is presented with the goal to further illustrate the diversity of descent characteristics and to identify differences with respect to the main findings of the previous section.

The synoptic situation during the Apr 2018 event was characterized by a cutoff low over Spain and southern France, instead of an upper-level trough as in the Feb 2017 case (not shown). The cutoff induced a synoptic environment conducive to the formation of south foehn (see also Jansing, 2023). Related to the synoptic weather evolution, winds in the middle to upper 445 troposphere above the hotspot region were relatively weak during most of the event, except for a period from 25 to 35 h, when strong southerly winds were detected at upper levels (Fig. 11a) due to the approach of the cutoff. Except for this period, the strongest winds were actually confined to a layer below 4 km. There, the horizontal winds blew from the south to southeast and reached their maximum 20–25 h after the start of the event at an altitude of 3 km. The lower troposphere was not only associated with stronger winds, but the stratification was also more stable compared to the middle and upper troposphere. Several periods 450 of stronger downward motion alternated with intermittent periods of weak vertical motion, potentially due to variability in the wave-induced subsidence.




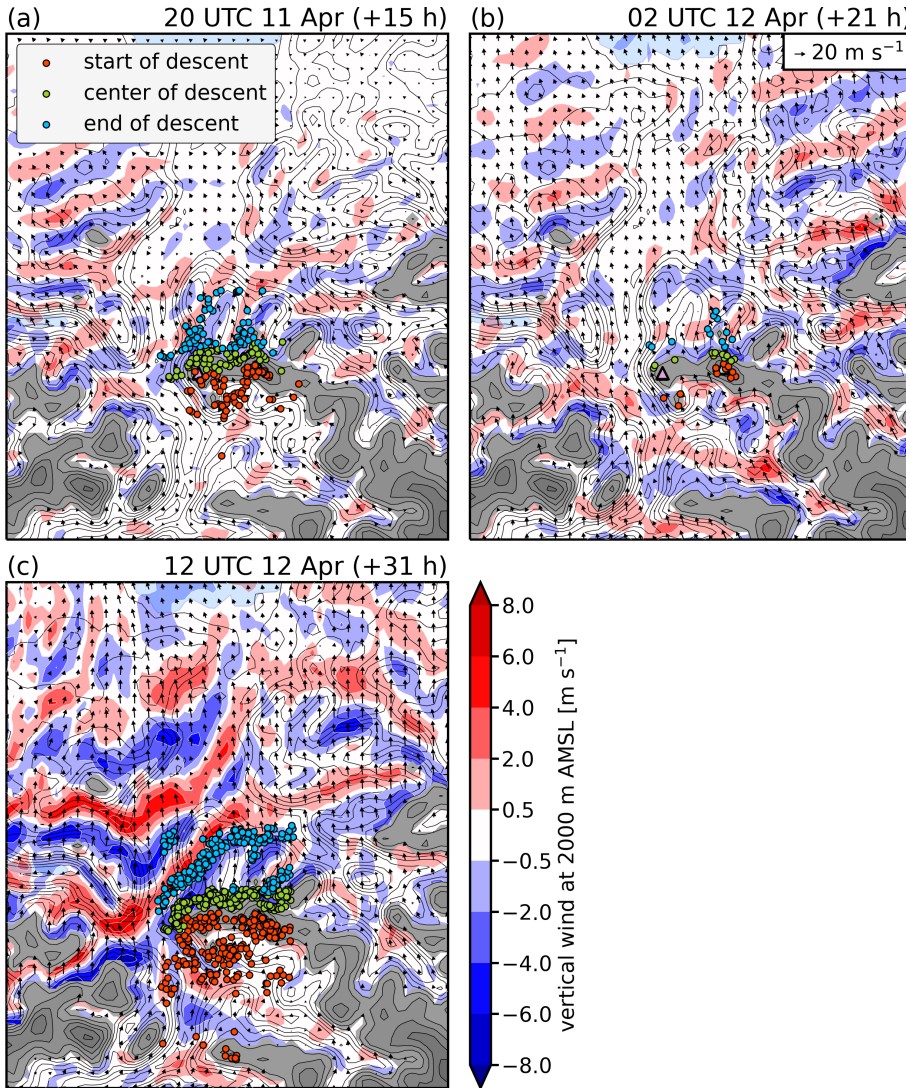

**Figure 12.** Same as Fig. 9 but for the Apr 2018 event and: (a) 20 UTC 11 April 2018; (b) 02 UTC 12 April 2018; (c) 12 UTC 12 April 2018.

The temporal evolution of descent activity during the Apr 2018 event differs from that of the Feb 2017 event (cf. Figs. 8b and 11b). During the event, several periods of enhanced descent activity interchange with periods of low activity. In general, the strongest descent is diagnosed in the time span of 25–35 h since event start. In contrast to the Feb 2017 event, the Apr 2018 event does not show a clear co-variability of descent activity and impinging wind speeds. Thus, other local factors seem to play a crucial role in this case. To investigate this in more detail, three time instants (see olive arrows in Fig. 11b) with different descent activities are selected to investigate the influence of local atmospheric conditions (15 h: the first, albeit weak, maximum of descent activity; 21 h: weak descent activity despite strong wind speeds above the crest; 31 h: peak of descent activity).



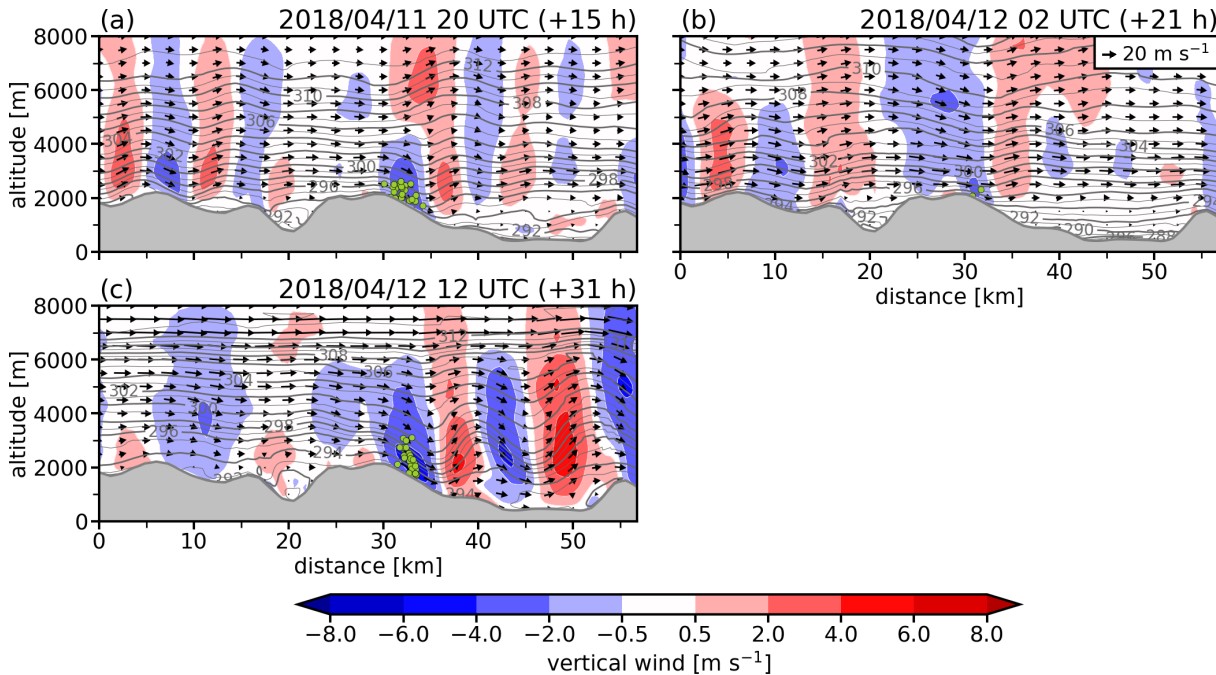

**Figure 13.** Same as Fig. 10 but for the cross section c3 (cf. Fig. 7b) during the three selected time instants: (a) 20 UTC 11 April 2018; (b) 02 UTC 12 April 2018; (c) 12 UTC 12 April 2018.

The first highlighted time of the event (15 h) also corresponds to a first maximum in descent activity. Above the peaks of
the Rätikon, moderate southeasterly winds prevail (Fig. 11a). The Falknis and the Schesaplana again excite two gravity waves,
which cause descending motion downwind of both peaks (Fig. 12a). On the one hand, air parcels descend along the northwest
slope of the Falknis into the wave trough of a horizontally propagating lee wave (Fig. 13a). On the other hand, air parcels also
descend within a gravity wave to the northeast of the Schesaplana into the Gamperdona Valley (see Fig. 7c for orientation).
Due to its position relative to the Schesaplana summit, the latter valley seems to be reached more easily by foehn air parcels
when southeasterly rather than southwesterly winds prevail.

Six hours later, 21 h after the start of the event, the registered descent activity was low (Fig. 11b). However, this seems
counterintuitive given the ambient conditions at that time. In fact, the horizontal winds in the lower troposphere even reached
their maximum during this period (Fig. 11a). Furthermore, considering the horizontal map, propagating waves of similar
amplitude as six hours earlier are present in the vicinity of the hotspot region (Fig. 12b). Anyhow, focusing on the hotspot
region itself, the wave signature (downslope winds) in the lee of Falknis is less pronounced at 2 km and essentially absent
at lower altitudes (not shown). The vertical cross section c3 (Fig. 7b) shows that, although a weak mountain wave is present
adjacent to the Falknis peak, the downward vertical velocities do not extend below 1.8 km (Fig. 13b). This may be due to the
stable stratification in the Rhine Valley during nighttime (see contracted isentropes), which prevents a further penetration of





the wave signal into the valley atmosphere. It is known that cold-air pools formed by nocturnal cooling effectively absorb the
wave energy of trapped lee waves and thus strongly dampen their amplitude (Jiang et al., 2006).

Finally, the time associated with the strongest descent activity is examined (31 h). The impinging flow above the peak
level now comes from sector south. The amplitude of the horizontally propagating waves is substantially larger than before,
especially over the valley axis of the Rhine Valley (Fig. 12c). Downward motion is observed in the lee of the entire Rätikon.
Thus, the descending air parcels are linearly aligned along the northern slopes of the mountain chain. Similar to Feb 2017, air
parcels descend along the northwestern slope of Falknis into the Rhine Valley, as well as into all northern tributaries of the
Rätikon. The vertical cross section reveals the presence of a horizontally propagating lee wave of larger amplitude than before
(Fig. 13c). Note that the descending air parcels adjacent to the slope are potentially able to reach the neutrally stratified valley
atmosphere below 1.7 km. These air parcels can thus in principle all be transported to the surface by vertical turbulent mixing
within the boundary layer. This mechanism could also explain the cluster of air parcels arriving near Vaduz (Fig. 12c).

The Apr 2018 event is, in contrast to Feb 2017, characterized by an intermittent descent activity that cannot merely be
explained by variations in the impinging flow. Rather, the investigation indicates a pronounced influence of the daily cycle.
While the stable stratification at night inhibits air parcels to subside into the valley atmosphere, a well-mixed boundary layer
during the day promotes the penetration of foehn air parcels towards the valley floor. The important role of diurnal heating
in increasing the potential temperature of the valley air and thereby facilitating the descent of foehn air parcels has been
previously documented (Mayr and Armi, 2010). It is considered the primary explanation for the pronounced daily cycle of the
climatological foehn frequency at many Alpine stations (Mayr et al., 2007; Gutermann et al., 2012).

## 5 Discussion

### 5.1 Comparison to the MAP literature

There exist both analogies and discrepancies of our study to the existing MAP literature, which are briefly discussed in the
following. For instance, Zängl et al. (2004a) diagnosed descending motion during a foehn event in the Rhine Valley by consid-
ering surface potential temperature maps. Similarly, variations in surface potential temperature in the Wipp and Ziller valleys
were attributed to the different source altitudes of air (Zängl et al., 2004b) subsiding into these valleys. Gohm et al. (2004)
identified several mountain ridges with leeside descending motion using surface potential temperature and the vertical wind
field. Among these regions was the ridge encompassing the Patscherkofel, which aligns with one of the hotspots identified in
our Lagrangian descent analysis (cf. Fig. S2d). Focusing on the Rhine Valley, Zängl et al. (2004a) identified a distinct max-
imum in surface potential temperature north of Vaduz and attributed it to the descent along the Rätikon. This finding agrees
well with our results, where the northern slopes of the Rätikon emerge as a major hotspot for descent.

The MAP publications focusing on IOP 10 revealed that downward sloping isentropes, and hence descending motion, occur
within vertically propagating gravity waves (Zängl et al., 2004a). In the region of the Wipp Valley, an overturning of the
isentropes, commonly associated with wave breaking, has been diagnosed (Gohm et al., 2004; Zängl et al., 2004b). However,
our second case study (Apr 2018) suggests the presence of propagating lee waves at the time of strongest descent activity. Still,





such a pattern of trapped lee waves has previously been reported to occur during south foehn (Zängl and Hornsteiner, 2007). Overall, the specific wave regime that most clearly favors strong descent of foehn air parcels remains unclear. Another aspect beyond the scope of this study pertains to the role of below-cloud evaporation of precipitation in stabilizing the atmosphere
and thus dampening the amplitude of gravity waves, as described by Zängl (2006). Nonetheless, vertical cross sections of the hydrometeors during the Apr 2018 event indicate no strong influence of evaporating hydrometeors on the stability of the valley atmosphere, at least at the times studied (not shown). In the local descent hotspots south of the Alpine crest, the above-mentioned effect is likely not negligible, since these air parcels experience evaporative cooling during their descent.

An interesting side note concerns the preferential descent of foehn air parcels into the Brandner Valley when large-scale
southwesterly to southerly flow prevails. In fact, Steinacker et al. (2003) already diagnosed a gravity wave along the leeside slopes of the Rätikon based on a model simulation. They attributed the frequent restriction of the foehn to the upper part of the Brandner Valley to the wave-induced perturbation in the surface pressure. However, an investigation of this local interaction between the downvalley propagation of the foehn and the gravity field aloft was beyond the scope of this paper and would require a higher-resolution simulation.

## 5.2 What other factors influence the descent?

The presented analysis proposes a strong influence of the diurnal cycle for the descent of foehn air during the Apr 2018 case study. Potentially, the nocturnal minimum in descent activity is caused by the stable stratification of the valley atmosphere, which is known to strongly attenuate the amplitude of gravity waves (Jiang et al., 2006). In alignment, previous studies stress the importance of daytime heating for foehn breakthrough (Mayr and Armi, 2010) and the climatological foehn frequency at
most surface stations in the Alps features a pronounced maximum during the afternoon hours (Mayr et al., 2007; Gutermann et al., 2012).

Several flow features that have been assigned an important role for the descent of foehn air by previous studies have not been observed or investigated in our analysis. Zängl et al. (2004a) suggested the region north of Vaduz as preferential for descending motion due to the upstream flow splitting at the junction of the Rhine Valley and the Seez Valley. The flow splitting
promotes subsiding motion for reasons of continuity: As part of the foehn flow is diverted into the Seez Valley, the reduced mass flux into the Rhine Valley is balanced by the descent of air parcels from higher levels, for instance along the slopes of the Rätikon. However, results from our simulations suggest that descent along the Rätikon can occur independently of the upwind flow splitting (not shown). To get a more reliable statement with respect to the role of the low-level flow splitting for foehn air descent in the Rhine Valley, additional analyses including all simulated events would be beneficial.

Besides, a rotor circulation within the Inn Valley has been attributed an important role for transporting air parcels into the valley atmosphere during a northwest foehn event (Saigger and Gohm, 2022). A rotor-like circulation did not emerge in our case studies, although a slight cross-valley component has been observed in the region of the concavity in the lee of the Falknis. Whether such rotor circulations play a role for the descent of foehn air in other regions remains for future research.



## 6    Conclusions

The rapid downwind descent of foehn air into leeside valleys has long been considered one of the greatest conundrums in research on the Alpine foehn. This study provides the first comprehensive assessment of the descent using a set of 15 COSMO-1 hindcasts of Alpine south foehn (Sect. 2.1). We invoke a Lagrangian perspective using online trajectories calculated along with the model simulations (Sect. 2.2). An algorithm allows us to identify strong and rapid downward motion along their pathway (Sect. 2.3). The paper identifies favorable descent hotspots along the entire Alpine arc, describes these descent hotspots in

terms of kinematic and thermodynamic characteristics, and examines the conditions that invigorate the descent in a particular hotspot region in the Rhine Valley. The main results are summarized here:

- Foehn descent occurs in distinct hotspots, often restricted to the immediate lee of mountain peaks and chains. The local terrain thus naturally constrains the regions of descending air parcels during foehn. Many of the hotspots are situated near the actual arrival locations of the foehn air parcels, rather than in the vicinity of the Alpine crest. The well-known

550       foehn valleys emerge as local descent maxima, and the overall most intense hotspot is located along the Rätikon range.

- The magnitude of the descent is governed by the elevation difference of the underlying terrain. Consequently, it reaches maximum values (exceeding 1500 m) in regions where the local peaks rank among the highest of the Alps and decline steeply on their downstream side, such as the Mont Blanc Massif and the Bernese Alps. The time span of the descent, i.e. the time needed to cover a given descent segment, usually ranges between 4 to 10 min. However, locally the descent can

555       occur even faster. The majority of air parcels descends approximately adiabatically and exhibits little change in specific humidity except for air parcels south of the Alpine crest, where evaporative cooling increases the specific humidity along the pathways of descending air parcels.

- The strong descent within the Rhine Valley hotspot is attributed to gravity waves excited by the peaks of the Rätikon, in particular the Falknis and the Schesaplana. On a local scale, the northwestern slope of the Falknis forms a topographic

560       concavity that redirects the downslope flow towards the Rhine Valley floor near Vaduz (see also schematic illustration in Fig. 14). This westward deflection of the low-level flow consistently occurs during both events, independent of the impinging wind direction, but is more pronounced during the Feb 2017 event (cf. Fig. S9). Besides, the Schesaplana excites a gravity wave that primarily guides subsiding air parcels into either the Brandner Valley or the Gamperdona Valley, depending on whether southwesterly or southeasterly winds impinge on the Rätikon (Fig. 14). Overall, the emerging

565       gravity wave patterns vary for the different highlighted times, suggesting that no preferential wave regime is most conducive for descent along the Rätikon. During Feb 2017, the gravity wave amplitudes generally increase in accordance with stronger lower- to mid-tropospheric winds within a stable layer above crest level. In contrast, the temporal evolution of the descent activity during the Apr 2018 event demonstrates that nocturnal cooling and the resulting formation of a cold-air pool can impede strong descent of foehn air by effectively attenuating the mountain waves.

Since the online trajectories explicitly resolve the rapid and small-scale downward motion in the lee of orography, our dataset enables us to follow the motion of air parcels as they cross the Alpine barrier. In contrast to previous qualitative and




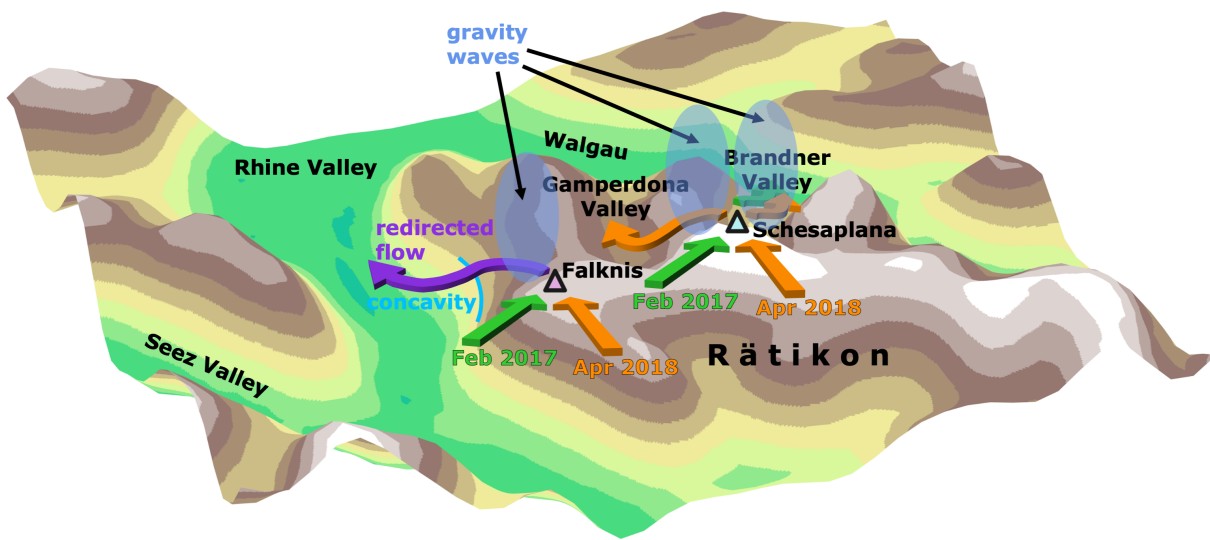

**Figure 14.** Schematic illustration showing some of the key findings obtained in Sect. 4 when investigating the descent hotspot along the Rätikon.

Eulerian approaches, the trajectory dataset therefore provides the novel opportunity to unequivocally identify descent hotspots and allows us to assess the foehn descent quantitatively and systematically using the Lagrangian perspective. Notably, our analysis encompasses a much broader region compared to existing studies, revealing that hotspots of foehn descent are not 575 limited to the Rhine Valley and the Wipp Valley but exist across all foehn regions north of the Alps.

Our study also comes along with some limitations, which are briefly specified here. First, the Lagrangian identification of descent does not require that the descending motion is of a persistent nature; rather, wave-like undulations of air parcels are classified as descent with the chosen approach. It is, however, debatable whether this really constitutes a methodological limitation, or rather reflects a typical characteristic of foehn flows, which are tightly linked to wave motions and thus often 580 momentarily reach the surface at a certain location, only to be lifted off-ground further downstream. In addition, the fraction of the foehn flow through gaps in the orographic barrier (e.g., Gotthard, Brenner) might not be adequately captured by the method, as these air parcels descend too slowly or do not cover enough elevation difference and thus fall below our detection thresholds.

Since all of our foehn events were selected based upon foehn occurrence at Altdorf in the Central Alps, the results, and 585 in particular the number of descending air parcels, might be biased towards this region. However, Alpine south foehn events are typically associated with a distinct large-scale synoptic situation and foehn is thus likely to co-occur at many stations. Accordingly, station observations confirm that foehn occurred at multiple stations across Switzerland during all of the simulated events (not shown). In particular, the measurements reveal that foehn also prevailed in the Rhine Valley during both events that were studied in greater detail in Sect. 4 (Feb 2017, Apr 2018).



Finally, these detailed case studies exclusively focused on the hotspot in the Rhine Valley. More insight into the mechanisms governing the descent could be obtained by considering additional case studies in a number of hotspot regions. Future research should, therefore, extend the analysis to other regions that are also associated with strong descent, but have been much less in the focus of research to this point.

Even though we present the first systematic analysis of foehn descent employing a Lagrangian methodology, our study raises

several questions to be addressed by prospective research. Firstly, it remains unclear what kind of gravity wave pattern (vertically propagating waves, breaking waves, trapped lee waves) promotes the strongest descent. While different wave regimes could be detected within the hotspot region along the Rätikon, it is unknown whether these findings carry over to other descent hotspots and foehn events. One challenge when investigating the role of gravity waves in the descent lies in their characterization based on 3D Eulerian fields. Here, we adopted a qualitative approach to describe the wave patterns at different time

instants. In the future, a more quantitative approach (e.g., following Kruse and Smith, 2015) could elucidate the gravity wave characteristics during foehn more systematically. In addition, although our study highlights the importance of gravity waves for the descent of foehn air, we did not investigate the role of other mechanisms. Rossmann (1950) suggested that evaporative cooling is an important driver of the descending motion. The online trajectories could serve as an appropriate dataset to partition the descent into a buoyancy-driven fraction ("thermodynamic mechanism") and a wave-driven fraction ("dynamic

mechanism"), provided that these two mechanisms can be clearly separated using a Lagrangian momentum budget. Moreover, the case study of Apr 2018 illustrates that nocturnal cold-air pool formation can effectively dampen the wave amplitude and thus inhibit strong descent. Still, further in-depth analysis is required to corroborate the significance of this effect on the descent. Besides the diurnal cycle, other factors might influence the descent of foehn air parcels as well. In this regard, the upstream flow splitting at the junction of the Rhine Valley and the Seez Valley have been mentioned by a previous study (Zängl

et al., 2004a). While we did not find a clear correlation of the flow splitting and the descent activity along the Rätikon, future studies are necessary to clarify the effect of the flow splitting on the descent.

In conclusion, the present paper provides new insights on one of the long-standing conundrums in foehn research. Using online trajectories, we have examined the descent of foehn air parcels from an unprecedented Lagrangian perspective. This methodology enabled us to investigate the phenomenon quantitatively and along the entire Alpine arc. Nevertheless, numerous

open questions still remain unanswered, for instance regarding the role of different wave regimes or the transferability of our findings to other hotspot regions, emphasizing the need for further research. Our novel trajectory dataset offers an opportunity to tackle these questions.

*Code and data availability.* Operational COSMO-1 analyses are available for research purposes upon request to MeteoSwiss. Processed data from the simulations are available from the authors upon request. The code used for the analysis and visualization is written in Python 3.9

and is available from the authors upon request.





## Appendix A: Details on the COSMO model setup

### A1 Specifics of the model setup

Using a climatology of different foehn types in the five-year period from November 2015 to November 2020 (Jansing et al., 2022), we selected 13 events that represent all seasons and different foehn types (Table A1) and calculated hindcasts for each
of these. The 13 events are complemented by two additional simulations that were performed at an earlier stage (the Nov 2016 and Feb 2017 events) with the goal of a more detailed analysis (see also Jansing and Sprenger, 2022, for the Nov 2016 event). Therefore, the simulated periods for these two events have been extended compared to the actual foehn period at Altdorf (see Table A1). Furthermore, also the Mar 2018 case study ran for a longer period in order to extend the simulated period of this particular shallow-foehn event (see also Jansing, 2023), that lasted only shortly in Altdorf, but longer at other foehn stations
closer to the Alpine crest.

For Nov 2016 as well as for Feb 2017, output is written to disk at higher temporal resolution (10 min). While the Nov 2016 and Feb 2017 events have been simulated with COSMO version 5.6 on central processing units (CPUs), the other case studies made use of the GPU-capability of both the COSMO model and the online trajectory module. Porting the existing online trajectory module to GPUs (see also Bukenberger et al., 2023) substantially enhanced the performance of the new model setup
and allowed us to conduct an extended number of simulations with COSMO version 5.9. Aside from these differences in the model version, the simulations were performed with virtually identical setups, except for minor changes of a tuning parameter and the soil model, to align the GPU runs with the latest operational setup of MeteoSwiss.

### A2 Specifics of the online trajectory setup

Here, we provide additional details on the exact setup of the online trajectory module. For all simulations, the spacing between
starting points has been set to 0.175° and 250 m in the horizontal and vertical, respectively. The lowest trajectories were released 20 m AGL, reaching up to a maximum of 5 km AMSL. In a tradeoff between the largest possible distance to the Alpine arc, while nevertheless taking into account all potential source regions for foehn air parcels, the trajectory starting points were cropped by a polygon with 50 km distance (violet contours in Figs. 1b,c) to the innermost closed 1500 m contour (blue contours in Figs. 1b,c) around the Alps (same as in Jansing and Sprenger, 2022).
Air parcels were released in two-hourly time intervals for the Nov 2016 and Feb 2017 case studies and in hourly time intervals for all other case studies (Table A2). The trajectory calculations started upon initialisation of the model and ended 8 h prior to the end of the simulated period. For the Feb 2017 case study, trajectories have only been released until 18 h prior to the end of the simulated period, as the foehn period was considerably shorter compared to the simulated period for this event. Each trajectory was calculated until reaching a maximum length of 36 h. All standard prognostic variables were traced along
the trajectories. For the Nov 2016 and Feb 2017 events, output was written at the highest possible temporal resolution (10 s). For all other case studies, the trajectory output was stored in 2 min steps. Therefore, to homogenize the trajectory dataset, the temporal resolution of the trajectory data from Nov 2016 and Feb 2017 were coarse-grained to 2 min for this study.





**Table A1.** Overview of the 15 foehn events that have been simulated with the COSMO model. The foehn types are determined following Jansing et al. (2022).

| Case study | Simulated period (Duration) | Foehn period at Altdorf | Output frequency | Model version | Predominant foehn type |
|---|---|---|---|---|---|
| Nov 2016 | 18 UTC 19 Nov – 18 UTC 25 Nov (144 h) | 05 UTC 20 Nov – 17 UTC 24 Nov | 10 min | v5.6 (CPU) | moist foehn |
| Feb 2017 | 12 UTC 26 Feb – 00 UTC 1 Mar (60 h) | 13 UTC 27 Feb – 04 UTC 28 Feb | 10 min | v5.6 (CPU) | moist foehn |
| Mar 2016 | 12 UTC 04 Mar – 08 UTC 05 Mar (20 h) | 18 UTC 04 Mar – 02 UTC 05 Mar | 30 min | v5.9 (GPU) | moist foehn |
| May 2016 | 06 UTC 05 May – 22 UTC 11 May (160 h) | 12 UTC 05 May – 16 UTC 11 May | 30 min | v5.9 (GPU) | moist foehn |
| Oct 2016 | 05 UTC 13 Oct – 00 UTC 15 Oct (43 h) | 11 UTC 13 Oct – 18 UTC 14 Oct | 30 min | v5.9 (GPU) | moist foehn |
| Jan 2017 | 18 UTC 26 Jan – 12 UTC 28 Jan (42 h) | 00 UTC 27 Jan – 06 UTC 28 Jan | 30 min | v5.9 (GPU) | dry foehn |
| Jul 2017 | 11 UTC 31 Jul – 22 UTC 01 Aug (35 h) | 17 UTC 31 Jul – 16 UTC 01 Aug | 30 min | v5.9 (GPU) | dry foehn |
| Mar 2018 | 00 UTC 24 Mar – 07 UTC 25 Mar (31 h) | 20 UTC 24 Mar – 01 UTC 25 Mar | 30 min | v5.9 (GPU) | shallow foehn |
| Apr 2018 | 05 UTC 11 Apr – 01 UTC 13 Apr (44 h) | 11 UTC 11 Apr – 19 UTC 12 Apr | 30 min | v5.9 (GPU) | moist foehn |
| May 2018 | 09 UTC 12 May – 03 UTC 13 May (18 h) | 15 UTC 12 May – 21 UTC 12 May | 30 min | v5.9 (GPU) | shallow foehn |
| Apr 2019 | 05 UTC 18 Apr – 03 UTC 20 Apr (46 h) | 11 UTC 18 Apr – 21 UTC 19 Apr | 30 min | v5.9 (GPU) | shallow foehn |
| Nov 2019 (1) | 07 UTC 04 Nov – 22 UTC 04 Nov (15 h) | 13 UTC 04 Nov – 16 UTC 04 Nov | 30 min | v5.9 (GPU) | gegenstrom foehn |
| Nov 2019 (2) | 10 UTC 22 Nov – 00 UTC 24 Nov (38 h) | 16 UTC 22 Nov – 18 UTC 23 Nov | 30 min | v5.9 (GPU) | moist foehn |
| Feb 2020 | 11 UTC 09 Feb – 07 UTC 10 Feb (20 h) | 17 UTC 09 Feb – 01 UTC 10 Feb | 30 min | v5.9 (GPU) | gegenstrom foehn |
| Oct 2020 | 03 UTC 02 Oct – 08 UTC 03 Oct (29 h) | 09 UTC 02 Oct – 02 UTC 03 Oct | 30 min | v5.9 (GPU) | dimmer foehn |

Note that the applied selection procedure to obtain foehn trajectories (cf. Sect. 2.2) results in a substantially varying total number of selected trajectories between the cases studied (Fig. S1), ranging from more than $10^5$ to less than $10^2$ trajectories





**Table A2.** Overview of the online trajectory setups for the 15 foehn events.

| Simulations | Starting box extents and number of trajectories ($n$) | Release frequency | Output frequency |
|---|---|---|---|
| CPU simulations (Nov 2016, Feb 2017) | 6–13.35°E, 43.175–45.8°N  $n = 9\,233$ | two-hourly | 10 s |
| gegenstrom-foehn events (Nov 2019 (1), Feb 2020) | 3.025–5.825°E, 43.175–47.55°N  6–10.2°E, 43.175–45.8°N  $n = 11\,644$ | hourly | 2 min |
| All other GPU simulations | 3.025–5.825°E, 43.175–45.8°N  6–13.35°E, 43.175–45.8°N  $n = 12\,906$ | hourly | 2 min |

(for the Mar 2018 case). This can have several different reasons. First of all, longer simulation periods, as for example for Nov 2016 or May 2016, will automatically result in a larger number of foehn trajectories, as more trajectories are released in total. Secondly, all events with a total of less than $10^3$ trajectories belong to the shallow-foehn type (Mar 2018, May 2018) or the gegenstrom-foehn type (Nov 2019 (1), Feb 2020; see also in Jansing, 2023). For these foehn types, defining a feasible starting setup is challenging. The weak large-scale flow during shallow foehn would require to start trajectories closer to the 660 Alpine crest, so that more of them are able to reach the northern foehn regions. The pronounced zonal flow above crest level during gegenstrom foehn, in turn, requires starting points to be positioned more to the west rather than to the south. Although the starting setup has been adjusted for the gegenstrom-foehn events (Fig. 1c), it is apparently still challenging to capture the pathways of foehn air parcels during these events. The selection procedure requires a trajectory to intersect with the Alpine crest line, which might poses a too rigid criterion for many trajectories during gegenstrom foehn, as they approach the foehn regions 665 from the west (with some exceptions). Finally, the foehn flow in the model might also be too weak during the simulations, preventing a larger number of trajectories to descend into the northern foehn regions. Despite this wide range in the number of trajectories per case, we decided to retain all events for the analyses, as the trajectory dataset is anyways collectively analyzed in large parts of the paper. Note that in particular the presented case studies (Feb 2017, Apr 2019; see Sect. 4) feature a reasonably large number of foehn trajectories to be investigated with further scrutiny (Fig. S1).

*Author contributions.* All co-authors equally contributed to the design of the study and the interpretation of the results. LJ conducted the numerical simulations, processed the data, performed the analysis and prepared the figures in the context of his PhD project. LJ drafted the paper, supported by MS and LP.



*Competing interests.* LP is a member of the editorial board of Weather and Climate Dynamics. Otherwise, the authors have no competing interests to declare.

*Acknowledgements.* This research has been supported by the Schweizerischer Nationalfonds zur Förderung der Wissenschaftlichen Forschung (grant no. 181992). We acknowledge the Swiss National Supercomputing Centre for providing access to the computational resources. Two of the simulations (Nov 2016, Feb 2017) were conducted within the development project d111/d111m, the remaining simulations were performed using resources of the project s1063. In this regard, we want to express our thanks to Sebastian Schemm for providing access to the resources of s1063. Furthermore, we acknowledge the Federal Office of Meteorology and Climatology MeteoSwiss for giving us access to
the COSMO-1 analysis data, which were used as initial and boundary conditions for the simulations. Additionally, we are grateful for the technical help of Annette Miltenberger and Annika Oertel in setting up the initial version of the online trajectory module. Stefan Rüdisühli, Katie Osterried and Sebastian Schemm are acknowledged for guiding the GPU porting of the online trajectory module. Besides, we express our gratitude to Heini Wernli for the valuable input.



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
