# Peer review of "A Lagrangian framework for detecting and characterizing the descent of foehn from Alpine to local scales"

_EGUsphere, 2023_

## Referee Comment (RC1)

**Review: On the descent of Alpine south foehn (Jansing, Papritz and Sprenger)**

**Summary**

The paper explores why foehn air descends in the lee of topographic obstacles through numerical simulations at the kilometer-scale of 15 south foehn events in the Swiss Alps. The paper is well structured, the writing is clear and the figures are well chosen. However, three fundamental issues exist. (1) The main conclusion that gravity waves trigger foehn descent rests upon a false premise, (2) an alternative hydraulic mechanism is mentioned but not explored, and (3) the numerical simulations are likely incapable of properly reproducing flow separation, which is crucial for how the flow descends. For these three reasons, the recommendation is to reject the paper. However, the paper could be a fruitful addition to the foehn literature once these three issues have been properly addressed.
The following explains the reasons for the rejection in more detail.

**Reasons for rejection**

**Descent of air shapes gravity waves, not the other way around:** The descent of the air behind topography makes it possible for large gravity waves to form, rather than gravity waves causing the descent as the paper argues. When air impinges on an obstacle in stably stratified air, a gravity wave will be launched. Its shape and amplitude, however, will not only depend on the non-dimensional height of the obstacle (the product of relative height above incoming isentrope, stability and the inverse of speed), which is a function of the *upstream flow only*, but also on the shape of the obstacle *including its lee side*.

Downstream descent is limited by real or virtual topography. Virtual topography is formed by the level of neutral buoyancy of the overflowing air with respect to the pre-existing air on the downwind side. Descent is only possible when the level of neutral buoyancy of the overflowing air in the downstream air is lower than the upstream altitude of the air, i.e. when the potential temperature of the upstream air is lower than the pre-existing downstream air at the same altitude. Armi & Mayr (2015) showed this in detail with observations from instrumented aircraft (in-situ and cloud radar), dropsondes, radiosondes and satellite data.

The present paper even acknowledges the fact that gravity waves depend on how far air can descend (and not vice versa) in lines 568-569 of the conclusions by stating that "nocturnal cooling and the resulting formation of a cold-air pool impede strong descent of foehn air by effectively attenuating the mountain waves" without concluding that the descent cannot go further than to where this isentrope is located on the downstream side.
The third row of Fig. 10  (09 UTC 28 Feb) shows minimal gravity wave activity because the underlying virtual topography produced by nocturnal cooling is so smooth. The previous and the subsequent afternoons (13 UTC) of rows 1 and 4, respectively, have warmer leeward temperatures caused by daytime warming mainly by sensible heat fluxes (and initially by the turbulent erosion of the cold pool). Consequently gravity waves are larger.

**Alternative descent mechanism ignored:** In the introduction the paper describes two main mechanisms that have been proposed in the literature to explain the descent - gravity waves and hydraulics - differential density (colder air upstream descending to its level of neutral buoyancy downstream). The claim in lines 75 - 77 that the hydraulic approach is applicable to shallow but not deep foehn is not correct (e.g. Armi & Mayr, 2007; Armi & Mayr, 2015; Winters & Armi, 2014).

The present paper only explores the gravity wave mechanism and ignores the hydraulic one despite the rich material available from the numerical simulations. The discussion subsection 5.2 ("What other factors influence the descent?") does not even mention the second mechanism anymore! This is a pity because the study has ample data to test both mechanisms whether they are the reasons for the descent and thus provide more material to further the discussion in the foehn community. The testable characteristic for the hydraulic response is that substantial descent starts when the overflowing air becomes colder (in terms of potential temperature) *relative* to the downstream air at altitudes below which it descends. Similarly, there need to be testable characteristics for the gravity wave mechanism that avoid, for example, the speculative attribution of the "intermittent periods of weak vertical motion" (lines 450-451) in the April 2018 case study to the "variability of wave-induced subsidence". With the alternative explanation this could be directly tested by examining whether the upstream air had become colder relative to the downstream one.
However, the most important aspect to test for the adequacy of the mechanisms is the evolution of the onset of subsidence in the simulations.

**Questionable whether numerical simulations correctly reproduce processes that lead to downstream descent:** COSMO, the numerical model used for the simulations, has terrain-following coordinates near the surface. The simulation setup also uses horizontal diffusion (lines 179-181). If the numerical diffusion acts along model surfaces, which are slanted in complex terrain, artificial vertical, cross-isentropic mixing will ensue since in general the terrain-following coordinates cross isentropes in a stably stratified atmosphere. As a result, air will not separate from steep downward sloping topography as in reality, but rather descend. There are no figures in the paper to substantiate this claim. However, this inference is supported by the fact that the strongest descent in the model *simulations* occurs along the steepest slopes behind the tallest mountains (lines 313-315 and Fig. 4b) whereas in *reality* steeper slopes are likelier to lead to flow separation. Pressure perturbations from a gravity wave alone will not be strong enough to force the flow down by 1500 m in stable stratification.
Observations in the ocean (Knight Inlet; Farmer & Armi, 2001, and references therein) and in the atmosphere (Owens Valley during the T-REX campaign; Mayr & Armi, 2010 see Figs. 5 and 6; Armi & Mayr, 2011) confirm that flow separates along steep slopes before it later descends when small-scale mixing forms a wedge of nearly stagnant and neutrally stratified air (away from the terrain) that separates the descending flow from the flow aloft.
The detailed observations of the oceanic "foehn" flow into Knight inlet demonstrated the importance of correctly simulating the boundary layer separation. Numerical simulations with an atmospheric model (Afanasyev & Peltier, 2001) and an oceanic model (Cummins, 2000) erroneously produced a large overturning wave soon after the flow started flowing over the sill (the oceanic equivalent of a mountain crest) contrary to what the observations showed. Only when Cummins (2000) modified the topography to force boundary layer separation did the oceanic model correctly simulate the evolution of the flow. The authors will need to show that the model adequately handles flow separation; currently there are no figures in the paper that would allow one to do that. This can be done by first examining the time it takes for the descent along the slopes to become established. If this time is not (much) larger than $2\pi$ /

N then the simulation will be incorrect. Second, the evolution of the descent must be visually inspected for congruence with observations of the initial flow separation and the way the wedge of the nearly stagnant and neutrally stratified air is formed that isolates the descending air from the flow above.

**References:**

Afanasyev, Y. D. & Peltier, W. R., (2001): On breaking internal waves over the sill in Knight Inlet. *Proceedings of the Royal Society London. A.,* **457**, 2799–2825. https://doi.org/10.1098/rspa.2000.0735.

Armi, L. & Mayr, G.J. (2007). Continuously stratified flows across an Alpine crest with a pass: Shallow and deep föhn. *Quarterly Journal of the Royal Meteorological Society.* **133**, 459 - 477. https://doi.org/10.1002/qj.22.

Armi, L. & Mayr, G.J. (2011): The Descending Stratified Flow and Internal Hydraulic Jump in the Lee of the Sierras. *Journal of Applied Meteorology and Climatology*, **50**, 1995–2011, https://doi.org/10.1175/JAMC-D-10-05005.1.

Armi, L. & Mayr, G. J. (2015): Virtual and Real Topography for Flows across Mountain Ranges. *Journal of Applied Meteorology and Climatology*, **54**, 723–731, https://doi.org/10.1175/JAMC-D-14-0231.1.

Cummins, P. F. (2000): Stratified flow over topography: time-dependent comparisons between model solutions and observations. *Dynamics of atmospheres and oceans*, **33**, 43-72. https://doi.org/10.1016/S0377-0265(00)00044-0

Farmer, D.M. & Armi, L. (2001): Stratified flow over topography: models versus observations *Proceedings of the Royal Society London. A.,* **457**, 2827–2830, https://doi.org/10.1098/rspa.2001.0802

Winters, K., & Armi, L. (2014): Topographic control of stratified flows: Upstream jets, blocking and isolating layers. *Journal of Fluid Mechanics, 753*, 80-103. https://doi.org/10.1017/jfm.2014.363

---

## Referee Comment (RC2)

**Review of "On the descent of Alpine south foehn" by Jansing, Papritz and Sprenger**

The authors present a climatological study of where along the Alpine ridge foehn descent occurs in high resolution model simulations. They present a novel Lagrangian diagnostic to identify foehn descent and thermodynamic history of foehn parcels during descent (temperature and specific humidity). Further, they discuss present a more detailed analysis of foehn descent for two case-studies centered at the Rätikon.

The paper is mostly well structured and clearly written. If the comments below are addressed, I am recommending the paper to be published.

**Major issue**

1. In the abstract the authors claim to investigate the descent process "with unprecedented detail". Indeed their study identifies foehn descent in a more spatio-temporally extensive data-set then previously, but their is no detailed analysis of the physical processes resulting in the downward motion of the air parcels. Discussion of the physical processes is limited to inference from a few cross-sections for two case-studies. Given the simulation data that they have, it would be very interesting to try and quantify the causes of downward acceleration of air parcels (buoyancy, vertical pressure gradient, …). They allude to this possibility in the conclusion, which is fine and I would encourage to highlight this even more. Indeed the paper would benefit from a more detailed physical analysis, but at least the abstract needs to be modified to accurately represent the contents of the paper.
2. The section of the introduction starting at p. 4, l.121 is not very well structured and open research questions could be stated more explicitly. Please consider rewriting.
3. Potential model issues in the representation of descent, e.g. potential issues of the turbulence scheme over complex terrain, need to be discussed in a more structured and prominent location (e.g. section in the conclusion / discussion). Hints at potential modeling problems are found throughout the manuscript, but it is failed to present them in structured manner and a discussion of their potential impacts on the results is missing.

**Minor issues**

1. p. 2, l. 42: Why would foehn flows ignite forest fires? I would rather expect they are produce atmospheric conditions, that are more conducive to igniting fires.
2. p. 2, l. 60: "foehn wall might inherit a key role for the downward acceleration": I do not understand this sentence: What is inherited and by what?
3. Section 2.1: in addition to the height of the lowest model level, it would be interesting to state the average vertical grid spacing in the valleys, e.g. the lowest 2 km.
4. Fig. 3: Would be interesting to see the distribution of foehn trajectories passing the locations of descent. I.e. is the distribution mirroring more frequent foehn events / large mass flux, e.g. along the Rhine valley and what is the percentage of foehn air parcels that descent in the specific regions.
5. p. 11, l. 274: „gravity waves […] force descending motion of air parcels": The Lagrangian diagnostic are just another perspective of the Eulerian velocity fields and vice-versa. So it cannot be claimed from the evidence presented that gravity waves force descent of air parcels. Descending air parcels in some sentence constitute the downward motion in the Eulerian perspective. Maybe a better wording instead of „force" would be „associated". Similar statements are made e.g. on p. 13 l. 321 and in a few other places, and also need modification.
6. p. 12, l. 311: „exact relation": Given the scatter in the data, I do not agree that this is an exact relation.
7. p. 16, l. 349: „and especially the cause for its formation": I would suggest to drop this statement. The following section does not provide any evidence for why a hotspot should form in particular behind the Rätikon and not other topographic features / locations along the Alpine chain.
8. p. 17, l. 366 ff: I would suggest to first discuss the general characteristics of the foehn event before providing details on the time instances discussed afterwards to reduce repetition.
9. p. 27, l. 548: „constrains": I am not quite sure what you want to say here. Local terrain determines regions of descent / is anchoring regions of descent?

10. p. 27. l. 551ff: Given the evidence (in the paper and the more general foehn literature), it would be more accurate to state that the elevation difference is an upper limit to the foehn descent and that (at least) in the model this is often (though not always - maybe you can even quantify how often) realized.

**Technical / language issues**
1. p. 1, l. 7: „thereby" seems to be inadequate here, please modify.
2. p. 5, l. 153: „grid resolution" -> „grid spacing"?
3. p. 5, l. 158: „recent **modeling** studies"
4. p.7, l. 198: „accordingly" seems to be inadequate here, please modify.
5. p. 8, l. 207: „compiled" > „computed"
6. p. 14, l. 326: „thereby" seems to be inadequate here, please modify.
7. p. 17, l. 355: „thereby" seems to be inadequate here, please modify.
8. p. 17, l. 360: please clarify whether the referred date of foehn break-through is identified in observations or model data (and if the former what observations and with what method). Reference to an older paper is not adequate.
9.

---

## Author Comment (AC1)

**Final response to reviewers**
* * *
**On the descent of the Alpine south foehn**
**Lukas Jansing | Lukas Papritz | Michael Sprenger**
**Submitted to WCD, egusphere-2023-1536**
**October 4, 2023**
* * *
**General statement:**

We would like to acknowledge both reviewers for taking the time to reviewing our manuscript. We will consider the valuable feedback and address the criticized aspects in a revised version of the manuscript and we are therefore confident that the revision will result in a significantly improved manuscript. In this document, we respond to each of the reviewers' comments and outline how we intend to address them (original reviewer comments in black, our answers in blue). We also highlight the specific changes we plan to make to the manuscript.

**Reviewer 1:**

We appreciate the reviewer's evaluation of our manuscript and we take note of the strong criticism expressed by the reviewer. We concur that our manuscript overemphasizes the gravity wave mechanism, while wrongfully neglecting the hydraulic mechanism. We would like to stress that, with this study, we did not intend to provide a definitive answer to the underlying physical mechanisms leading to the descent. Instead, our primary objective was to introduce a novel Lagrangian framework that offers the possibility to investigate the descent using a mesoscale NWP model. We believe that this aspect was not emphasized enough and therefore also not sufficiently appreciated by the reviewer. Furthermore, we disagree with some of the reviewer's statements. Nevertheless, we are committed to addressing the concerns by making appropriate changes to the manuscript. For more details, please refer to the specific responses to all three points below.

**Descent of air shapes gravity waves, not the other way around:** The descent of the air behind topography makes it possible for large gravity waves to form, rather than gravity waves causing the descent as the paper argues. When air impinges on an obstacle in stably stratified air, a gravity wave will be launched. Its shape and amplitude, however, will not only depend on the non-dimensional height of the obstacle (the product of relative height above incoming isentrope, stability and the inverse of speed), which is a function of the *upstream flow only,* but also on the shape of the obstacle *including its lee side*.

Downstream descent is limited by real or virtual topography. Virtual topography is formed by the level of neutral buoyancy of the overflowing air with respect to the pre-existing air on the downwind side. Descent is only possible when the level of neutral buoyancy of the overflowing air in the downstream air is lower than the upstream altitude of the air, i.e. when the potential temperature of the upstream air is lower than the pre-existing downstream air at the same altitude. Armi & Mayr (2015) showed this in detail with

observations from instrumented aircraft (in-situ and cloud radar), dropsondes, radiosondes and satellite data.

The present paper even acknowledges the fact that gravity waves depend on how far air can descend (and not vice versa) in lines 568-569 of the conclusions by stating that "nocturnal cooling and the resulting formation of a cold-air pool impede strong descent of foehn air by effectively attenuating the mountain waves" without concluding that the descent cannot go further than to where this isentrope is located on the downstream side.

The third row of Fig. 10 (09 UTC 28 Feb) shows minimal gravity wave activity because the underlying virtual topography produced by nocturnal cooling is so smooth. The previous and the subsequent afternoons (13 UTC) of rows 1 and 4, respectively, have warmer leeward temperatures caused by daytime warming mainly by sensible heat fluxes (and initially by the turbulent erosion of the cold pool). Consequently gravity waves are larger.

Many thanks for this detailed discussion of the effect of stratification and virtual topography on the descent of air parcels in the lee of mountain ranges. We partly agree with this comment, especially regarding the controlling factors that affect the shape and amplitude of the gravity waves excited by mountain peaks. We also agree with the statement that air in the lee will descend until it reaches the level of neutral buoyancy (under the assumption of adiabatic, stably stratified flow). However, we also disagree with several aspects raised by the reviewer:

- Based on the reviewer's statement, the virtual topography seems to be the most important factor for the characteristics of the resulting gravity wave, which we do not agree with. We do not think that gravity wave formation, e.g., along the Rätikon, is only controlled by the stratification and virtual topography to the lee of the mountain, but also by the upstream flow conditions (upstream stratification and upstream wind speed) and the shape of the obstacle (as mentioned by the reviewer at the beginning of his statement).

- We are convinced that gravity waves play a more active role in the descent than the reviewer's comment suggests. Several studies have shown that descending flow and locally enhanced foehn winds are associated with gravity waves forming in the lee of local orographic features. Relevant for our target region are, e.g., the studies by Drobinksi et al. (2003, 2007), Gohm et al. (2004), Zängl and Hornsteiner (2007), Zängl et al. (2004a,b). A more active role was also recently attributed to gravity waves in the west foehn in the Inn Valley (Saigger and Gohm, 2022). They state that the penetration of the westerly flow into the Inn Valley is partly controlled by the formation (or absence) of a gravity wave at the western boundary of the valley. All the aforementioned studies show that an 'active' role can and (partly) must be attributed to gravity waves in controlling and/or modulating the descent and the near-surface winds during foehn events. Of course, we acknowledge that stratification and virtual topography can be important controlling factors, as explicitly shown in the studies provided by the reviewer. However, we argue against such a strict and exclusive role of these two parameters. In addition, the level of neutral buoyancy (LNB), and thus the virtual topography, can also be modulated by the downstream effects of gravity waves, so that a clear separation of the different factors may not be possible.

- In our opinion, descending motion is an intrinsic feature of orographic gravity waves. The descent of air and the associated gravity wave occur and accentuate simultaneously. The reviewer's statement, in our opinion, overemphasizes the role of the virtual topography and neglects the intrinsically coupled nature of the two phenomena. Is it the gravity wave that shapes the descent of the air, or does the

descent of the air shape the gravity wave? Since these two effects of stratified flow past orography are intrinsically coupled, there is no clear answer, but the reviewer only considers one side.

Nevertheless, the reviewer raises an important point by highlighting that our study is biased towards a gravity wave perspective. In the revised manuscript, we will therefore:
- Emphasize that the descent is an intrinsic feature of gravity waves, rather than claiming a unidirectional causal relationship between these two aspects (e.g., L. 16-17; L. 273-274; L. 321-322; L. 329-330; L. 374-375; L. 405-406; L. 460-461; L. 558-559; L. 595-596).
- Introduce the concept of virtual topography and its potential influence on the descent in the introduction, including the respective references (e.g., in the paragraph spanning L. 66-78).
- Discuss the limitations of the gravity wave perspective and of our manuscript more carefully (in a new limitations section 5.3 – see also next reviewer comment).

**Alternative descent mechanism ignored:** In the introduction the paper describes two main mechanisms that have been proposed in the literature to explain the descent - gravity waves and hydraulics - differential density (colder air upstream descending to its level of neutral buoyancy downstream). The claim in lines 75 - 77 that the hydraulic approach is applicable to shallow but not deep foehn is not correct (e.g. Armi & Mayr, 2007; Armi & Mayr, 2015; Winters & Armi, 2014).

The present paper only explores the gravity wave mechanism and ignores the hydraulic one despite the rich material available from the numerical simulations. The discussion subsection 5.2 ("What other factors influence the descent?") does not even mention the second mechanism anymore! This is a pity because the study has ample data to test both mechanisms whether they are the reasons for the descent and thus provide more material to further the discussion in the foehn community. The testable characteristic for the hydraulic response is that substantial descent starts when the overflowing air becomes colder (in terms of potential temperature) relative to the downstream air at altitudes below which it descends. Similarly, there need to be testable characteristics for the gravity wave mechanism that avoid, for example, the speculative attribution of the "intermittent periods of weak vertical motion" (lines 450-451) in the April 2018 case study to the "variability of wave-induced subsidence". With the alternative explanation this could be directly tested by examining whether the upstream air had become colder relative to the downstream one. However, the most important aspect to test for the adequacy of the mechanisms is the evolution of the onset of subsidence in the simulations.

The reviewer is correct in stating that we strongly focus on the gravity wave mechanism, but only briefly touch on the hydraulic/density-driven perspective as a potential mechanism for the descent. It was never the goal of the study to provide a comprehensive analysis of the different descent mechanisms. To highlight this more clearly, we will modify the manuscript as follows:
- The title of the study is too general, implying that we are doing such a comprehensive analysis. In the revised version, we will change the title to avoid implying a comprehensive examination of the underlying mechanisms.

- Since we do not provide an analysis of all possible descent mechanisms, we will explicitly mention this limitation of our study in the newly created discussion section (new section 5.3, see also first reviewer comment).
- We understand the reviewer's concern that the hydraulic mechanism was not even mentioned in the discussion section 5.2, and we apologize for this omission. Section 5.2 will be extended in the revised manuscript to include the hydraulic mechanism, including references to the relevant literature (e.g., Armi and Mayr, 2007; Armi and Mayr, 2015).
- We will also modify L. 75-77 to explicitly state that hydraulics have been applied to both shallow *and* deep foehn events in the respective previous studies.

Of course, a comprehensive study of 'all' mechanisms of foehn descent would be most welcome. However, we think that this would go far beyond what we can present in this paper, whose principal goal is to establish a Lagrangian framework for characterizing foehn descent. Nevertheless, we have started such an analysis (Fig. R1), taking into account several potential factors influencing the descent along the Rätikon, namely:
- the low-level stratification in the valley, assuming that a weakly stratified valley atmosphere favors the descent along the Rätikon.
- the wind speed upstream of the Rätikon, assuming that a strong upstream flow promotes the descent.
- the flow-splitting upstream of the considered Rhine Valley section, assuming the mass flux into the Rhine valley affects the foehn descent along the Rätikon by reasons of continuity.
- the maximum height difference of the 3-km isentrope above the Falknis peak within the hotspot region, assuming that a large height difference corresponds to

[Figure]

***Figure R1.*** *Overview of the preliminary analysis of different controlling factors on the descent. (a) An example time step of forward trajectories starting along a line near Bad Ragaz (upstream of Rätikon hotspot). These forward trajectories were used to determine to what extent the air is directed westward into the Seez Valley, and to what extent it is directed into the Rhine Valley. (b) The fraction of descent (i.e., the number of descent segments relative to all descent segments in the Rätikon hotspot) binned according to the potential temperature difference between 1.5 km AGL and the surface at Vaduz (see location in Fig. 7c in the manuscript). Shown are the relations for the Feb 2017 case study (green), the Apr 2018 case study (blue), and for all simulated cases combined (brown). (c) Same but for the mean upstream wind speed at 3 km AMSL along an upstream line along the Rätikon. (d) Same but for the fraction of air passing through the Rhine Valley. This fraction is calculated by comparing the number of forward trajectories intersecting a cross section perpendicular to the Rhine Valley relative to all forward trajectories (see also panel a). (e) Same but for the maximum height difference of the 3-km isentrope above the Falknis peak within the hotspot region (see location in Fig. 7c in the manuscript).*

a strongly inclined isentrope and thus a stronger descent. This could also be considered as a proxy for the maximum height difference of the virtual topography in the target region.

However, the interpretation of these preliminary results proved to be challenging. For instance, descent can occur under relatively stable stratification in the Rhine Valley (Fig. R1b) and under strongly varying flow splitting regimes (Fig. R1d). This calls for a more systematic investigation to disentangle the different factors influencing the descent. In the current manuscript, we refrained from doing so, also in order to not extend the manuscript's already substantial length.

Similar to our preliminary analysis, the reviewer suggests a testable characteristic for the hydraulic response, namely the potential temperature difference of the overflowing air compared to the air below which it descends. However, we think that this testable characteristic would not provide an unambiguous answer with respect to the underlying mechanism. In fact, upstream air that is potentially colder than the downstream air at the same level, and thus descends, could also be associated with a gravity wave. Extending this reasoning, we do not explore the role of the two mechanisms (hydraulic response, gravity wave mechanism), since we lack the testable characteristics to unambiguously disentangle them. We will mention this in our limitation section.

In addition to the physics of the foehn descent, we want to highlight that the study also introduces a sophisticated and, as far as we see, novel method for diagnosing foehn descent. So far, most studies have looked at the foehn descent from a Eulerian perspective, whereas we present a Lagrangian view. We think that this methodological aspect should be well recognized, as it allows for example to diagnose descent time scales. It is a pity that this novelty was not adequately appreciated by the reviewer, which can be attributed to the fact that we put too little focus on it in the text. We will emphasize this methodological aspect more in the revised manuscript, e.g., by referring to it already in the title, but also by specifically addressing it in the abstract and the conclusions of the paper.

**Questionable whether numerical simulations correctly reproduce processes that lead to downstream descent:** COSMO, the numerical model used for the simulations, has terrain-following coordinates near the surface. The simulation setup also uses horizontal diffusion (lines 179-181). If the numerical diffusion acts along model surfaces, which are slanted in complex terrain, artificial vertical, cross-isentropic mixing will ensue since in general the terrain-following coordinates cross isentropes in a stably stratified atmosphere. As a result, air will not separate from steep downward sloping topography as in reality, but rather descend. There are no figures in the paper to substantiate this claim. However, this inference is supported by the fact that the strongest descent in the model simulations occurs along the steepest slopes behind the tallest mountains (lines 313-315 and Fig. 4b) whereas in reality steeper slopes are likelier to lead to flow separation. Pressure perturbations from a gravity wave alone will not be strong enough to force the flow down by 1500 m in stable stratification.
Observations in the ocean (Knight Inlet; Farmer & Armi, 2001, and references therein) and in the atmosphere (Owens Valley during the T-REX campaign; Mayr & Armi, 2010 see Figs. 5 and 6; Armi & Mayr, 2011) confirm that flow separates along steep slopes before it later

descends when small-scale mixing forms a wedge of nearly stagnant and neutrally stratified air (away from the terrain) that separates the descending flow from the flow aloft. The detailed observations of the oceanic "foehn" flow into Knight inlet demonstrated the importance of correctly simulating the boundary layer separation. Numerical simulations with an atmospheric model (Afanasyev & Peltier, 2001) and an oceanic model (Cummins, 2000) erroneously produced a large overturning wave soon after the flow started flowing over the sill (the oceanic equivalent of a mountain crest) contrary to what the observations showed. Only when Cummins (2000) modified the topography to force boundary layer separation did the oceanic model correctly simulate the evolution of the flow. The authors will need to show that the model adequately handles flow separation; currently there are no figures in the paper that would allow one to do that. This can be done by first examining the time it takes for the descent along the slopes to become established. If this time is not (much) larger than $2\pi / N$ then the simulation will be incorrect. Second, the evolution of the descent must be visually inspected for congruence with observations of the initial flow separation and the way the wedge of the nearly stagnant and neutrally stratified air is formed that isolates the descending air from the flow above.

The reviewer expresses a fundamental issue: Is the COSMO model, and NWP at the kilometer-scale in general, capable of correctly capturing the descent in the lee of orographic obstacles? In fact, we argue that there are a few good reasons to believe that the essence is reasonably captured:

- The reviewer claims that the horizontal diffusion in the model simulations leads to artificial vertical mixing and thus introduces an unwanted, stronger descent in the lee of local mountain peaks. COSMO does calculate the horizontal diffusion along slanted model surfaces, but the horizontal diffusion along slopes is corrected by orographic flux limiting (see also Doms and Baldauf, 2021). The diffusive fluxes are gradually decreased as the elevation difference between adjacent grid points (i.e., the steepness of the coordinate surfaces) increases. When the elevation difference is greater than 250 m, the horizontal diffusive fluxes are set to zero. Using a grid spacing of 1.1 km, this results in a maximum slope angle of ~13°, above which the fluxes are set to zero. This value is well below the steepest slopes in the model (~30°). Considering this flux limiting, we are optimistic that the artificial vertical mixing should be less of an issue than suggested by the reviewer.

- Overall, our foehn episodes simulated by COSMO agree reasonably well with observations. A comparison between some of the COSMO simulations and station observations is provided in Jansing (2023). While mesoscale NWP simulations of foehn are known to be associated with distinct model biases (e.g., Wilhelm, 2012; Sandner, 2020), the mesoscale forcing is often adequately represented (e.g., Umek et al., 2021). However, if the COSMO model would not be able to capture the essential mechanisms of foehn descent, we would also not expect the model to reproduce typical foehn characteristics at valley stations (e.g., temperature increase upon foehn onset). Of course, there are also foehn episodes that are not represented well in the simulations, indicating that some mechanisms are still missed by the model. As an example, the representation of cold-air pools prior to foehn onset in mesoscale NWP simulations is still challenging (Umek et al., 2021).

[Figure]

**Figure R2.** *Vertical cross sections of vertical wind (colormap), isentropes (gray contours) and wind along cross sections (vectors). The topography is indicated by gray shading. The map inset shows the location of the vertical cross section (red, dashed line). The yellow and the green arrows indicate estimations of the downstream half-width ($A_d$) and the maximum height of the peak ($h_m$).*

[Figure]

**Figure R3.** *Regime diagram of different leeside flow responses as a function of the non-dimensional mountain height (N\*h/U) and the leeside slope of the obstacle ($h_m/A_d$). Reproduction of Figure 5.8 from Baines (1995), figure copied from Ambaum and Marshall (2005). The blue lines indicate the values estimated from the vertical cross section in Fig. R2a.*

Another critical comment from the reviewer concerns the representation of flow separation in the model. The reviewer suggests that the time it takes for the descent to become established along the slopes should be checked. However, inserting a typical value for N (~0.02 $s^{-1}$) leaves us with $2 * \pi / N \approx 5$ min, which is well below the output frequency of the 3D fields for our simulations. Such an approach is therefore not feasible for us.

Following Baines (1995), leeside flow separation does especially occur along steep slopes and for low values of the non-dimensional mountain height (N \* $h_m$/U). In Fig. R2., we show two vertical cross sections for the Feb 2017 and the Apr 2018 events going through Mont Blanc, which corresponds to the highest peak of the Alps that is associated with a steep northern slope. Following the regime diagram (Fig. R3) of Baines (1995), and inserting numbers roughly estimated from the vertical cross sections (see also yellow and green arrows in Fig. R2a) to calculate the leeside flow response, yields:

$h_m / A_d$ = 3000 m / 8 km = 0.375
N \* $h_m$ / U = 0.02 $s^{-1}$ \* 3000 m / 25 $ms^{-1}$ = 2.4

The estimated values suggest that the flow either should not separate on the obstacle, or that post-wave separation should occur (see blue lines in Fig. R3). This fits well with the observed pattern along the two vertical cross sections (Fig. R2, but also for other times of these two events) and contradicts the reviewer's statement that we should expect flow separation. Of course, it is not guaranteed that the flow response will be exactly the same in reality, as the model topography is smoothed and wind speed and stratification may be biased in the simulations. In conclusion, the flow response in the model is consistent with the expectations from theory, but it is unclear how well this matches the real flow response in the very rough and complex terrain of the Alps. We will therefore emphasize this last point as a limitation of our study (see also issue 3 raised by reviewer 2 on potential model problems).

**Reviewer 2**

We want to express our thanks to the reviewer for thoroughly reading our manuscript and for the positive evaluation. The reviewer's comments and input will help us to improve the manuscript significantly. We provide answers to all of his major and minor issues below.

**Major issue**

1. In the abstract the authors claim to investigate the descent process "with unprecedented detail". Indeed their study identifies foehn descent in a more spatio-temporally extensive dataset then previously, but their is no detailed analysis of the physical processes resulting in the downward motion of the air parcels. Discussion of the physical processes is limited to inference from a few cross-sections for two case-studies. Given the simulation data that they have, it would be very interesting to try and quantify the causes of downward acceleration of air parcels (buoyancy, vertical pressure gradient, …). They allude to this possibility in the conclusion, which is fine and I would encourage to highlight this even more. Indeed the paper would benefit from a more detailed physical analysis, but at least the abstract needs to be modified to accurately represent the contents of the paper.

We fully agree with the reviewer's comment. It is a limitation of our paper that we do not investigate the actual physical mechanisms of the descent in more detail. We will adjust the abstract and also mention this limitation more clearly in a separate discussion section that explicitly mentions all the limitations of our study (see also answers to issues 1 and 2 raised by reviewer 1). We will also further emphasize the potential for future studies investigating the descent mechanisms with our dataset.

2. The section of the introduction starting at p. 4, l.121 is not very well structured and open research questions could be stated more explicitly. Please consider rewriting.

We thank the reviewer for pointing this out, and we will re-write the respective section. In the following, we present some preliminary suggestions for the new structure:
- We will summarize the MAP results more concisely, emphasizing that the descent has so far only been diagnosed qualitatively and for the Rhine and Wipp valleys.
- We will highlight Saigger and Gohm (2022) as a key paper that motivated us to adopt the Lagrangian approach.
- We will highlight more explicitly the open questions related to the descent (e.g., the fundamental properties of descending air parcels).

3. Potential model issues in the representation of descent, e.g. potential issues of the turbulence scheme over complex terrain, need to be discussed in a more structured and prominent location (e.g. section in the conclusion / discussion). Hints at potential modeling problems are found throughout the manuscript, but it is failed to present them in structured manner and a discussion of their potential impacts on the results is missing.

As before, we agree with the reviewer's concern. We will explicitly list the potential model issues in our new limitation section. These potential model problems include:

- The descent is strongly influenced by the terrain characteristics, but the topography is smoothed in the COSMO model. In reality, the descent might thus occur at different locations and with a different magnitude compared to the model simulations. For instance, flow separation might occur at the sharp edges of local mountain peaks, a feature not represented in our simulations as the kilometer-scale grid spacing is still too coarse and the terrain is smoothed (see also issue 3 raised by reviewer 1).
- Turbulence/turbulent exchange is misrepresented in mesoscale NWP models (1D parameterizations that are designed for horizontally homogeneous terrain). This also effects the representation of foehn flows in such simulations (e.g., Vosper et al., 2018).
- Nocturnal CAP formation can inhibit the descent, as has been seen in the second case study (Section 4.2 of the manuscript). However, the maintenance of CAPs is difficult for mesoscale models (e.g., Umek et al., 2021). Therefore, the frequency and magnitude of the descent might be overestimated in our model simulations.
- Gravity wave patterns look different in large-eddy simulations compared to kilometer-scale simulations (Umek et al., 2022). This also suggests that the small-scale features of the descent are not adequately captured with kilometer-scale model simulations.

**Minor issues**

1. p. 2, l. 42: Why would foehn flows ignite forest fires? I would rather expect they are produce atmospheric conditions, that are more conducive to igniting fires.

Agreed, we will rephrase the sentence.

2. p. 2, l. 60: "foehn wall might inherit a key role for the downward acceleration": I do not understand this sentence: What is inherited and by what?

What we meant was that upon flowing over the crest, hydrometeors in the clouds (i.e., the foehn wall) begin to evaporate, causing latent cooling and thus a downward acceleration of the respective air. We will rephrase the respective sentence.

3. Section 2.1: in addition to the height of the lowest model level, it would be interesting to state the average vertical grid spacing in the valleys, e.g. the lowest 2 km.

Over flat terrain and at a distance from the orography, there are 34 model levels below 2 km, resulting in an average vertical grid spacing of ~60 m. We will add such a statement to Section 2.1.

4. Fig. 3: Would be interesting to see the distribution of foehn trajectories passing the locations of descent. I.e. is the distribution mirroring more frequent foehn events / large mass flux, e.g. along the Rhine valley and what is the percentage of foehn air parcels that descent in the specific regions.

We are not sure what the reviewer is referring to here. The distribution of foehn air parcels descending in specific regions can already be seen in Fig. 3, also when looking at the two histograms along the edges of Fig. 3. A case-by-case overview of the descent locations is found in Fig. S3 in the Supplement.

5. p. 11, l. 274: „gravity waves [...] force descending motion of air parcels": The Lagrangian diagnostic are just another perspective of the Eulerian velocity fields and vice-versa. So it cannot be claimed from the evidence presented that gravity waves force descent of air parcels. Descending air parcels in some sentence constitute the downward motion in the Eulerian perspective. Maybe a better wording instead of „force" would be „associated". Similar statements are made e.g. on p. 13 l. 321 and in a few other places, and also need modification.

We fully agree and we will modify all the respective statements (see also issue 1 raised by reviewer 1).

6. p. 12, l. 311: „exact relation": Given the scatter in the data, I do not agree that this is an exact relation.

Agreed, we will rephrase.

7. p. 16, l. 349: „and especially the cause for its formation": I would suggest to drop this statement. The following section does not provide any evidence for why a hotspot should form in particular behind the Rätikon and not other topographic features / locations along the Alpine chain.

We will drop this statement.

8. p. 17, l. 366 ff: I would suggest to first discuss the general characteristics of the foehn event before providing details on the time instances discussed afterwards to reduce repetition.

Agreed, we will omit these details here and only give more information later when discussing the time instances.

9. p. 27, l. 548: „constrains": I am not quite sure what you want to say here. Local terrain determines regions of descent / is anchoring regions of descent?

Yes, this is what we meant. We will rephrase.

10. p. 27. l. 551ff: Given the evidence (in the paper and the more general foehn literature), it would be more accurate to state that the elevation difference is an upper limit to the foehn descent and that (at least) in the model this is often (though not always - maybe you can even quantify how often) realized.

We will rephrase the statement. We are however not able to quantify how often this is the case, as our measure of elevation difference ($\Delta$topo) only provides the local elevation difference, while the slope of the terrain might still extend further. Moreover, our simulation results actually suggest that a considerable fraction of the air parcels descend further compared to the changes in the underlying topography (Fig. 5), implying that they must arrive closer to the ground than where they began their descent.

**Technical / language issues**

We will adjust all the technical issues mentioned.

**References**

Ambaum, M. H. and Marshall, D. P.: The effects of stratification on flow separation, J. Atmos. Sci., 62, 2618–2625, https://doi.org/10.1175/JAS3485.1, 2005.

Armi, L. and Mayr, G. J.: Continuously stratified flows across an Alpine crest with a pass: Shallow and deep föhn, Q. J. Roy. Meteorol. Soc, 133, 459–477, https://doi.org/10.1002/qj.22, 2007.

Armi, L. and Mayr, G. J.: Virtual and real topography for flows across mountain ranges, J. Appl. Meteorol. Clim., 54, 723–731, https://doi.org/10.1175/JAMC-D-14-0231.1, 2015.

Baines, P. G.: Topographic Effects in Stratified Flows, Cambridge University Press, 482 pp., 1995.

Doms, G. and Baldauf, M.: A description of the nonhydrostatic regional COSMO-model – Part I: Dynamics and Numerics, Tech. rep., https://doi.org/10.5676/DWD_pub/nwv/cosmo-doc_6.00_I, 2021.

Drobinski, P., Haeberli, C., Richard, E., Lothon, M., Dabas, A., Flamant, P., Furger, M., and Steinacker, R.: Scale interaction processes during the MAP IOP 12 south föhn event in the Rhine Valley, Q. J. Roy. Meteorol. Soc., 129, 729–753, https://doi.org/10.1256/qj.02.35, 2003.

Drobinski, P., Steinacker, R., Richner, H., Baumann-Stanzer, K., Beffrey, G., Benech, B., Berger, H., Chimani, B., Dabas, A., Dorninger, M., Dürr, B., Flamant, C., Frioud, M., Furger, M., Gröhn, I., Gubser, S., et al.: Föhn in the Rhine Valley during MAP: A review of its multiscale dynamics in complex valley geometry, Q. J. Roy. Meteorol. Soc., 133, 897–916, https://doi.org/10.1002/qj.70, 2007.

Gohm, A., Zängl, G., and Mayr, G. J.: South foehn in the Wipp Valley on 24 October 1999 (MAP IOP 10): Verification of high-resolution numerical simulations with observations, Mon. Weather Rev., 132, 78–102, https://doi.org/10.1175/1520-0493(2004)132<0078:SFITWV>2.0.CO;2, 2004.

Jansing, L.: A Lagrangian perspective on the Alpine Foehn, Ph.D. thesis, ETH Zurich, https://doi.org/10.3929/ethz-b-000619589, 2023.

Saigger, M. and Gohm, A.: Is it north or west foehn? A Lagrangian analysis of Penetration and Interruption of Alpine Foehn intensive observation period 1 (PIANO IOP 1), Weather Clim. Dyn., 3, 279–303, https://doi.org/10.5194/wcd-3-279-2022, 2022.

Sandner, V.: Verification of COSMO-1 forecasts of foehn breakthrough and interruption in the region of Innsbruck. M.S. thesis, University of Innsbruck, URL https://resolver.obvsg.at/urn:nbn:at:at-ubi:1-65459, 2022.

Umek, L., Gohm, A., Haid, M., Ward, H. C., and Rotach, M. W.: Large-eddy simulation of foehn–cold pool interactions in the Inn Valley during PIANO IOP 2, Q. J. Roy. Meteorol. Soc., 147, 944–982, https://doi.org/10.1002/qj.3954, 2021.

Umek, L., Gohm, A., Haid, M., Ward, H. C., and Rotach, M. W.: Influence of grid resolution of large-eddy simulations on foehn-cold pool interaction, Q. J. Roy. Meteorol. Soc., 148, 1840–1863, https://doi.org/10.1002/qj.4281, 2022.

Vosper, S. B., Ross, A. N., Renfrew, I. A., Sheridan, P., Elvidge, A. D., and Grubišić, V.: Current challenges in orographic flow dynamics: turbulent exchange due to low-level gravity-wave processes, Atmosphere, 9, 361, https://doi.org/10.3390/atmos9090361, 2018.

Wilhelm, M.: COSMO-2 model performance in forecasting foehn: a systematic process-oriented verification. Veröff. MeteoSchweiz, 89, 61 pp., URL https://www.meteoschweiz.admin.ch/dam/jcr:d46ec92c-946b-40ff-bed8-c528f048eed7/Veroeff-89.pdf, 2012.

Zängl, G. and Hornsteiner, M.: Can trapped gravity waves be relevant for severe foehn windstorms? A case study, Meteorol. Z., 16, 203–212, https://doi.org/10.1127/0941-2948/2007/0199, 2007.

Zängl, G., Chimani, B., and Häberli, C.: Numerical simulations of the foehn in the Rhine Valley on 24 October 1999 (MAP IOP 10), Mon. Weather Rev., 132, 368–389, https://doi.org/10.1175/1520-0493(2004)132<0368:NSOTFI>2.0.CO;2, 2004a.

Zängl, G., Gohm, A., and Geier, G.: South foehn in the Wipp Valley – Innsbruck region: Numerical simulations of the 24 October 1999 case (MAP-IOP 10), Meteorol. Atmos. Phys., 86, 213–243, https://doi.org/10.1007/s00703-003-0029-8, 2004b.

---

## Author Response (AR1)

**Author's response to reviewers**
* * *
**A novel Lagrangian framework for detecting and characterizing the descent of foehn from Alpine to local scales**
**Lukas Jansing | Lukas Papritz | Michael Sprenger**
**Submitted to WCD, egusphere-2023-1536**
**October 20, 2023**
* * *
**General statement:**

We would like to acknowledge both reviewers for taking the time to review our manuscript. We considered the valuable feedback and addressed the criticized aspects in a revised version of the manuscript and we are therefore confident that the revision resulted in a significantly improved manuscript. In this document, we respond to each of the reviewers' comments and outline how we addressed them (original reviewer comments in black, our answers in blue). We also mark the changes we made in the manuscript, including line numbers that point to the initial submission of the manuscript. Changes to the manuscript are also highlighted in the track change document.

**Reviewer 1:**

We appreciate the reviewer's evaluation of our manuscript and we take note of the strong criticism expressed by the reviewer. We concur that our manuscript overemphasizes the gravity wave mechanism, while wrongfully neglecting the hydraulic mechanism. We would like to stress that, with this study, we did not intend to provide a definitive answer to the underlying physical mechanisms leading to the descent. Instead, our primary objective was to introduce a novel Lagrangian framework that offers the possibility to investigate the descent using a mesoscale NWP model. We believe that this aspect was not emphasized enough and therefore also not sufficiently appreciated by the reviewer. Nevertheless, we addressed the reviewer's concerns by making appropriate changes to the manuscript. For more details, please refer to the specific responses to all three points below.

**Descent of air shapes gravity waves, not the other way around:** The descent of the air behind topography makes it possible for large gravity waves to form, rather than gravity waves causing the descent as the paper argues. When air impinges on an obstacle in stably stratified air, a gravity wave will be launched. Its shape and amplitude, however, will not only depend on the non-dimensional height of the obstacle (the product of relative height above incoming isentrope, stability and the inverse of speed), which is a function of the _upstream flow only,_ but also on the shape of the obstacle _including its lee side_.

Downstream descent is limited by real or virtual topography. Virtual topography is formed by the level of neutral buoyancy of the overflowing air with respect to the pre-existing air on the downwind side. Descent is only possible when the level of neutral buoyancy of the overflowing air in the downstream air is lower than the upstream altitude of the air, i.e.

when the potential temperature of the upstream air is lower than the pre-existing downstream air at the same altitude. Armi & Mayr (2015) showed this in detail with observations from instrumented aircraft (in-situ and cloud radar), dropsondes, radiosondes and satellite data.

The present paper even acknowledges the fact that gravity waves depend on how far air can descend (and not vice versa) in lines 568-569 of the conclusions by stating that "nocturnal cooling and the resulting formation of a cold-air pool impede strong descent of foehn air by effectively attenuating the mountain waves" without concluding that the descent cannot go further than to where this isentrope is located on the downstream side.

The third row of Fig. 10 (09 UTC 28 Feb) shows minimal gravity wave activity because the underlying virtual topography produced by nocturnal cooling is so smooth. The previous and the subsequent afternoons (13 UTC) of rows 1 and 4, respectively, have warmer leeward temperatures caused by daytime warming mainly by sensible heat fluxes (and initially by the turbulent erosion of the cold pool). Consequently gravity waves are larger.

Many thanks for this detailed discussion of the effect of stratification and virtual topography on the descent of air parcels in the lee of mountain ranges. We partly agree with this comment, especially regarding the controlling factors that affect the shape and amplitude of the gravity waves excited by mountain peaks. We also agree with the statement that air in the lee will descend until it reaches the level of neutral buoyancy (under the assumption of adiabatic, stably stratified flow). However, we also disagree with several aspects raised by the reviewer:

- Based on the reviewer's statement, the virtual topography seems to be the most important factor for the characteristics of the resulting gravity wave, which we do not agree with. We do not think that gravity wave formation, e.g., along the Rätikon, is only controlled by the stratification and virtual topography to the lee of the mountain, but also by the upstream flow conditions (upstream stratification and upstream wind speed) and the shape of the obstacle (as mentioned by the reviewer at the beginning of his statement).

- We are convinced that gravity waves play a more active role in the descent than the reviewer's comment suggests. Several studies have shown that descending flow and locally enhanced foehn winds are associated with gravity waves forming in the lee of local orographic features. Relevant for our target region are, e.g., the studies by Drobinski et al. (2003, 2007), Gohm et al. (2004), Zängl and Hornsteiner (2007), Zängl et al. (2004a, b). A more active role was also recently attributed to gravity waves in the west foehn in the Inn Valley (Saigger and Gohm, 2022). They state that the penetration of the westerly flow into the Inn Valley is partly controlled by the formation (or absence) of a gravity wave at the western boundary of the valley. All the aforementioned studies show that an 'active' role can and (partly) must be attributed to gravity waves in controlling and/or modulating the descent and the near-surface winds during foehn events. Of course, we acknowledge that stratification and virtual topography can be important controlling factors, as explicitly shown in the studies provided by the reviewer. However, we argue against such a strict and exclusive role of these two parameters. In addition, the level of neutral buoyancy (LNB), and thus the virtual topography, can also be modulated by the downstream effects of gravity waves, so that a clear separation of the different factors may not be possible.

- In our opinion, descending motion is an intrinsic feature of orographic gravity waves. The descent of air and the associated gravity wave occur and accentuate simultaneously. The reviewer's statement, in our opinion, overemphasizes the role of

the virtual topography and neglects the intrinsically coupled nature of the two phenomena. Is it the gravity wave that shapes the descent of the air, or does the descent of the air shape the gravity wave? Since these two effects of stratified flow past orography are intrinsically coupled, there is no clear answer, but the reviewer only considers one side.

Nevertheless, the reviewer raises an important point by highlighting that our study is biased towards a gravity wave perspective. In the revised manuscript, we therefore made the following adjustments:

- We now emphasize that the descent is just an intrinsic feature of gravity waves, rather than claiming a unidirectional causal relationship between these two aspects. To this end, we rephrased numerous statements related to gravity waves:
  - L. 13: We rephrased.
  - L. 16-20: We rephrased.
  - L. 99: We rephrased.
  - L. 102-103: We rephrased as the sentence implied a unidirectional causal relationship of gravity waves and descent.
  - L. 121ff: We completely re-wrote this paragraph (see also issue 2 by reviewer 2). The new paragraph puts less emphasis on gravity waves.
  - L. 264: We rephrased.
  - L. 273-276: We completely dropped these statements, as they were of purely speculative nature.
  - L. 321-322: We rephrased and included another influencing factor here, namely the potential of reaching the LNB prior to arriving at the leeside valley floor.
  - L. 330: We again dropped this statement, as it was of speculative nature.
  - L. 373-374: We dropped this speculative statement.
  - L. 374-375: We rephrased.
  - L. 379: We rephrased.
  - L. 406: We rephrased.
  - L. 422-423: We included the stable stratification as a potential reason for the absence of notable descent at this time instant. Note however, that it is probably *not* the reason for the minimal gravity wave activity and the absence of descent. Considering vertical cross sections along line c1 (see Fig. 7b in manuscript), the wave activity ceases *prior* to the increase in stability, and instead coincides with the decrease in wind speed at crest level (not shown).
  - L. 434-435: We rephrased to highlight the inherent interconnection of descent and gravity waves instead of claiming a unidirectional relationship.
  - L. 450-451: We dropped this speculative statement.
  - L. 461: We rephrased.
  - L. 508: We rephrased.
  - L. 558: We rephrased.
  - L. 563: We rephrased to not imply a causal relationship anymore.
  - L. 596: We rephrased.
- We introduced the concept of virtual topography and the importance of cross-barrier potential temperature differences in influencing the descent:
  - L. 22-23: The concept is mentioned in the abstract.
  - L. 73: The concept is introduced in the introduction.

o L. 121f: We now mention it as potential influencing factor when formulating the research questions. Note that we re-wrote this entire paragraph (see issue 2 raised by reviewer 2).
o L. 474: We now explicitly state that the descent is limited as air parcels quickly reach the LNB and also refer to the concept of virtual topography here.
o We discuss the concept of virtual topography in the context of the Apr 2018 case study in the discussion section (L. 522-523).
o We mention the concept in the conclusions (L. 569 & L. 606-607).
- We discuss the limitations of the gravity-wave perspective, as well as other limitations of our study, in a new limitation section 5.3 (L. 538ff).

**Alternative descent mechanism ignored:** In the introduction the paper describes two main mechanisms that have been proposed in the literature to explain the descent - gravity waves and hydraulics - differential density (colder air upstream descending to its level of neutral buoyancy downstream). The claim in lines 75 - 77 that the hydraulic approach is applicable to shallow but not deep foehn is not correct (e.g. Armi & Mayr, 2007; Armi & Mayr, 2015; Winters & Armi, 2014).

The present paper only explores the gravity wave mechanism and ignores the hydraulic one despite the rich material available from the numerical simulations. The discussion subsection 5.2 ("What other factors influence the descent?") does not even mention the second mechanism anymore! This is a pity because the study has ample data to test both mechanisms whether they are the reasons for the descent and thus provide more material to further the discussion in the foehn community. The testable characteristic for the hydraulic response is that substantial descent starts when the overflowing air becomes colder (in terms of potential temperature) relative to the downstream air at altitudes below which it descends. Similarly, there need to be testable characteristics for the gravity wave mechanism that avoid, for example, the speculative attribution of the "intermittent periods of weak vertical motion" (lines 450-451) in the April 2018 case study to the "variability of wave-induced subsidence". With the alternative explanation this could be directly tested by examining whether the upstream air had become colder relative to the downstream one. However, the most important aspect to test for the adequacy of the mechanisms is the evolution of the onset of subsidence in the simulations.

The reviewer is correct in stating that we strongly focus on the gravity wave mechanism, but only briefly touch on the hydraulic/density-driven perspective as a potential mechanism for the descent. It was never the goal of the study to provide a comprehensive analysis of the different descent mechanisms. To highlight this more clearly, we modified the manuscript as follows:
- The title of the study is too general, implying that we are doing such a comprehensive analysis. In the revised version, we changed the title to avoid implying a comprehensive examination of the underlying mechanisms.
- L. 3: We included the hydraulic approach as one of the theoretical foundations
- L. 75-75: We now explicitly state that hydraulics have been applied to both shallow *and* deep foehn events in the respective previous studies and we also cite two example studies.
- We understand the reviewer's concern that the hydraulic mechanism was not even mentioned in the discussion section 5.2, and we apologize for this omission. Section

5.2 has been extended in the revised manuscript to include the hydraulic mechanism (L. 527f), including references to the relevant literature (e.g., Armi and Mayr, 2011; Armi and Mayr, 2015).

- Since we do not provide an analysis of all possible descent mechanisms, we explicitly mention this limitation of our study in the newly created discussion section (new section 5.3, see also first reviewer comment).

Of course, a comprehensive study of 'all' mechanisms of foehn descent would be most welcome. However, we think that this would go far beyond what we can present in this paper, whose principal goal is to establish a Lagrangian framework for characterizing foehn descent. Nevertheless, we have started such an analysis (Fig. R1), taking into account several potential factors influencing the descent along the Rätikon, namely:

- the low-level stratification in the valley, assuming that a weakly stratified valley atmosphere favors the descent along the Rätikon.
- the wind speed upstream of the Rätikon, assuming that a strong upstream flow promotes the descent.
- the flow-splitting upstream of the considered Rhine Valley section, assuming the mass flux into the Rhine valley affects the foehn descent along the Rätikon by reasons of continuity.
- the maximum height difference (i.e., the difference of the maximum and the minimum height within the hostpot region) of the 3-km isentrope (i.e., the isentrope at 3 km AMSL) above the Falknis peak, assuming that a large height difference corresponds to a strongly inclined isentrope and thus a stronger descent. This could also be considered as a proxy for the maximum height difference of the virtual topography in the target region.

[Figure]

**Figure R1.** *Overview of the preliminary analysis of different controlling factors on the descent. (a) An example time step of forward trajectories starting along a line near Bad Ragaz (upstream of Rätikon hotspot). These forward trajectories were used to determine to what extent the air is directed westward into the Seez Valley, and to what extent it is directed into the Rhine Valley. (b) The fraction of descent (i.e., the number of descent segments relative to all descent segments in the Rätikon hotspot) binned according to the potential temperature difference between 1.5 km AGL and the surface at Vaduz (see location in Fig. 7c in the manuscript). Shown are the relations for the Feb 2017 case study (green), the Apr 2018 case study (blue), and for all simulated cases combined (brown). (c) Same but for the mean upstream wind speed at 3 km AMSL along an upstream line along the Rätikon. (d) Same but for the fraction of air passing through the Rhine Valley. This fraction is calculated by comparing the number of forward trajectories intersecting a cross section perpendicular to the Rhine Valley relative to all forward trajectories (see also panel a). (e) Same but for the maximum height difference of the 3-km isentrope above the Falknis peak within the hotspot region (see location in Fig. 7c in the manuscript).*

However, the interpretation of these preliminary results proved to be challenging. For instance, descent can occur under relatively stable stratification in the Rhine Valley (Fig. R1b) and under strongly varying flow splitting regimes (Fig. R1d). This calls for a more systematic investigation to disentangle the different factors influencing the descent. In the current manuscript, we refrained from doing so, also in order to not extend the manuscript's already substantial length.

Similar to our preliminary analysis, the reviewer suggests a testable characteristic for the hydraulic response, namely the potential temperature difference of the overflowing air compared to the air below which it descends. However, we think that this testable characteristic would not provide an unambiguous answer with respect to the underlying mechanism. In fact, upstream air that is potentially colder than the downstream air at the same level, and thus descends, could also be associated with a gravity wave. Extending this reasoning, we do not explore the role of the two mechanisms (hydraulic response, gravity wave mechanism), since we lack the testable characteristics to unambiguously disentangle them. We mentioned this in our new limitation section (L. 538ff).

In addition to the physics of the foehn descent, we want to highlight that the study also introduces a sophisticated and, as far as we see, novel method for diagnosing foehn descent. So far, most studies have looked at the foehn descent from a Eulerian perspective, whereas we present a Lagrangian view. We think that this methodological aspect should be well recognized, as it allows for example to diagnose descent time scales. It is a pity that this novelty was not adequately appreciated by the reviewer, which can be attributed to the fact that we put too little focus on it in the text. We now emphasize this methodological aspect more in the revised manuscript, e.g., by referring to it already in the title, but also by additionally addressing it in the abstract (L. 24-26) and the conclusions of the paper (L. 603-605).

**Questionable whether numerical simulations correctly reproduce processes that lead to downstream descent:** COSMO, the numerical model used for the simulations, has terrain-following coordinates near the surface. The simulation setup also uses horizontal diffusion (lines 179-181). If the numerical diffusion acts along model surfaces, which are slanted in complex terrain, artificial vertical, cross-isentropic mixing will ensue since in general the terrain-following coordinates cross isentropes in a stably stratified atmosphere. As a result, air will not separate from steep downward sloping topography as in reality, but rather descend. There are no figures in the paper to substantiate this claim. However, this inference is supported by the fact that the strongest descent in the model simulations occurs along the steepest slopes behind the tallest mountains (lines 313-315 and Fig. 4b) whereas in reality steeper slopes are likelier to lead to flow separation. Pressure perturbations from a gravity wave alone will not be strong enough to force the flow down by 1500 m in stable stratification.

Observations in the ocean (Knight Inlet; Farmer & Armi, 2001, and references therein) and in the atmosphere (Owens Valley during the T-REX campaign; Mayr & Armi, 2010 see Figs. 5 and 6; Armi & Mayr, 2011) confirm that flow separates along steep slopes before it later descends when small-scale mixing forms a wedge of nearly stagnant and neutrally stratified air (away from the terrain) that separates the descending flow from the flow aloft.

The detailed observations of the oceanic "foehn" flow into Knight inlet demonstrated the importance of correctly simulating the boundary layer separation. Numerical simulations with an atmospheric model (Afanasyev & Peltier, 2001) and an oceanic model (Cummins, 2000) erroneously produced a large overturning wave soon after the flow started flowing over the sill (the oceanic equivalent of a mountain crest) contrary to what the observations showed. Only when Cummins (2000) modified the topography to force boundary layer separation did the oceanic model correctly simulate the evolution of the flow. The authors will need to show that the model adequately handles flow separation; currently there are no figures in the paper that would allow one to do that. This can be done by first examining the time it takes for the descent along the slopes to become established. If this time is not (much) larger than $2\pi / N$ then the simulation will be incorrect. Second, the evolution of the descent must be visually inspected for congruence with observations of the initial flow separation and the way the wedge of the nearly stagnant and neutrally stratified air is formed that isolates the descending air from the flow above.

The reviewer expresses a fundamental issue: Is the COSMO model, and NWP at the kilometer-scale in general, capable of correctly capturing the descent in the lee of orographic obstacles? In fact, we argue that there are a few good reasons to believe that the essence is reasonably captured:

- The reviewer claims that the horizontal diffusion in the model simulations leads to artificial vertical mixing and thus introduces an unwanted, stronger descent in the lee of local mountain peaks. COSMO does calculate the horizontal diffusion along slanted model surfaces, but the horizontal diffusion along slopes is corrected by orographic flux limiting (see also Doms and Baldauf, 2021). The diffusive fluxes are gradually decreased as the elevation difference between adjacent grid points (i.e., the steepness of the coordinate surfaces) increases. When the elevation difference is greater than 250 m, the horizontal diffusive fluxes are set to zero. Using a grid spacing of 1.1 km, this results in a maximum slope angle of ~13°, above which the fluxes are set to zero. This value is well below the steepest slopes in the model (~30°). Considering this flux limiting, we are optimistic that the artificial vertical mixing should be less of an issue than suggested by the reviewer.

- Overall, our foehn episodes simulated by COSMO agree reasonably well with observations. A comparison between some of the COSMO simulations and station observations is provided in Jansing (2023). While mesoscale NWP simulations of foehn are known to be associated with distinct model biases (e.g., Wilhelm, 2012; Sandner, 2020), the mesoscale forcing is often adequately represented (e.g., Umek et al., 2021). However, if the COSMO model would not be able to capture the essential mechanisms of foehn descent, we would also not expect the model to reproduce typical foehn characteristics at valley stations (e.g., temperature increase upon foehn onset). Of course, there are also foehn episodes that are not represented well in the simulations, indicating that some mechanisms are still missed by the model. As an example, the representation of cold-air pools prior to foehn onset in mesoscale NWP simulations is still challenging (Umek et al., 2021).

[Figure]

***Figure R2.*** *Vertical cross sections of vertical wind (colormap), isentropes (gray contours) and wind along cross sections (vectors). The topography is indicated by gray shading. The map inset shows the location of the vertical cross section (red, dashed line). The yellow and the green arrows indicate estimations of the downstream half-width ($A_d$) and the maximum height of the peak ($h_m$).*

[Figure]

***Figure R3.*** *Regime diagram of different leeside flow responses as a function of the non-dimensional mountain height (N\*h/U) and the leeside slope of the obstacle ($h_m/A_d$). Reproduction of Figure 5.8 from Baines (1995), figure copied from Ambaum and Marshall (2005). The blue lines indicate the values estimated from the vertical cross section in Fig. R2a.*

Another critical comment from the reviewer concerns the representation of flow separation in the model. The reviewer suggests that the time it takes for the descent to become established along the slopes should be checked. However, inserting a typical value for N (~0.02 $s^{-1}$) leaves us with 2 \* $\pi$ / N ≈ 5 min, which is well below the output frequency of the 3D fields for our simulations. Such an approach is therefore not feasible for us.

Following Baines (1995), leeside flow separation does especially occur along steep slopes and for low values of the non-dimensional mountain height (N \* $h_m$/U). In Fig. R2., we show two vertical cross sections for the Feb 2017 and the Apr 2018 events going through Mont Blanc, which corresponds to the highest peak of the Alps that is associated with a steep northern slope. Following the regime diagram (Fig. R3) of Baines (1995), and inserting numbers roughly estimated from the vertical cross sections (see also yellow and green arrows in Fig. R2a) to calculate the leeside flow response, yields:

$h_m$ / $A_d$ = 3000 m / 8 km = 0.375
N \* $h_m$ / U = 0.02 $s^{-1}$ \* 3000 m / 25 $ms^{-1}$ = 2.4

The estimated values suggest that the flow either should not separate on the obstacle, or that post-wave separation should occur (see blue lines in Fig. R3). This fits well with the observed pattern along the two vertical cross sections (Fig. R2, but also for other times of these two events) and contradicts the reviewer's statement that we should expect flow separation. Of course, it is not guaranteed that the flow response will be exactly the same in reality, as the model topography is smoothed and wind speed and stratification may be biased in the simulations. In conclusion, the flow response in the model is consistent with the expectations from theory, but it is unclear how well this matches the real flow response in the very rough and complex terrain of the Alps. We therefore emphasize this last point as a limitation of our study in Section 5.3 (see also issue 3 raised by reviewer 2 on potential model problems).

**Reviewer 2**

We want to express our thanks to the reviewer for thoroughly reading our manuscript and for the positive evaluation. The reviewer's comments and input helped us to improve the manuscript significantly. We provide answers to all of his major and minor issues below.

**Major issue**

1.  In the abstract the authors claim to investigate the descent process "with unprecedented detail". Indeed their study identifies foehn descent in a more spatio-temporally extensive dataset then previously, but their is no detailed analysis of the physical processes resulting in the downward motion of the air parcels. Discussion of the physical processes is limited to inference from a few cross-sections for two case-studies. Given the simulation data that they have, it would be very interesting to try and quantify the causes of downward acceleration of air parcels (buoyancy, vertical pressure gradient, …). They allude to this possibility in the conclusion, which is fine and I would encourage to highlight this even more. Indeed the paper would benefit from a more detailed physical analysis, but at least the abstract needs to be modified to accurately represent the contents of the paper.

We fully agree with the reviewer's comment. It is a limitation of our paper that we do not investigate the actual physical mechanisms of the descent in more detail. To account for this issue, we made several changes:

*   We changed the title of the manuscript to more clearly reflect the contents of the study (see also issue 2 by reviewer 1).
*   We rephrased the abstract (L. 3-8; L. 24-26) and also mention this limitation more clearly in a separate discussion section (Section 5.3; L. 538ff) that explicitly lists all the limitations of our study (see also answers to issues 1 and 2 raised by reviewer 1).
*   We also modified L. 541-542 in the conclusions to more accurately represent the contents of the paper.
*   We additionally emphasize the potential for future studies investigating the descent mechanisms with our dataset in the conclusions (L. 603-605).

2.  The section of the introduction starting at p. 4, l.121 is not very well structured and open research questions could be stated more explicitly. Please consider rewriting.

We thank the reviewer for pointing this out, and we re-wrote the respective section. We now structured the paragraph according to the three research questions that we pose afterwards, namely:

1)  *Where do air parcels descend during south foehn?* This question has only been addressed by three previous studies during MAP (Gohm et al., 2004, Zängl et al., 2004a, b), focusing solely on the Rhine Valley and the Wipp Valley. We emphasized this aspect more strongly.

2)  *How can the descent be characterized?* For this question, the study by Saigger and Gohm (2022) serves as a pioneering publication that showcases how the Lagrangian approach is ideal to address not only the question where the air descends, but also

how this process is characterized. We highlight that this approach is yet to be applied to south-foehn events.

3) *What governs the descent on a local scale?* The previous studies proposed different influencing factors that are linked to the descent (gravity waves, upstream flow splitting, cross-barrier potential temperature differences, …). Here we want to emphasize that the role of these different mechanisms is still an open question and we try to partly address it in our Section 4 that focuses on the descent in one particular hotspot (i.e., on a local scale).

3. Potential model issues in the representation of descent, e.g. potential issues of the turbulence scheme over complex terrain, need to be discussed in a more structured and prominent location (e.g. section in the conclusion / discussion). Hints at potential modeling problems are found throughout the manuscript, but it is failed to present them in structured manner and a discussion of their potential impacts on the results is missing.

As before, we agree with the reviewer's concern. We now explicitly list the potential model issues in our new limitation section. These potential model problems include:

- The descent is strongly influenced by the terrain characteristics, but the topography is smoothed in the COSMO model. In reality, the descent might thus occur at different locations and with a different magnitude compared to the model simulations. For instance, flow separation might occur at the sharp edges of local mountain peaks, a feature not represented in our simulations as the kilometer-scale grid spacing is still too coarse and the terrain is smoothed (see also issue 3 raised by reviewer 1).
- Turbulence/turbulent exchange is misrepresented in mesoscale NWP models (1D parameterizations that are designed for horizontally homogeneous terrain). This also effects the representation of foehn flows in such simulations (e.g., Vosper et al., 2018).
- Nocturnal CAP formation can inhibit the descent, as has been seen in the second case study (Section 4.2 of the manuscript). However, the maintenance of CAPs is difficult for mesoscale models (e.g., Umek et al., 2021). Therefore, the frequency and magnitude of the descent might be overestimated in our model simulations.
- Gravity wave patterns look different in large-eddy simulations compared to kilometer-scale simulations (Umek et al., 2022). This also suggests that the small-scale features of the descent are not adequately captured with kilometer-scale model simulations.

**Minor issues**

1. p. 2, l. 42: Why would foehn flows ignite forest fires? I would rather expect they are produce atmospheric conditions, that are more conducive to igniting fires.

Agreed, we rephrased the sentence.

2. p. 2, l. 60: "foehn wall might inherit a key role for the downward acceleration": I do not understand this sentence: What is inherited and by what?

What we meant was that upon flowing over the crest, hydrometeors in the clouds (i.e., the foehn wall) begin to evaporate, causing latent cooling and thus a downward acceleration of the respective air. We rephrased the respective sentence.

3.  Section 2.1: in addition to the height of the lowest model level, it would be interesting to state the average vertical grid spacing in the valleys, e.g. the lowest 2 km.

Over flat terrain and at a distance from the orography, there are 34 model levels below 2 km, resulting in an average vertical grid spacing of ~60 m. We added such a statement to Section 2.1 (L. 186).

4.  Fig. 3: Would be interesting to see the distribution of foehn trajectories passing the locations of descent. I.e. is the distribution mirroring more frequent foehn events / large mass flux, e.g. along the Rhine valley and what is the percentage of foehn air parcels that descent in the specific regions.

We are not sure what the reviewer is referring to here. The distribution of foehn air parcels descending in specific regions can already be seen in Fig. 3, also when looking at the two histograms along the edges of Fig. 3. A case-by-case overview of the descent locations is found in Fig. S3 in the Supplement.

5.  p. 11, l. 274: „gravity waves […] force descending motion of air parcels": The Lagrangian diagnostic are just another perspective of the Eulerian velocity fields and vice-versa. So it cannot be claimed from the evidence presented that gravity waves force descent of air parcels. Descending air parcels in some sentence constitute the downward motion in the Eulerian perspective. Maybe a better wording instead of „force" would be „associated". Similar statements are made e.g. on p. 13 l. 321 and in a few other places, and also need modification.

We agree that the paper does not provide any evidence for such a statement. Related to issue 1 of reviewer 1, we completely dropped these statements (L. 274-276). Further modifications have been made to numerous statements that address the association of gravity waves and the descent, see issue 1 of reviewer 1 for the further changes.

6.  p. 12, l. 311: „exact relation": Given the scatter in the data, I do not agree that this is an exact relation.

Agreed, we omitted the "exact".

7.  p. 16, l. 349: „and especially the cause for its formation": I would suggest to drop this statement. The following section does not provide any evidence for why a hotspot should form in particular behind the Rätikon and not other topographic features / locations along the Alpine chain.

We dropped this statement.

8.  p. 17, l. 366 ff: I would suggest to first discuss the general characteristics of the foehn event before providing details on the time instances discussed afterwards to reduce repetition.

Agreed, we omitted these details in L. 366ff and merged them with the paragraph in L. 384ff instead.

9.  p. 27, l. 548: „constrains": I am not quite sure what you want to say here. Local terrain determines regions of descent / is anchoring regions of descent?

Yes, this is what we meant. We rephrased.

10. p. 27. l. 551ff: Given the evidence (in the paper and the more general foehn literature), it would be more accurate to state that the elevation difference is an upper limit to the foehn descent and that (at least) in the model this is often (though not always - maybe you can even quantify how often) realized.

We rephrased the statement. We are however not able to quantify how often this is the case, as our measure of elevation difference ($\Delta$topo) only provides the local elevation difference, while the slope of the terrain might still extend further. Moreover, our simulation results actually suggest that a considerable fraction of the air parcels descend further compared to the changes in the underlying topography (Fig. 5), implying that they must arrive closer to the ground than where they began their descent.

**Technical / language issues**

1. p. 1, l. 7: „thereby" seems to be inadequate here, please modify.
We omitted "thereby" (we rephrased the sentence anyway).

2. p. 5, l. 153: „grid resolution" -> „grid spacing"?
Agreed, we corrected it.

3. p. 5, l. 158: „recent modeling studies"
We added "modeling".

4. p.7, l. 198: „accordingly" seems to be inadequate here, please modify.
We rephrased.

5. p. 8, l. 207: „compiled" > „computed"
We rephrased.

6. p. 14, l. 326: „thereby" seems to be inadequate here, please modify.
We omitted "thereby".

7. p. 17, l. 355: „thereby" seems to be inadequate here, please modify.
We rephrased.

8. p. 17, l. 360: please clarify whether the referred date of foehn break-through is identified in observations or model data (and if the former what observations and with what method). Reference to an older paper is not adequate.
The date was determined using the observational-based foehn index at Vaduz. The method is described in Dürr (2008). We added this statement to L. 360. Note that the foehn onset in the model occurs only with a minor temporal offset, which can be seen in timeseries of modelled surface variables (Fig. E4 in Jansing, 2023).

**References**

Ambaum, M. H. and Marshall, D. P.: The effects of stratification on flow separation, J. Atmos. Sci., 62, 2618–2625, https://doi.org/10.1175/JAS3485.1, 2005.

[revised manuscript text omitted]

---

## Referee Report (RR1)

**On the descent of the Alpine south foehn**

Lukas Jansing et al.

I thank the authors for their constructive and thoughtful responses to my review and the second reviewer's comments. The addition of subsection 5.3 on the limitations of the study is particularly helpful for readers to properly interpret the results. It also provides valuable suggestions for future research by the authors and other researchers. In light of these changes, I  withdraw my recommendation to reject the manuscript and instead recommend accepting the manuscript conditional on undertaking some major revision steps. The authors have produced such a unique dataset from which more insights can be gleaned!

*Note: Line numbers refer to the version of the revised manuscript that tracks the changes compared to the original version.*

**Major comments:**

1. **Scope of manuscript:** The new title clearly conveys the focus of the manuscript on a new method for detecting and characterizing descending air behind obstacles. This change should also be reflected in the abstract and throughout the manuscript:
   a. Abstract, second paragraph: Please specify which topographic features favor descent. The sentence "the small-scale elevation differences of the underlying terrain largely determine the magnitude of the descent" contradicts statements in the main article that the level of neutral buoyancy (virtual topography) is decisive, along with gravity waves (although I disagree with the latter).
   b. Line 25: Virtual topography, which per definition applies to the properties of the incoming flow, can be formed by many other mechanisms than nocturnal cooling.
   c. The last paragraph (lines 27++) can be removed as it is a summary of a summary and the parts about multiple case studies and different foehn regions can easily be incorporated earlier.
   d. Most importantly, the abstract should state that the results are based on numerical simulations, which have a considerable degree of uncertainty. As a result, the cause of descent cannot be definitively resolved. Instead, the authors should state that, within the limitations of the model simulations, they have identified characteristics of the descent.

2. The **explanation of hydraulic theory** should be expanded to explicitly state that it takes into account the most common south foehn situation in the Alps that air masses on the downstream side of the crest are colder, whether for synoptic, mesoscale or valley-scale (e.g. nocturnal cooling) reasons. Similarly, the one-sentence explanation (lines 92-93) why isentropes descend to the lee in the gravity wave theory needs to be expanded. If isentropes descend because of orographic drag, which (among other factors) depends on the effective height of the obstacle, then a smaller effective mountain due to the virtual topography *both upstream (by blocking) and downstream (by cooler air)* will cause a smaller deflection and thus make a descent to the floor of the downstream topography unlikely.

3. **Difficulty of simulations:** The manuscript cites the difficulty of simulations with a 1-km grid to properly handle the interaction of the flow with the cold pool (e.g. Umek et al. 2022)l. This difficulty is particularly relevant to south foehn in the Alps, where colder air is typically present on the northern side *already from below the crest onwards*, not just further downstream in the lowest parts of the valleys. This difficulty therefore affects the handling of the whole descent process in the numerical simulations, which must be clearly stated in the manuscript.

4. **Gravity wave vs. hydraulic explanations** of foehn descent and distinguishing between them: This is an excellent data set, despite the uncertainties of the numerical simulations! it may still be possible to get closer to finding a definitive answer to the question of gravity wave vs. hydraulic explanations of foehn descent. Here are some specific suggestions: First, after foehn air has descended the flow response will be indistinguishable between gravity wave and hydraulic explanation. It is therefore paramount to examine the **onset** of the foehn descent. I envision several possibilities of doing that:
   a. Examine vertical profiles upstream and downstream of the obstacle from before onset until after the onset, similar to Mayr and Armi (2010).
   b. Alternatively, find regions where foehn has descended as well as similar topographic obstacles behind which foehn has not descended yet. What are the differences? Are the upstream conditions not similar? Note that Reinecke and Durran (1990) found extreme sensitivity to initial conditions in a 70-member ensemble simulating foehn during the TREX campaign. Descent and consequently leeward wind speeds differed despite similar upstream conditions prior to the onset of foehn. This undermines the argument in the manuscript that increased upstream wind speed would favor descent.
   c. Examine vertical sections across the obstacle from before till after onset, and also for regions where foehn does not descend much: Do the wavelengths of the gravity waves correspond to the shape of the virtual topography or to that of the real topography upstream experienced by the impinging flow (to test if your statement that incoming wind speed also plays a role; cf. Mayr and Armi, 2010)?

5. **Extracting the effects of gravity waves:** Section 5.3 on the limitations of the study states in lines 598-599 that the effects of gravity waves are difficult to extract from mesoscale NWP data. Although this statement is part of a discussion on obtaining a Lagrangian momentum budget, I think it also holds more generally. Maybe the emphasis on gravity waves in the previous parts of the article can be reduced and the focus put more strongly on the core findings of the paper - how to identify descending particles, and to examine their characteristics?!

**Minor comments:**

1. **Lines 4-5:** Can you be quantitative about the fraction of all descents examined that are dry adiabatic and along isentropes. And also state, when and where descent is not along isentropes?
2. **Line 7:** Care needs to be used with the formulation "novel approach" here and in other parts of the manuscript. It is misleading since e.g. Miltenberger et al. (2016) and Saigger and Gohm (2022), both of which are cited in the manuscript, also used detailed trajectory analysis.

3. **Lines 67-68:** With negative buoyancy from evaporation, should there not be convection? Also: "waterfall theory" is an expression that is not commonly used. Should that expression describe a hydraulic response such as in water descending behind a weir?
4. **Lines 99-100:** Delete that sentence as it is not congruent with the previous exposition of two competing explanations of foehn. It does not bolster arguments in the manuscript. "Intrinsic" means independent of factors outside of a system, in this case, outside of gravity waves. Hydraulic theory posits that gravity waves are launched as a *response* to air that descends.
5. Please **add** information of the temporal resolution at which model output is stored. You mention in your response only that it is longer than 5 minutes.
6. **Lines 203-205:** Please add the important information that along-level diffusion is turned off for slope angles of more than 13 degrees (in this case), as you stated in your response. Other readers, not just this reviewer, might be unaware of it.

**Textual comments:**

- Delete "novel" from title since the publication of the article implies that this is a new approach. (And it shortens the long title a little)
- Line 3: Delete "modern". Both theories are more than half of a century old.
- Line 9: "precisely" is not needed. It implies that an accuracy metric is specified in the manuscript - which there is none.

**References:**

Mayr, G. J., & Armi, L. (2010). The influence of downstream diurnal heating on the descent of flow across the Sierras. *Journal of Applied Meteorology and Climatology*, *49*(9), 1906-1912.

Miltenberger, A. K., Reynolds, S., & Sprenger, M. (2016). Revisiting the latent heating contribution to foehn warming: Lagrangian analysis of two foehn events over the Swiss Alps. *Quarterly Journal of the Royal Meteorological Society*, *142*(698), 2194-2204.

Reinecke, P. A., & Durran, D. R. (2009). Initial-condition sensitivities and the predictability of downslope winds. *Journal of the atmospheric sciences*, *66*(11), 3401-3418.

Saigger, M., & Gohm, A. (2022). Is it north or west foehn? A Lagrangian analysis of Penetration and Interruption of Alpine Foehn intensive observation period 1 (PIANO IOP 1). *Weather and Climate Dynamics*, *3*(1), 279-303.

Umek, L., Gohm, A., Haid, M., Ward, H. C., & Rotach, M. W. (2022). Influence of grid resolution of large-eddy simulations on foehn-cold pool interaction. *Quarterly Journal of the Royal Meteorological Society*, *148*(745), 1840-1863.

---

## Referee Report (RR2)

**On the descent of the Alpine south foehn**

Lukas Jansing et al.

The authors make this review process a real pleasure by so carefully responding to the reviews, undertaking additional analyses, and producing additional figures. Thanks!
The research they undertook is a treasure trove and a few more pieces have been lifted from it in the course of the review process. And a few more will be lifted in the course of what I think will be the last round of the revisions.

**Major comments:**

1. **Influence of up/downstream air mass difference on descent - up/downstream profiles:** I agree that it is difficult to determine a suitable upstream location for studying the influence of potential temperature differences on foehn descent within the complex topography of the Alps. The Sierra Nevada topography studied by Mayr and Armi (2010) is much simpler. The figures R1 and R2 produced for the response confirm this. I disagree with the authors on parts of their interpretation and urge to choose different upstream and downstream locations for the vertical potential temperature profiles.

   a. **Interpretation:** I attach an annotated version of the authors' Fig. R1 (Fig. A1-R1) and added a crucial missing part: the lowest elevation over which foehn can actually flow (estimated to be around 2.2 km). Contrary to what the response states, it becomes clear that the upstream air can only descend a few hundred meters at the first time shown (Fig. A1-R1a). The next time step (Fig. A1-R1b) confirms that upstream air needs to be colder for descent to the downstream bottom. The subsequent time steps (c-e) point to a problem with the choice of the upstream and downstream locations. I marked "cap" to show the top of downstream foehn layers. They are mostly well below the lowest possible crossing elevation and indicate that the upstream had been significantly modified by the many ups and downs between Milano and Vaduz as Fig. 3 of the manuscript shows. Milano-Vaduz is actually one of the longest distances where foehn descent is modeled..

   b. **Profile locations:** Therefore I suggest redoing the figures for a downstream location within the subregion d2, where the majority of trajectories descend much closer to the Alpine crest (Fig. 3 of manuscript) and use the computed trajectories to place the upstream location about 20 km upstream of the Alpine crest. Then add and discuss the figure in the main text of the manuscript.

2. **Hydraulic response and virtual topography:** The carefully prepared Figs. R3 and R4 (even with an inset of the cross-section location!) are a big help! Thanks! However, I disagree with their interpretation.

   a. **Descent limited to difference in potential temperature:** These figures actually show that the upstream air can only descend approximately as far as its level of neutral buoyancy on the downstream side! I marked up Fig. R3c-h and attach it as Fig. A2-R3. The obstacle I refer to In the following is between km 30 and 40 in the figure. In the first

cross section (left column, Fig. A2-R3c,e,g), the 292 K isentrope marked up in orange rises upstream as time progresses, i.e. the upstream air becomes colder. Between 5 UTC and 9 UTC the upstream air becomes still colder whereas the air downstream warms as is visible by the sinking of the 290 K and 288 K isentropes close to the next obstacle. In the final time step shown at 09 UTC, the 292 K isentrope has risen just far enough to cross the obstacle and can descend fairly far downstream with accompanying higher flow speeds. In the second cross section (second column, Figs. A2-R3d,f,h), a similar upstream cooling is shown by the rise of the 294 K isentrope (marked up in orange) and needed before the air can descend further downstream. The momentum of the descent is sufficiently large to *under*shoot the level of its neutral buoyancy and thus deform the isentrope downwards closest to the slope (which is reminiscent of positively buoyant air parcels in convection *over*shooting their level of neutral buoyancy).

b. **Waves are a response to virtual topography:** Fig. A2-R3 actually disentangles the seeming chicken-and-egg question of whether (gravity) waves modify the virtual topography or whether the waves are triggered by the virtual topography. A black outline in the second cross section of Fig. A2-R3 shows the region where the answer becomes obvious. There is no wave above the actual real topography. The wave is in response to the descending flow and the shape of the virtual topography that it causes. The wavelength corresponds to the virtual topography (an inverted peak) downstream of the real peak. For a further discussion, see Armi and Mayr (2011, section 4c) and Armi and Mayr (2015, section 3b first paragraph).

c. **Key features of hydraulic flow**: Focusing just on the last time step and the region between km 25 and 45 in Fig. A2-R3(g, h), one finds these key features of hydraulic flow: (i) descent of the overflowing layer already ahead of the hydraulic control location (= peak), (ii) asymmetric flow across the obstacle with accelerating flow on the downstream side, (iii) a rebound in a hydraulic jump to the conditions further downstream, and (iv) a less stably stratified and slow layer on the downstream side (between 298 K and 300 K) separating the foehn flow from the flow aloft.

Parts of Figs. R3 (or R4) should also be included in the final manuscript to discuss this topic.

References:

Armi, L. & Mayr, G.J. (2011): The Descending Stratified Flow and Internal Hydraulic Jump in the Lee of the Sierras. *Journal of Applied Meteorology and Climatology*, **50**, 1995–2011, https://doi.org/10.1175/JAMC-D-10-05005.1.

Armi, L. & Mayr, G. J. (2015): Virtual and Real Topography for Flows across Mountain Ranges. *Journal of Applied Meteorology and Climatology*, **54**, 723–731, https://doi.org/10.1175/JAMC-D-14-0231.1.

Mayr, G. J., & Armi, L. (2010). The influence of downstream diurnal heating on the descent of flow across the Sierras. *Journal of Applied Meteorology and Climatology*, *49*(9), 1906-1912. https://doi.org/10.1175/2010JAMC2516.1

[Figure]

Fig. A1-R1

[Figure]

Fig. A2-R3

---

## Author Response (AR2)

**Author's response to 2nd round of reviewer comments**
* * *
**A Lagrangian framework for detecting and characterizing the descent of foehn from Alpine to local scales**
**Lukas Jansing | Lukas Papritz | Michael Sprenger**
**Submitted to WCD, egusphere-2023-1536**
**December 15, 2023**
* * *
We would like to thank the reviewer for reading our manuscript a second time and we are pleased to note that the evaluation was more positive. We genuinely appreciate the effort the reviewer made to come up with many suggestions for improvement! To recognize the effort, we explicitly mention the reviewer in our acknowledgements. Whenever we have considered the comments to be actionable, we also implemented them in the manuscript. More detailed explanations, including specific statements regarding the changes we have made, are found below (our answers in blue; the line numbers refer to the first revised version of the manuscript).

**Major comments:**

1. **Scope of manuscript:** The new title clearly conveys the focus of the manuscript on a new method for detecting and characterizing descending air behind obstacles. This change should also be reflected in the abstract and throughout the manuscript:
   a. Abstract, second paragraph: Please specify which topographic features favor descent. The sentence "the small-scale elevation differences of the underlying terrain largely determine the magnitude of the descent" contradicts statements in the main article that the level of neutral buoyancy (virtual topography) is decisive, along with gravity waves (although I disagree with the latter).
   b. Line 25: Virtual topography, which per definition applies to the properties of the incoming flow, can be formed by many other mechanisms than nocturnal cooling.
   c. The last paragraph (lines 27++) can be removed as it is a summary of a summary and the parts about multiple case studies and different foehn regions can easily be incorporated earlier.
   d. Most importantly, the abstract should state that the results are based on numerical simulations, which have a considerable degree of uncertainty. As a result, the cause of descent cannot be definitively resolved. Instead, the authors should state that, within the limitations of the model simulations, they have identified characteristics of the descent.

We agree with the reviewer that the focus conveyed by the title should also be reflected in the abstract and the manuscript. We believe that this focus is indeed conveyed throughout the manuscript – our change in title from the first round of revisions only intended to more clearly highlight the key message. Nevertheless, we appreciate the reviewer's additional comments and have tried to incorporate them where we think they improve the manuscript:
   a. It is impossible for us to specify particular topographic features that favor descent more clearly than by referencing "local mountain peaks and chains". Looking at Fig. 3 in the

manuscript, a large number of mountain peaks and chains along the entire Alpine arc are associated with descending motion, so it is not feasible to make a specific list of these peaks. The statement "the small-scale elevation differences of the underlying terrain largely determine the magnitude of the descent" is inferred from the clear relationship between $\Delta z$ and $\Delta topo$ (Fig. 5 in the manuscript). As gravity waves themselves are strongly tied to the local terrain characteristics, this statement would still be valid if they were the decisive factor for the descent. However, considering the spread in Fig. 5, it is clear that other factors also play a role, such as the virtual topography. Taking all this into account, we rephrased our statement in the abstract, not implying a causal relationship anymore, and we now also refer to the influence by other factors (L. 10-11).

b. We agree that virtual topography can be formed by other processes (e.g., synoptic cold-air advection). However, in our study, we specifically investigated a case where it formed due to nocturnal cooling, so it would be misleading to generalize the statement in the abstract. However, we have rephrased it to make it clear that our statement refers only to the second case study and is not meant in a more general sense (L. 21-22).

c. We would like to retain these concluding remarks. Despite the partly repetitive nature, we consider it important to conclude the abstract with an overarching statement that highlights the significance of our work. Besides, such summarizing remarks also serve as a motivation for readers to delve deeper into our paper and build upon our work for future investigations.

d. We acknowledge the importance of explicitly mentioning this limitation in the abstract. In L. 5 we already state that we employ model simulations. In addition, we now added a sentence to L. 23 in order to clearly convey that, given the model limitations, our study does not intend to definitively resolve the causes for the descent.

2. The **explanation of hydraulic theory** should be expanded to explicitly state that it takes into account the most common south foehn situation in the Alps that air masses on the downstream side of the crest are colder, whether for synoptic, mesoscale or valley-scale (e.g. nocturnal cooling) reasons. Similarly, the one-sentence explanation (lines 92-93) why isentropes descend to the lee in the gravity wave theory needs to be expanded. If isentropes descend because of orographic drag, which (among other factors) depends on the effective height of the obstacle, then a smaller effective mountain due to the virtual topography both upstream (by blocking) and downstream (by cooler air) will cause a smaller deflection and thus make a descent to the floor of the downstream topography unlikely.

We appreciate the suggestion of the reviewer. Based on the comment, we have slightly revised and restructured the paragraph introducing the hydraulic theory to make this aspect clearer (L. 79ff).

Regarding the explanation of gravity waves and descent, we think our current statement in the manuscript is misleading, because it brings us back to the "chicken and egg" discussion: Which comes first, the deflection of isentropes, or the gravity wave – and is there a causal relationship between these two features? Therefore, we rephrased the respective sentence (L. 85-86). We have also added a statement listing several possible influencing factors that determine the amplitude of mountain gravity waves (L. 85-86).

3. **Difficulty of simulations:** The manuscript cites the difficulty of simulations with a 1-km grid to properly handle the interaction of the flow with the cold pool (e.g. Umek et al. 2022). This difficulty is particularly relevant to south foehn in the Alps, where colder air is typically present on the northern side *already from below the crest onwards*, not just further downstream in the lowest parts of the valleys. This difficulty therefore affects the handling of the whole descent process in the numerical simulations, which must be clearly stated in the manuscript.

We want to thank the reviewer for this comment. Picking up the input, we now explicitly mention the fact that air is typically colder already from below crest level onwards on the northern side of the Alps, making it particularly challenging to simulate south foehn events and the descent process. We added such a statement to our limitation section (L. 567ff).

4. **Gravity wave vs. hydraulic explanations** of foehn descent and distinguishing between them: This is an excellent data set, despite the uncertainties of the numerical simulations! it may still be possible to get closer to finding a definitive answer to the question of gravity wave vs. hydraulic explanations of foehn descent. Here are some specific suggestions: First, after foehn air has descended the flow response will be indistinguishable between gravity wave and hydraulic explanation. It is therefore paramount to examine the onset of the foehn descent. I envision several possibilities of doing that:
   a. Examine vertical profiles upstream and downstream of the obstacle from before onset until after the onset, similar to Mayr and Armi (2010).
   b. Alternatively, find regions where foehn has descended as well as similar topographic obstacles behind which foehn has not descended yet. What are the differences? Are the upstream conditions not similar? Note that Reinecke and Durran (1990) found extreme sensitivity to initial conditions in a 70-member ensemble simulating foehn during the TREX campaign. Descent and consequently leeward wind speeds differed despite similar upstream conditions prior to the onset of foehn. This undermines the argument in the manuscript that increased upstream wind speed would favor descent.
   c. Examine vertical sections across the obstacle from before till after onset, and also for regions where foehn does not descend much: Do the wavelengths of the gravity waves correspond to the shape of the virtual topography or to that of the real topography upstream experienced by the impinging flow (to test if your statement that incoming wind speed also plays a role; cf. Mayr and Armi, 2010)?

We acknowledge the reviewer's substantial efforts to provide these additional suggestions for further analysis! We fully agree that our dataset presents a promising opportunity to address this fundamental question: Is the descent occurring due to gravity waves, or can we explain it with hydraulic theory? Or do we need both explanations?

Before elaborating on the specific suggestions, we would like to emphasize the challenge common to all of them, adding to the complexity of drawing overarching conclusions: How to define the upstream conditions? While defining upstream conditions for simple, quasi-2D problems may be relatively straightforward, the Alps pose a 3D problem. The air parcels descending in the lee of the Falknis (refer to Fig. 7 for an overview) already crossed numerous

[Figure]

**Figure R1.** *Vertical profiles of potential temperature upstream (i.e., at Milano) and downstream (i.e., at Vaduz) of the orographic obstacle for the Feb 2017 event. Shown are different time instants.*

mountains and valleys before impinging on the Schesaplana mountain range. Additionally, the Alpine range exhibits a distinctive curvature as well. One could consider the local upstream conditions just a few kilometers to the south, or the mesoscale air mass differences, as demonstrated by Mayr and Armi (2010) for the Sierras. When comparing the vertical profiles, we adopted a similar logic as in Mayr and Armi (2010) and therefore compared the profile of Milano (9.28 °W / 45.43 °N) with that of Vaduz (see location in Fig. 7 in the manuscript).

a. We computed vertical potential temperature profiles both upstream and downstream for the two events, as illustrated in Figs. R1 and R2. Focusing on Feb 2017 (Fig. R1), we see that the upstream air mass is already colder *before* the first air parcels descend along the Rätikon (Fig. R1a; descent timeseries can be found in Fig. 8b in the manuscript). This indicates that the potential temperature difference across the Alps is not the only factor driving the descent. Ten hours later (Fig. R1b), a substantial temperature difference is discernible, consistent with the strong descent at that time. However, at the time of the strongest descent during this event (Fig. R1c), there is actually a layer of warmer upstream air at 2 km AMSL, suggesting that the descent is weaker or absent at that time. A few hours later, when we observe a temporary pause in descent activity (Fig. R1d), the downstream profile indeed became colder, which points to the importance of cross-Alpine potential temperature differences. Finally, at 13:00 UTC 28 Feb (Fig. R1e), a second peak in descent can be observed, although the differences between the upstream and the downstream profiles are rather small.

Focusing on the Apr 2018 event, the same finding as for the Feb 2017 holds true: The upstream profile is already colder before the first descent is diagnosed in the hotspot along the Rätikon (Fig. R2a). Later in the afternoon, despite pronounced potential temperature differences across the Alps, the descent activity was actually quite weak. Comparing the 20:00 UTC and the 16:00 UTC profiles (Figs. R2b,c), one would expect a stronger descent at 16:00 UTC if the potential temperature differences were the only driving factor. However, it is the other way around (see Fig. 11b in the manuscript). (see Fig. 11b in the manuscript). During the night, as nocturnal cooling formed a stable layer (Fig. R2d), the lowest levels at Vaduz are indeed colder compared to the upstream levels. The following day (Fig. R2e), we have a deep mixed layer downstream, which facilitates the descent of the colder upstream air.

[Figure]

**Figure R2.** *Same as Fig. R1 but for the Apr 2018 event.*

To conclude this first part of our response, the cross-Alpine temperature difference does partially correlate with the temporal variability of the descent. In both cases, however, the analysis shows that the upstream air was already colder before the descent began. Furthermore, there are periods when the profiles suggest descent (colder air upstream), but it does not occur, and vice versa. In essence, while the cross-Alpine potential temperature differences are undeniably linked to the descent, they do not appear to be either a necessary condition (e.g., Fig. R1c) or a sufficient condition (e.g., Fig. R2a) for descent, at least according to our dataset.

b. We acknowledge the reviewer's second idea. However, due to the challenges associated with defining a proper upstream location for trajectories traversing the 3D Alpine arc, we did not pursue this second suggestion. However, these thoughts on the onset are really interesting and worth investigating given the proper analysis framework. We leave it for future studies.

c. Taking up the reviewer's third suggestion, we plotted vertical cross section along the lines c1 and c2 for all time instants during the Feb 2017 event. We cannot show all time instants in this document, instead we will only focus on the onset of the event (Fig. R3).

At 23:00 UTC 26 Feb (Figs. R3a,b), the virtual topography formed by the 294 K and 296 K isentropes is very smooth, coinciding with virtually no across-ridge wind component. This situation results in negligible vertical motion. By 01:00 UTC 27 Feb (Figs. R3c,d), a very small across-ridge potential temperature gradient is evident. At the same time, the first gravity wave forms in the lee of the Schesaplana and a local lowering of the 296 K surface is discernible. We cannot say whether the wave causes the deformation in the virtual topography, or whether the wave results from the deformation in the topography (i.e., the local descent), as both features arise simultaneously. This corroborates our statement from the first round of revisions, namely that these two phenomena are intrinsically coupled, making it very challenging to deduce a causal relationship. Moving to 05:00 UTC, a wave begins to form in the lee of Falknis as well, being associated with a local deformation of the isentropes (Fig. R3e). Focusing on the cross section through the Schesaplana (Fig. R3f), the amplitude of the wave surpasses the height difference across the ridge formed by the virtual topography (i.e.., the 296 K isentrope), despite the presence of an across-ridge gradient in potential temperature. This points towards an active role of gravity waves in shaping the virtual topography, and hence influencing the descent. At 09:00 UTC (Figs. R3g,h), the waves are accentuated along both vertical cross

[Figure]

*Figure R3.* *Vertical cross sections of vertical wind (colormap), isentropes (gray contours) and wind along cross sections (vectors). The topography is indicated by gray shading. The left column shows the cross section c1, while the right column shows the cross section c2 (refer to Fig. 7 in the manuscript for orientation).*

sections. The wavelength of the wave in the lee of the Schesaplana approximately aligns with the shape of the real topography. However, a cold-air pool persists at the lowest levels, impeding the complete descent to the downstream valley floor.

We can provide the reviewer with another example that, in our opinion, suggests a more active role for gravity waves in the descent. For this purpose, we focus on the time period during the Apr 2018 event when we diagnosed a temporal break in the descent due to nocturnal cooling (see the manuscript for details). Figure R4 shows the temporal evolution in half-hourly steps along the cross section c3 during this night. At 02:00 UTC (Fig. R4a; this corresponds to Fig. 13b in the manuscript), the 294 K isentrope forms a smooth virtual topography that inhibits descent below 1.8 km AMSL. During the subsequent hours, a pattern of trapped lee waves emerges downstream of the Falknis peak (Figs. R4b-h). These waves are associated with a deformation of the 294 K isentrope, even in the absence of an across-ridge gradient in potential temperature along the cross sections. Based on this,

[Figure]

***Figure R4.*** *Vertical cross section showing the same fields as in Fig. R3, but for the cross section c3 (refer to Fig. 7 in the manuscript for orientation).*

we hypothesize that the gravity waves actively disturb the virtual topography (i.e., the cold air in the valley) by turbulent mixing. This disturbance facilitates the descent in the early morning hours, before bottom-up erosion by diurnal heating could play a significant role. Once again, this points towards a more active role of gravity waves in the descent process.

Wrapping up our response to the fourth reviewer's comment, we cannot draw definitive conclusions from the additional analyses we performed. The cross-Alpine potential temperature differences appear to be relevant for the descent, but do not emerge as the only decisive factor. The vertical cross sections indicate that gravity waves could also play an active role in the descent. In this light, we refrain from including any of these analyses in the manuscript, as they would only add to the already considerable length of the manuscript without providing a definitive answer regarding the causes of the descent. While this question is interesting and relevant, it is not straightforward to answer with our dataset. We acknowledge that properly addressing it would require a sophisticated framework beyond the

scope of our present study. Consequently, we leave this very interesting aspect for future research.

5. **Extracting the effects of gravity waves:** Section 5.3 on the limitations of the study states in lines 598-599 that the effects of gravity waves are difficult to extract from mesoscale NWP data. Although this statement is part of a discussion on obtaining a Lagrangian momentum budget, I think it also holds more generally. Maybe the emphasis on gravity waves in the previous parts of the article can be reduced and the focus put more strongly on the core findings of the paper - how to identify descending particles, and to examine their characteristics?!

We fully agree that this statement about the challenges of extracting gravity wave effects from NWP data applies more generally. We now emphasize this at L. 549-550. Following the suggestion to further reduce the emphasis put on gravity waves, we have removed several non-essential statements related to gravity waves in Section 4 of the manuscript:

- L. 363
- L. 368
- L. 394-395
- L. 457-458

We want to emphasize that we have already reduced the focus on gravity waves in our initial round of revisions. Notably, in Section 3, references to gravity waves have been minimized, appearing only in two instances in the revised manuscript. We therefore hope that, in combination with our additional adjustments, we have reduced the emphasis on gravity waves sufficiently, allowing the reader to focus more on the core results presented in the paper.

**Minor comments:**

1. **Lines 4-5:** Can you be quantitative about the fraction of all descents examined that are dry adiabatic and along isentropes. And also state, when and where descent is not along isentropes?

How many of the trajectories are dry-adiabatic and along isentropes? The answer to this question can be found in Figs. 6a,c in the manuscript. However, the exact quantification is difficult – as no trajectory experiences *exactly* 0 K change in potential temperature and 0 g kg$^{-1}$ change in specific humidity. If one considers the narrow intervals of $\Delta\theta \in$ [-0.5 K, 0.5 K] and $\Delta q_v \in$ [-0.5 g kg$^{-1}$, 0.5 g kg$^{-1}$], then 58% of the air parcels descend adiabatically and 86% of the air parcels descend with no major specific humidity changes. However, as these thresholds are somewhat arbitrary, we would prefer to refrain from such an explicit statement in the abstract or the manuscript.

Where is descent not along isentropes? The locations where air parcels do not descend dry-adiabatically are found in Figs. 6b,d. They are predominantly located to the south of the Alpine crest, where the impinging air mass during south foehn is still humid and oftentimes precipitation occurs during ascent, followed by evaporative cooling during descent. We now mention this in the abstract as well (L. 12ff).

[Figure]

**Figure R5.** *Temporal evolution of the mean potential temperature change (Δθ) for all descent segments (blue lines with markers) and its spread (10th to 90th percentile; blue shaded area) within two-hourly windows for the 11 events where more than 1000 trajectories were selected.*

When does the descent not follow isentropes? To this end, we would need to perform a temporal analysis of the events, which we only presented in the manuscript for the Feb 2017 and Apr 2018 events and for the hotspot region. Below one can find the temporal evolution of Δθ and Δq$_v$ for all descent segments and all events with more than 1000 trajectories (Figs. R5 and R6). No distinct temporal evolution emerges. Besides a clear event-to-event variability, some cases exhibit a diurnal cycle in Δθ, which is however not as clearly visible in Δq$_v$. To not extend the manuscript's already substantial length, we refrain from including any of these findings. Note that if the reviewer is interested, an analysis of the temporal evolution of all events (descent and its magnitude) can be found in the main author's dissertation (Jansing, 2023).

2. **Line 7:** Care needs to be used with the formulation "novel approach" here and in other parts of the manuscript. It is misleading since e.g. Miltenberger et al. (2016) and Saigger and Gohm (2022), both of which are cited in the manuscript, also used detailed trajectory analysis.

We think our approach is indeed novel in the sense that we are the first ones to systematically identify and characterize descent using trajectories, while the earlier studies either focused on warming mechanisms (Miltenberger et al., 2016) or used the trajectories more qualitatively, i.e., without an algorithm that objectively filters out the descending motion. However, to be more careful with our wording, we adjusted the manuscript accordingly. In the title, we omitted "novel" (see also below). In the abstract (L. 6), we replaced "novel" by "innovative". In the conclusions, we omitted "novel" (L. 623) or replaced it by "extensive" (L. 655).

[Figure]

***Figure R6.*** *Same as Fig. R5 but for the moisture change (△q$_v$)*

3. **Lines 67-68:** With negative buoyancy from evaporation, should there not be convection? Also: "waterfall theory" is an expression that is not commonly used. Should that expression describe a hydraulic response such as in water descending behind a weir?

We are not quite sure what the reviewer is referring to. If the reviewer is referring to downdrafts in convective cells, which are known to gain negative buoyancy through evaporative cooling, then this is indeed analogous to the "waterfall theory". The respective publications argued for an important role of evaporative cooling in the descending motion. The term "waterfall theory" refers to the visual similarity of the descending air masses from the foehn wall to a waterfall. It is mentioned in several publications on the Alpine foehn (e.g., Steinacker, 2006; Sprenger et al., 2016). We have added these references to the manuscript (L. 64) to emphasize that the term was not invented by us, but taken from the existing literature.

4. **Lines 99-100:** Delete that sentence as it is not congruent with the previous exposition of two competing explanations of foehn. It does not bolster arguments in the manuscript. "Intrinsic" means independent of factors outside of a system, in this case, outside of gravity waves. Hydraulic theory posits that gravity waves are launched as a response to air that descends.

This sentence cites a statement from an accepted paper that went through the peer-review process (Elvidge et al., 2020). We will therefore not delete it. However, we rephrased it to make it clear that this conclusion is made in Elvidge et al. (2020) and not by us (L. 92-93).

5. Please **add** information of the temporal resolution at which model output is stored. You mention in your response only that it is longer than 5 minutes.

Thanks for mentioning this. This information is actually found in Table A1 in the Appendix of the manuscript.

6. **Lines 203-205:** Please add the important information that along-level diffusion is turned off for slope angles of more than 13 degrees (in this case), as you stated in your response. Other readers, not just this reviewer, might be unaware of it.

We agree that this is important information worth mentioning. We added the information to L. 173ff.

**Textual comments:**

1. Delete "novel" from title since the publication of the article implies that this is a new approach. (And it shortens the long title a little)

We agree that the title is rather long, and therefore we deleted "novel".

2. Line 3: Delete "modern". Both theories are more than half of a century old.

Agreed, we deleted "modern".

3. Line 9: "precisely" is not needed. It implies that an accuracy metric is specified in the manuscript - which there is none.

Agreed, we omitted "precisely".

**References:**

Elvidge, A. D., Kuipers Munneke, P., King, J. C., Renfrew, I. A., and Gilbert, E.: Atmospheric drivers of melt on Larsen C Ice Shelf: Surface energy budget regimes and the impact of foehn, J. Geophys. Res.: Atmos., 125, e2020JD032463, https://doi.org/10.1029/2020JD032463, 2020.

Jansing, L.: A Lagrangian perspective on the Alpine Foehn, Ph.D. thesis, ETH Zurich, https://doi.org/10.3929/ethz-b-000619589, 2023.

Mayr, G. J. and Armi, L.: The influence of downstream diurnal heating on the descent of flow across the Sierras, J. Appl. Meteorol. Clim., 49, 1906–1912, https://doi.org/10.1175/2010JAMC2516.1, 2010.

Miltenberger, A. K., Reynolds, S., and Sprenger, M.: Revisiting the latent heating contribution to foehn warming: Lagrangian analysis of two foehn events over the Swiss Alps, Q. J. Roy. Meteorol. Soc., 142, 2194–2204, https://doi.org/10.1002/qj.2816, 2016.

Sprenger, M., Dürr, B., and Richner, H.: Foehn studies in Switzerland, in: From weather observations to atmospheric and climate sciences in Switzerland, edited by Willemse, S. and Furger, M., chap. 11, pp. 215–248, vdf Hochschulverlag AG, Zurich, https://doi.org/10.3218/3746-3, 2016.

Steinacker, R.: Alpiner Föhn – eine neue Strophe zu einem alten Lied, Promet, 32, 3–10, 2006.

---

## Author Response (AR3)

**Author's response to 3rd round of reviewer comments**
* * *
**A Lagrangian framework for detecting and characterizing the descent of foehn from Alpine to local scales**
**Lukas Jansing | Lukas Papritz | Michael Sprenger**
**Submitted to WCD, egusphere-2023-1536**
**February 8, 2024**
* * *
We appreciate the reviewer's effort and thorough examination of our additional analyses provided in our last response. However, we have chosen not to incorporate most of the suggestions into the manuscript. In our view, the first comment extends beyond the scope of the manuscript, and we partly disagree with the interpretations in both comments. In an effort to bridge the gap, and as suggested by the reviewer, we have expanded the results section to elucidate some key features of hydraulic flow observed during foehn in our two case studies. Furthermore, we have further highlighted the potential significance of the hydraulic theory in our discussion and conclusions. We hope that this compromise reflects our commitment to addressing the reviewer's last concerns. For further clarification, please refer to the detailed explanations provided below (our answers in blue; the line numbers refer to the second revised version of the manuscript).

**Major comments:**

1. **Influence of up/downstream air mass difference on descent - up/downstream profiles:** I agree that it is difficult to determine a suitable upstream location for studying the influence of potential temperature differences on foehn descent within the complex topography of the Alps. The Sierra Nevada topography studied by Mayr and Armi (2010) is much simpler. The figures R1 and R2 produced for the response confirm this. I disagree with the authors on parts of their interpretation and urge to choose different upstream and downstream locations for the vertical potential temperature profiles.
   a. **Interpretation:** I attach an annotated version of the authors' Fig. R1 (Fig. A1-R1) and added a crucial missing part: the lowest elevation over which foehn can actually flow (estimated to be around 2.2 km). Contrary to what the response states, it becomes clear that the upstream air can only descend a few hundred meters at the first time shown (Fig. A1-R1a). The next time step (Fig. A1-R1b) confirms that upstream air needs to be colder for descent to the downstream bottom. The subsequent time steps (c-e) point to a problem with the choice of the upstream and downstream locations. I marked "cap" to show the top of downstream foehn layers. They are mostly well below the lowest possible crossing elevation and indicate that the upstream had been significantly modified by the many ups and downs between Milano and Vaduz as Fig. 3 of the manuscript shows. Milano-Vaduz is actually one of the longest distances where foehn descent is modeled..
   b. **Profile locations:** Therefore I suggest redoing the figures for a downstream location within the subregion d2, where the majority of trajectories descend much closer to the Alpine crest (Fig. 3 of manuscript) and use the computed trajectories to place the upstream location about 20 km upstream of the Alpine crest. Then add and discuss the figure in the main text of the manuscript.

We thank the reviewer for his thorough consideration of our additional analyses provided in the previous response. However, we have decided not to translate the suggestions into specific changes in the manuscript and provide several reasons for this decision:

- In our last response, we presented the vertical profiles of potential temperature between Milano (upstream) and Vaduz (downstream). Based on these figures, we concluded that the cross-Alpine temperature differences only partially correlate with the temporal variability of the descent and appear to be neither a necessary nor a sufficient condition for descent to occur. The reviewer suggests that this conclusion may be influenced by our choice of upstream and downstream locations. This is surprising considering that Milano is an upstream location commonly used in many Alpine foehn studies to infer the upstream profile (e.g., Mayr and Armi, 2008; Würsch and Sprenger, 2015; Tian et al., 2024). To circumvent this challenge, the reviewer suggests that we focus on other upstream and downstream locations. However, such an approach is simply outside the scope of our manuscript, as our case studies specifically focus on a descent hotspot in the Rhine Valley and not in a valley further west. We believe that an analysis involving additional descent hotspots in other regions could be a valuable aspect of future research, as already mentioned in the manuscript's limitation section (L. 593ff) and in the conclusions (L. 662-663).

- As mentioned above, the reviewer acknowledges the significant challenge in selecting a suitable upstream location for foehn studies in the Alps. Using the online trajectories could help to deduce the regions from which air parcels originate. However, recent Lagrangian foehn studies have clearly highlighted that the upstream source regions of foehn air parcels exhibit substantial variability (e.g., Jansing and Sprenger., 2022; Jansing et al., 2022). Merely averaging the upstream positions of all air parcel trajectories would not provide a representative upstream profile, particularly considering that the origin of foehn air parcels also evolves over time (e.g., Jansing and Sprenger, 2022).

- The reviewer argues that the descent, as inferred from the potential temperature difference of the two profiles, appears to be limited to a few hundred meters at the first times shown (Figs. R1a, R2a). Our previous response does not deny this. However, if the temperature differences were the only decisive factor, we would still expect to detect some descending trajectories, as we do not prescribe the trajectories to reach the downstream bottom directly. It is plausible that none of the air parcels descend more than 500 m during these early stages of the two events, potentially falling below our detection thresholds. Nevertheless, this scenario seems unlikely given the typical undershooting of downward-accelerated air, as described in the reviewer's second comment.

In conclusion, we find that the proposed analysis framework, which compares two profiles, is too simplified for the complex, three-dimensional orography of the Alpine range. Our experimentation with this approach yielded inconclusive results with respect to the significance of the cross-Alpine potential temperature differences. Moreover, any additional analyses of other regions and with more elaborate methodology would go beyond the scope of our manuscript. Consequently, we have opted against incorporating any of these analyses into the manuscript.

2. **Hydraulic response and virtual topography:** The carefully prepared Figs. R3 and R4 (even with an inset of the cross-section location!) are a big help! Thanks! However, I disagree with their interpretation.

   a. **Descent limited to difference in potential temperature:** These figures actually show that the upstream air can only descend approximately as far as its level of neutral buoyancy on the downstream side! I marked up Fig. R3c-h and attach it as Fig. A2-R3. The obstacle I refer to In the following is between km 30 and 40 in the figure. In the first cross section (left column, Fig. A2-R3c,e,g), the 292 K isentrope marked up in orange rises upstream as time progresses, i.e. the upstream air becomes colder. Between 5 UTC and 9 UTC the upstream air becomes still colder whereas the air downstream warms as is visible by the sinking of the 290 K and 288 K isentropes close to the next obstacle. In the final time step shown at 09 UTC, the 292 K isentrope has risen just far enough to cross the obstacle and can descend fairly far downstream with accompanying higher flow speeds. In the second cross section (second column, Figs. A2-R3d,f,h), a similar upstream cooling is shown by the rise of the 294 K isentrope (marked up in orange) and needed before the air can descend further downstream. The momentum of the descent is sufficiently large to *under*shoot the level of its neutral buoyancy and thus deform the isentrope downwards closest to the slope (which is reminiscent of positively buoyant air parcels in convection *over*shooting their level of neutral buoyancy).

   b. **Waves are a response to virtual topography:** Fig. A2-R3 actually disentangles the seeming chicken-and-egg question of whether (gravity) waves modify the virtual topography or whether the waves are triggered by the virtual topography. A black outline in the second cross section of Fig. A2-R3 shows the region where the answer becomes obvious. There is no wave above the actual real topography. The wave is in response to the descending flow and the shape of the virtual topography that it causes. The wavelength corresponds to the virtual topography (an inverted peak) downstream of the real peak. For a further discussion, see Armi and Mayr (2011, section 4c) and Armi and Mayr (2015, section 3b first paragraph).

   c. **Key features of hydraulic flow**: Focusing just on the last time step and the region between km 25 and 45 in Fig. A2-R3(g, h), one finds these key features of hydraulic flow: (i) descent of the overflowing layer already ahead of the hydraulic control location (= peak), (ii) asymmetric flow across the obstacle with accelerating flow on the downstream side, (iii) a rebound in a hydraulic jump to the conditions further downstream, and (iv) a less stably stratified and slow layer on the downstream side (between 298 K and 300 K) separating the foehn flow from the flow aloft.

Parts of Figs. R3 (or R4) should also be included in the final manuscript to discuss this topic.

We again thank the reviewer for his detailed consideration of the vertical cross sections presented in our previous response. We provide specific answers to each of the three comments:

   a. We agree that the upstream air tends to cool as seen by rising isentropes in Fig. A2-R3. However, we disagree with the reviewer's interpretation of the vertical cross sections. In fact, descent is already visible in Figs. A2-R3d,f *before* the 294 K isentrope reached the obstacle's height. Furthermore, upon considering, for instance, the 298 K isentrope in Fig. A2-R3f, it becomes apparent that a small across-ridge potential temperature gradient corresponds to a considerable deformation of the isentrope downstream of the obstacle. This can hardly be attributed to the undershooting effect alone. Thus, it is plausible that the descent is not solely influenced by the across-ridge difference in potential temperature, but also by the emerging gravity wave! The same

rationale similarly applies to Fig. R4 provided in the previous response, with which we argued for a more active role of gravity waves in the descent process.

b.  As already mentioned in the previous response, this analysis does, in our opinion, *not* solve the chicken-and-egg question. The wave is anchored to the real topography and emerges *simultaneously* with the first deformation in the virtual topography (Fig. A2-R3d). While the reviewer suggests that "*The wave is in response to the descending flow and the shape of the virtual topography that it causes*", one could just as plausibly argue that "*the descending flow and the shape of the virtual topography are in response to the wave that formed*". The qualitative examination does not allow this question to conclusively be addressed. In fact, the second set of vertical cross sections in the previous response (Fig. R4) also points towards an active role of gravity waves in influencing the descent of foehn! As already outlined in the manuscript (L. 546ff) and our prior response, a more elaborated framework would be necessary to draw general conclusions regarding the driving mechanisms.

c.  We concur that some key features reminiscent of hydraulic flow are discernible in the vertical cross sections in Fig. A2-R3, but also in the manuscript (Figs. 10a,b). To adequately address this observation, we have incorporated a description of these key features in the revised manuscript (L. 398ff). Furthermore, we have further emphasized the potential significance of the hydraulic theory to explain the descent in the discussion (L. 530ff). Finally, we also added a similar statement to our conclusions (L. 625ff).

In summary, while we acknowledge certain points raised by the reviewer, we find ourselves in partial disagreement. Our analysis does, in our opinion, not confirm the hypothesis that the descent must solely be attributed to the differences in potential temperature, and that gravity waves merely form in response to the descent and the shape of the virtual topography. The qualitative nature of our analysis does not allow us to draw clear conclusions with respect to the driving mechanism. We have already taken this caveat of our study into account by adopting the scope of our manuscript. For example, we changed the title of our study, adopted the abstract and included a comprehensive limitation section, as detailed in our previous responses. Furthermore, beyond the limitations of our study, a variety of publications emphasizes the active role of gravity waves in foehn flows (e.g., Zängl et al. 2004a; Zängl et al. 2004b; Drobinksi et al. 2007; Saigger and Gohm, 2022). Giving the existing disagreement within the literature regarding the driving mechanism, we refrain from making definitive statements in either direction, as our analysis does not yield a conclusive answer. We trust the reviewer will accept our diverting view on this matter.

**References:**

Drobinski, P., Steinacker, R., Richner, H., Baumann-Stanzer, K., Beffrey, G., Benech, B., Berger, H., Chimani, B., Dabas, A., Dorninger, M., Dürr, B., Flamant, C., Frioud, M., Furger, M., Gröhn, I., Gubser, S., et al.: Föhn in the Rhine Valley during MAP: A review of its multiscale dynamics in complex valley geometry, Q. J. Roy. Meteorol. Soc., 133, 897–916, https://doi.org/10.1002/qj.70, 2007.

Jansing, L. and Sprenger, M.: Thermodynamics and airstreams of a south foehn event in different Alpine valleys, Q. J. Roy. Meteorol. Soc., 148, 2063–2085, https://doi.org/10.1002/qj.4285, 2022.

Jansing, L., Papritz, L., Dürr, B., Gerstgrasser, D., and Sprenger, M.: Classification of Alpine south foehn based on 5 years of kilometre-scale analysis data, Weather Clim. Dyn., 3, 1113–1138, https://doi.org/10.5194/wcd-3-1113-2022, 2022.

Mayr, G. J. and Armi, L.: Föhn as a response to changing upstream and downstream air masses, Q. J. R. Meteorol. Soc., 134, 1357–1369. https://doi.org/10.1002/qj.295, 2008.

Saigger, M. and Gohm, A.: Is it north or west foehn? A Lagrangian analysis of Penetration and Interruption of Alpine Foehn intensive observation period 1 (PIANO IOP 1), Weather Clim. Dyn., 3, 279–303, https://doi.org/10.5194/wcd-3-279-2022, 2022.

Tian, Y., Duarte, J. Q. and Schmidli, J.: A station-based evaluation of near-surface south foehn evolution in COSMO-1, Q. J. R. Meteorol. Soc., 150, 290–317, https://doi.org/10.1002/qj.4597, 2024.

Würsch, M. and Sprenger, M.: Swiss and Austrian Foehn revisited: A Lagrangian-based analysis, Meteorol. Z., 24, 225–242, https://doi:10.1127/metz/2015/0647, 2015.

Zängl, G., Chimani, B., and Häberli, C.: Numerical simulations of the foehn in the Rhine Valley on 24 October 1999 (MAP IOP 10), Mon. Weather Rev., 132, 368–389, https://doi.org/10.1175/1520-0493(2004)132<0368:NSOTFI>2.0.CO;2, 2004a.

Zängl, G., Gohm, A., and Geier, G.: South foehn in the Wipp Valley – Innsbruck region: Numerical simulations of the 24 October 1999 case (MAP-IOP 10), Meteorol. Atmos. Phys., 86, 213–243, https://doi.org/10.1007/s00703-003-0029-8, 2004b.